# A clinically applicable connectivity signature for glioblastoma includes the tumor network driver *CHI3L1*

Ling Hai [1,2,3,4,21], Dirk C. Hoffmann [2,3,4,21], Robin J. Wagener[2,3], Daniel D. Azorin[2], David Hausmann[2,3], Ruifan Xie[2], Magnus-Carsten Huppertz[5], Julien Hiblot [5], Philipp Sievers [6,7], Sophie Heuer[2,3], Jakob Ito[2], Gina Cebulla[2], Alexandros Kourtesakis[2,3,4], Leon D. Kaulen[2,3], Miriam Ratliff [2,8], Henriette Mandelbaum[2], Erik Jung[2,3], Ammar Jabali[9,10,11,12], Sandra Horschitz[9,10,11], Kati J. Ernst [13,14], Denise Reibold[2], Uwe Warnken[2], Varun Venkataramani[2,3,15], Rainer Will [16], Mario L. Suvà [17,18], Christel Herold-Mende[19], Felix Sahm [6,7], Frank Winkler [2,3], Matthias Schlesner[1,20], Wolfgang Wick [2,3] & Tobias Kessler [2,3] ✉

Tumor microtubes (TMs) connect glioma cells to a network with considerable relevance for tumor progression and therapy resistance. However, the determination of TM-interconnectivity in individual tumors is challenging and the impact on patient survival unresolved. Here, we establish a connectivity signature from single-cell RNA-sequenced (scRNA-Seq) xenografted primary glioblastoma (GB) cells using a dye uptake methodology, and validate it with recording of cellular calcium epochs and clinical correlations. Astrocyte-like and mesenchymal-like GB cells have the highest connectivity signature scores in scRNA-sequenced patient-derived xenografts and patient samples. In large GB cohorts, TM-network connectivity correlates with the mesenchymal subtype and dismal patient survival. *CHI3L1* gene expression serves as a robust molecular marker of connectivity and functionally influences TM networks. The connectivity signature allows insights into brain tumor biology, provides a proof-of-principle that tumor cell TM-connectivity is relevant for patients' prognosis, and serves as a robust prognostic biomarker.

Glioblastoma (GB) is the most common malignant primary brain tumor and patients have a median survival of about 15–20 months despite full standard therapy[1]. Resistance is pre-existing or acquired early and regularly, with no targeted therapy today that is effective[2]. Tumor heterogeneity plays a major role in treatment resistance, as objective responses are seen, but regrowth is fast and frequent. Although not yet related to clinical resistance, glioblastoma cells (GCs) exist in at least four main cellular states that recapitulate distinct brain cell types, are influenced by the tumor microenvironment, and exhibit plasticity[3].

We have recently discovered that long cellular protrusions named tumor microtubes (TMs) connect about half of the tumor cells to a multicellular network in GB preclinical models and patient samples[4] and also contribute to incurable pediatric glioma types[5]. Integration of malignant cells into these networks promotes resistance against radiotherapy[4], chemotherapy, and surgical lesions[6]. TM networks facilitate long-range communication of GCs by intercellular Ca$^{2+}$ waves, which is used for directed tumor self-repair and better cellular homeostasis[4,7,8]. TM networks receive synaptic neuronal input that activates glioma network communication, further driving glioma

invasion and proliferation[5,9,10]. Tumor network connectivity appears variable between individual tumors[4] and the degree of TM-connectivity is relevant for the level of resistance[4,6]. So far, little is known about single cell heterogeneity within a TM network, and only a few molecular drivers of TMs and their networks are identified[4,11,12]. Moreover, the detection and quantification of the degree of connectivity is difficult to assess in patient samples[8].

Here, we establish a gene expression signature of tumor network connectivity that improves the cellular and molecular understanding of TM network connectivity, reveals candidate structures for intervention, and proves to be a straightforward, reliable, and prognostic biomarker for this central cellular hallmark of glioma malignancy.

## Results

### Development of a connectivity signature for GB

We first explored the transcriptomic landscape of TM-connected human patient-derived glioblastoma cells (PDGCs). Three PDGC lines (PDGCLs, Supplementary Fig. 1a, b, Supplementary Table 1) over-expressing turbo green fluorescent protein (tGFP) were xenografted into mouse brains (Fig. 1a) and formed TM networks (Fig. 1b) that reflected the GB patient TM connectivity (Fig. 1b). To study the TM-mediated interconnection of PDGCs we administered the fluorescent dye Sulforhodamine 101 (SR101, Fig. 1a) that is specifically taken up by astrocytes[13] and spread to the GB network via gap-junction connections onto PDGCs´ TMs[10]. Within the GB network, TM-connected PDGCs reached higher SR101 intensities compared to TM-unconnected PDGCs (Fig. 1c, d). Importantly, SR101 intensities correlated positively with the extent of PDGC-interconnection, but not with astrocyte density (Supplementary Fig. 1c–g[11]), i.e., SR101 uptake levels of PDGCs are pivotally linked to the extent of TM-connections to vicinal PDGCs (Supplementary Fig. 1g[11,14]). Eventually, highly (tGFP[high], SR101[high]) and lowly connected (tGFP[high], SR101[low]) PDGCs, of which the vast majority resided parenchymally (Supplementary Fig. 1h, i), were separated (Supplementary Fig. 1j–l) and their transcripts subjected to RNA-Seq and scRNA-Seq (Fig. 1a).

The SR101 xenograft scRNA-Seq dataset included a total of 35,822 PDGCs with a median of 5686 PDGCs per PDGCL and SR101 intensity group and 2086 genes per cell (Supplementary Table 2). We identified differentially expressed genes (DEGs) between SR101[high] and SR101[low] groups in each PDGCL. DEGs with the same regulated direction and presence in all three PDGCLs or large fold changes in two PDGCLs were considered as connectivity related (Fig. 1e, see methods). The obtained 71 DEGs (Fig. 1f and Supplementary Data 1) included the extensively characterized TM regulators delta like canonical notch ligand (*DLL*)1 and *DLL3*[14], growth associated protein 43 (*GAP43*)[4,6,15] and the TM marker apolipoprotein E (*APOE*)[9].

245 DEGs between SR101[high] and SR101[low] groups were identified in RNA-Seq data (Fig. 1e, Supplementary Fig. 2a, and Supplementary Data 2, see methods) and the fold-changes of DEGs individually identified in the scRNA-Seq and RNA-Seq datasets were highly correlated (Supplementary Fig. 2b). 13 DEGs were mutual in both datasets (R = 0.77, Fig. 1g) and the number of overlapping genes would have been even higher if each individual PDGCL´s contribution was taken into account (Supplementary Fig. 2c). Neurogenesis related gene ontology (GO) terms were commonly enriched (Fig. 1h) in both datasets and the SR101 DEG profile further independently correlated with genes summarized in the Neurogenesis term of the gene set enrichment analysis (GSEA, Supplementary Fig. 2d). This supported that genes involved in TM network regulation are also important for neurodevelopment[4,10,12].

To quantify the degree of TM-connectivity, we calculated scores based on the aggregated expression levels of the RNA-Seq and scRNA-Seq derived gene sets. A high overall concordance between the scores of both gene sets was observed in scRNA-Seq and RNA-Seq datasets (R = 0.87 in SR101 xenograft scRNA-Seq dataset (Fig. 1i) and R = 0.89 in

The Cancer Genome Atlas [TCGA] GB RNA-Seq dataset [Fig. 1j]). Both gene sets could well-distinguish SR101[high] and SR101[low] groups (p = 0.0049, Supplementary Fig. 2e) in the SR101 xenograft RNA-Seq dataset. Strikingly, both gene sets had a high accuracy to distinguish SR101[high] and SR101[low] PDGCs, of which the scRNA-Seq score performed slightly better (0.83 vs. 0.79, Supplementary Table 3), whereas a random generated control gene set resulted in expected poor distinction (accuracy = 0.49, Supplementary Table 3).

Therefore, we decided to use the scRNA-Seq derived gene set, termed connectivity signature, for further evaluation and the term connectivity signature score to describe the extent of connectivity.

### Two distinct PDGC subpopulations are characterized by high connectivity signature scores

A proof of concept application of the connectivity signature score to the scRNA-Seq dataset expectedly demonstrated that SR101[high] PDGCs co-localized with high scoring PDGCs in the Uniform Manifold Approximation and Projection (UMAP, Fig. 2a and Supplementary Fig. 3a–c). Furthermore, a clear distinction of highly connected (SR101[high]) and lowly connected (SR101[low]) PDGCs with a strong difference in connectivity signature scores was given in all PDGCLs (Fig. 2b, c).

Recent single-cell studies have identified distinct GB cell states[3]: astrocyte-like (AC), mesenchymal-like (MES), oligodendrocyte-progenitor-like (OPC), and neural-progenitor-like (NPC). In our SR101 xenograft scRNA-Seq dataset highly connected SR101[high] PDGCs were predominantly assigned to the AC and MES1 cell states while NPC1 and OPC cell states were enriched in lowly connected SR101[low] PDGCs (Fig. 2d–f). Of note, higher connectivity signature scores of SR101[high] than SR101[low] PDGCs were found in each cell state (Supplementary Fig. 3d), rendering the connectivity signature to be a cell state-independent surrogate marker for connectivity.

Most of the 40 upregulated DEGs in highly connected PDGCs were primarily expressed in PDGCs of the AC or/and MES cell states, while the 31 downregulated DEGs mainly correlated with OPC or/and NPC cell states (Fig. 2g). We also found a high overlap between the connectivity signature genes and cell state-defining genes, in particular in the AC, MES1 and NPC1 cell states (AC 10/40, 25%; MES1 7/51, 14%; NPC1 12/51, 24%, Fig. 2h). Further highlighting a correlation of connectivity and cell states, several proven TM associated genes part of the connectivity signature, like *APOE*[9], or independent of it, like connexin 43 (*GJA1*)[4] and tweety-homolog 1 (*TTYH1*)[12], are also AC and MES cell state-defining genes[3].

To understand the fates of SR101[high] and SR101[low] PDGCs we performed RNA velocity analysis (Fig. 2i, j and Supplementary Fig. 3e, f). SR101[high] PDGCs had a transition potential towards AC. This suggested an ongoing fostering of the connected network with AC being the terminal cell state of ultimately connected PDGCs. In contrast, SR101[low] PDGCs had transition potentials toward various cell states. This might reflect their hybrid role in GB malignancy: They first invade the brain, settle in the network niche and finally serve as a seed of a multicellular network[10,14].

In summary, the SR101 xenograft scRNA-Seq data allowed us to link highly connected and lowly connected PDGCs to distinct GB cell states and provide a broad map of their transcriptomic properties. Through our interactive web app (https://connectivity-glioma.dkfz.de) we have made these data available (Supplementary Fig. 4).

### The connectivity signature score reflects cell-to-cell connections

To exclude any model-based bias, we used further methods to correlate the SR101 xenograft model-derived connectivity signature with several proven parameters of tumor cell connectivity[4,9,12,16].

Of these parameters, some are based on the ability of connected PDGCs to transfer Ca[2+] transients via TMs[4,17] (Fig. 3a and Supplementary Movie 1). Notably, the number of Ca[2+] peaks per PDGC increased

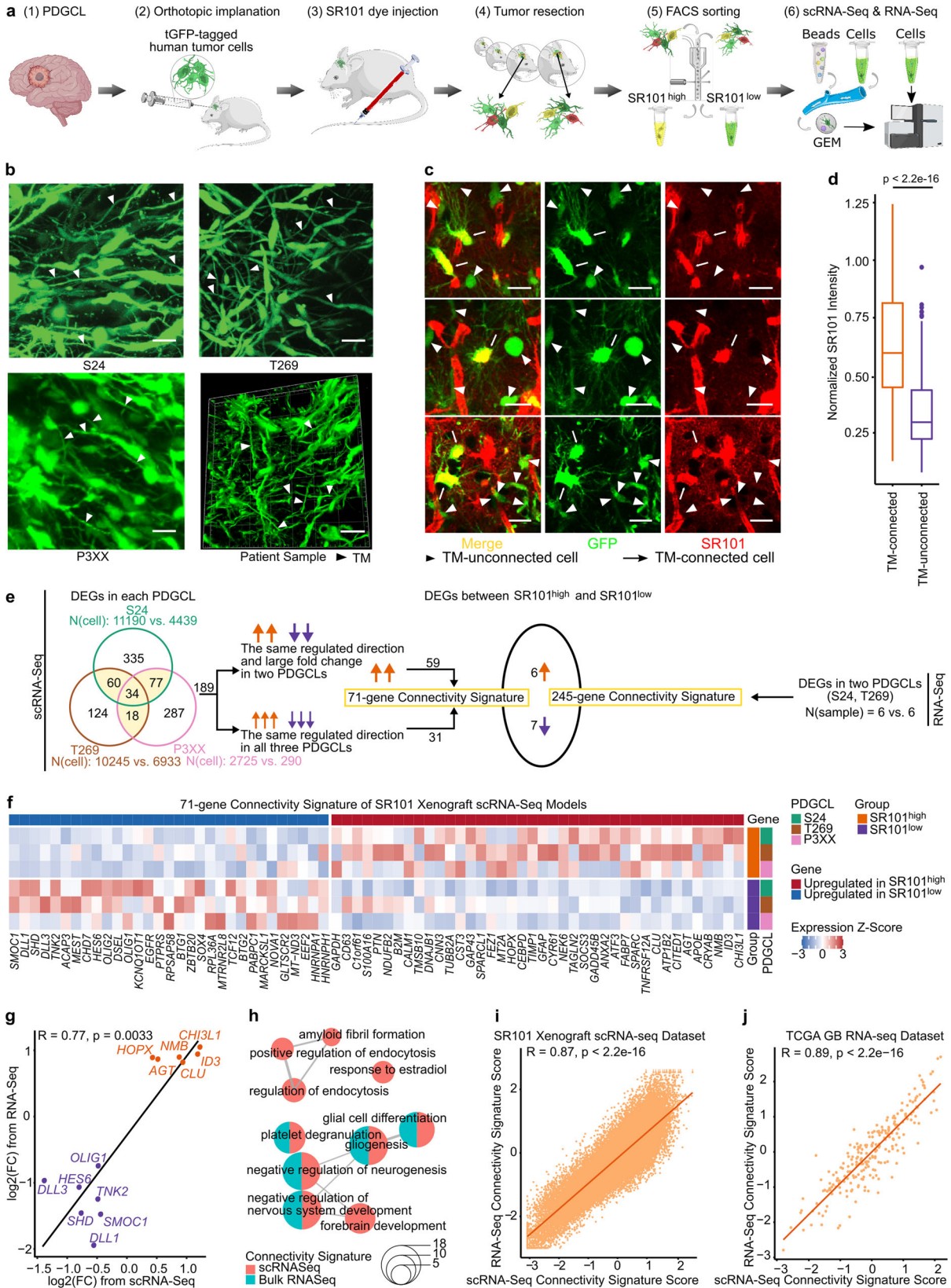

**a** (1) PDGCL  (2) Orthotopic implanation  (3) SR101 dye injection  (4) Tumor resection  (5) FACS sorting  (6) scRNA-Seq & RNA-Seq

**f** 71-gene Connectivity Signature of SR101 Xenograft scRNA-Seq Models

▶ TM-unconnected cell    → TM-connected cell

with the number of both morphological and functional (Ca²⁺ coactivity-based) cell-to-cell connections (Fig. 3b, c and Supplementary Fig. 5a, b). Based on this, we hypothesized that the accumulated Ca²⁺ level can serve as a reliable surrogate marker for PDGC connectivity and utilized to further assess the transcriptional underpinnings of differentially connected PDGCs.

To assess this hypothesis, we took advantage of Caprola₆. This fluorescent molecular recorder of calcium transients[18] enabled us to sort PDGCs in three groups based on their calcium history (i.e. low, medium, and high labeling intensities) and to perform transcriptional analysis. The same approach was conducted with PDGCs expressing a constitutively active control construct, Caprola_on, to eliminate genes

**Fig. 1 | Development of the connectivity signature. a** Experimental design of the connectivity signature development. Partly created with BioRender.com. **b** Intravital two-photon microscopy images of the xenografted tGFP over-expressing patient derived glioblastoma cell lines (PDGCLs) used for scRNA-seq, images representative of $n = 3$ mice. Bottom right; Representative confocal microscopy 3D rendering of a patient GB visualized with anti-nestin immuno-fluorescence. Arrowheads showing TMs. Scale bars depict 20 μm. **c** Two-photon microscopy images of xenografted S24 PDGCs with differential SR101 uptake (red) and constitutive tGFP expression (green), images representative of $n = 3$ mice. Arrow marks showing highly connected PDGCs and arrowheads showing lowly connected PDGCs. Scale bar depicts 20 μm. **d** Normalized SR101 intensity in highly and lowly connected xenografted S24 PDGCs. Boxes show 25th to 75th percentile, its middle line the median, whiskers the 5th to 95th percentile and individually plotted data points the outliers. $n = 287$ PDGCs (TM-connected) vs. $n = 228$ PDGCs (TM-unconnected) from $n = 5$ regions of interest (ROIs) of $n = 3$ mice. Two-tailed Mann-Whitney U test. **e** Development of the connectivity signatures. 13

differentially expressed genes (DEGs) between SR101$^{high}$ and SR101$^{low}$ PDGCs overlapped. See methods. **f** Heat map showing average expression levels of 71 scRNA-Seq-derived connectivity genes in SR101$^{high}$ and SR101$^{low}$ PDGCs from three xenografted PDGCLs. **g** Scatter plot showing the log2 fold changes of overlapping DEGs in scRNA-Seq and RNA-Seq datasets. Upregulated genes in red, downregulated genes in blue. Two-sided Spearman correlation test. **h** Enrichment map showing the most enriched GO biological processes in the scRNA-Seq-derived and RNA-Seq-derived gene sets. The pie chart size indicates the number of overlapping genes between gene sets. Lines connect GOs with overlapping genes. **i, j** Scatter plots showing connectivity signature scores based on connectivity genes derived from scRNA-Seq and RNA-Seq. Two-sided Pearson correlation test. **i**, SR101 xenograft scRNA-seq dataset. $n = 35,822$ PDGCs **j** TCGA *IDH* wt GB RNA-Seq dataset. $n = 230$ samples. **f, i, j** Values were Z-score scaled and centered across samples/PDGCs and winsorized to −3 and 3. Exact p-values are shown in the figure. Source data are provided as a Source Data file.

that might have solely contributed to differential dye uptake. Of the obtained DEGs between the three Caprola$_6$ groups and three Caprola$_{on}$ groups we excluded the few overlapping genes to establish a diffusion-corrected 171-gene calcium signature (Supplementary Fig. 5c, d and Supplementary Data 3–5).

Six genes of the calcium signature, including *CHI3L1, DLL1,* and *DLL3*, overlapped with the 71 genes comprising the connectivity signature ($p < 9.16 \times 10^{-6}$, Fig. 3d).

The calcium signature was enriched in the Neurogenesis GO term (Supplementary Fig. 5e) as was the connectivity signature (Fig. 1h). We observed a high correlation between the calcium and connectivity signature scores (R = 0.68, Fig. 3e) and the calcium signature scores distinguished SR101$^{high}$ and SR101$^{low}$ groups (Fig. 3f). MES1 and AC cell states, enriched in the SR101$^{high}$ group, were also enriched in higher calcium signature score groups, while OPC, NPC1 and NPC2 cell states predominantly occurred in the low calcium signature score group (Fig. 3g).

Of note, similar findings were obtained for the Caprola$_6$ signature, but not for the Caprola$_{on}$ signature (Supplementary Fig. 5f–k). Confirmingly, a strong positive correlation of the connectivity signature scores and labeling intensities was only found with Caprola$_6$ (Fig. 3h), but not with Caprola$_{on}$ groups (Supplementary Fig. 5l). Moreover, the connectivity signature excluded from overlapping genes with the Caprola$_{on}$ signature still showed a good distinction between SR101$^{high}$ and SR101$^{low}$ groups (Supplementary Fig. 5m). This all implied that both connectivity and calcium signature indicate TM-connectivity and Ca$^{2+}$ activity rather than unspecific dye uptake.

**Negative modulation of PDGC interconnectivity in TM-networks is accompanied by lower connectivity signature scores**
Next, we utilized BTP2 (YM-58483), a pharmacologic antagonist of the stromal interaction molecule 1 (STIM1)-regulated store operated Ca$^{2+}$ entry, to substantiate the interrelation of morphological connectivity, functional connectivity, and connectivity signature. Treatment with BTP2 resulted in lower Ca$^{2+}$ dependent labeling intensities in Caprola$_6$ expressing PDGCs (Fig. 4a and Supplementary Fig. 5n), decreased the number of TMs per live cell (Fig. 4b, c) without affecting the cell viability (Fig. 4d) and led to a reduction of the connectivity signature score (Fig. 4e).

We also demonstrated the robustness of the connectivity score in dye-independent setups. In a dense GB network[19–21] (Fig. 4f) PDGCs had more morphological connections per cell (Fig. 4g) and reached higher connectivity signature scores compared to a sparse in vitro condition (Fig. 4h). Additionally, an alternative serum-based induction of 2D TM networks (Supplementary Fig. 6a, b) also caused an increase of the connectivity signature score (Supplementary Fig. 6c).

Together, these dye-free and dye-dependent but SR101-independent methods demonstrate a meaningful interrelation of

cellular connectivity and SR101 method-derived, connectivity signature score-determined molecular connectivity.

**The connectivity signature in GB patient samples**
To test the performance of the connectivity signature in patient GB cells (GCs), 21 GB tumor samples were collected and subjected to single nucleus (sn)RNA-Seq (Supplementary Table 4). A median of 11,192 cells per sample and 995 genes per cell passed quality control, totaling in 213,444 single cells (Fig. 5a and Supplementary Table 4).

We classified malignant and non-malignant cells using copy number variation (CNV) analysis and previously defined marker genes[3,22,23] (Fig. 5a, b, Supplementary Fig. 7a–f and Supplementary Data 6). Within the malignant cells, the AC cells were predominant in most tumors although a high degree of heterogeneity in the cell states was observed between the tumors (Fig. 5c). The connectivity signature score was also heterogeneous between tumors (Fig. 5d), but consistently higher in patient GCs in AC and MES1 cell states and lower in cells in OPC and NPC cell states (Fig. 5e–g). The scoring intensities of cells in a specific cell state were retained when applying the calcium signature to our snRNA-Seq patient dataset (Supplementary Fig. 7g). This further underlined the correlation of both signatures (R = 0.55, Supplementary Fig. 7h) and corroborated the results from the xenograft mouse models. Most importantly, these results were substantiated in an independent cohort of 110 GBs[24] (Supplementary Fig. 7i–n).

It is the ultimate goal to utilize the connectivity signature for connectivity grading of clinical samples to allow for a biological start- and endpoints in clinical trials investigating anti-TM strategies. Therefore, we assessed the TM length in histological sections of tumors subjected to RNA-Seq for correlation with the connectivity signature score. Longer TMs were found in samples with high connectivity signature scores (Fig. 5h–j, Supplementary Fig. 8).

Taken together, we validated in human GB samples that particular patient GCs in AC and MES1 cell states have a high cell-to-cell connectivity. Application of the connectivity signature onto human GB specimens is feasible and could be used for reliable assessment of TM networks.

**Higher connectivity is found in tumors of the mesenchymal expression subtype and with the NF1 mutation**
After substantiating the connectivity signature in preclinical models and clinical cases, we investigated the associations between connectivity signature scores, gene mutations and expression subtypes[25] in the TCGA and Chinese Glioma Genome Atlas (CGGA) GB patient cohorts. Tumors of the mesenchymal (MS) subtype were associated with the highest connectivity signature score, while the lowest scores were observed in proneural (PN) subtype GBs (Fig. 6a). MS subtype

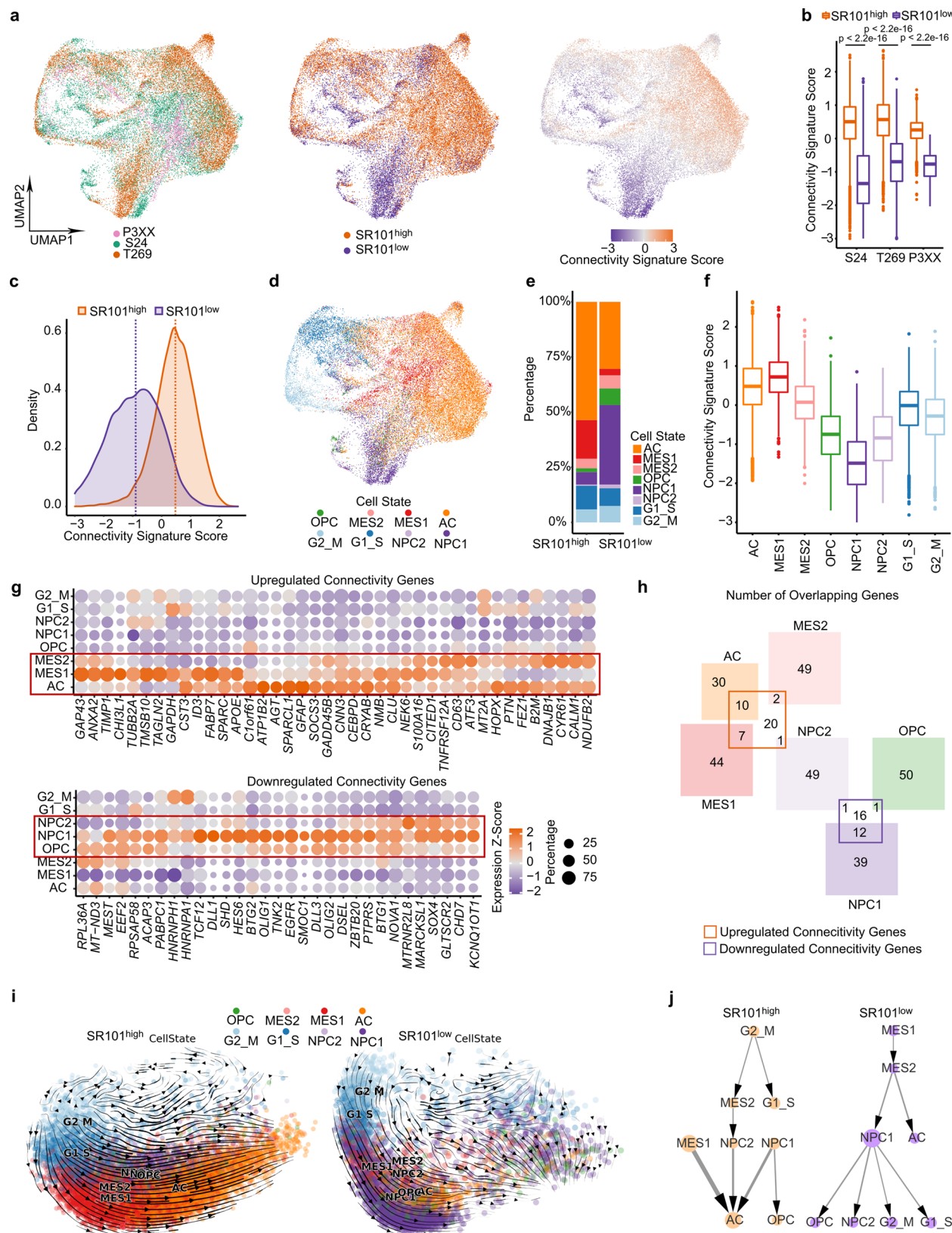

tumors correlated mainly with MES1 and AC cell states, whereas classical (CL) tumors were almost exclusively associated with the AC cell state and PN tumors showed a high frequency of OPC and NPC cell states (Fig. 6b).

Amongst the genes recurrently mutated in at least 5% of the GB patients *NF1* mutations, which are enriched in the MS TCGA

subtype[26], correlated with higher connectivity signature scores (Supplementary Fig. 9a). This association was not only valid in the whole cohort but was also observed in tumors of only the MS TCGA expression subtype (Supplementary Fig. 9a), suggesting an independence of *NF1* mutations and the MS subtype. Additionally, mutations in phosphatase and tensin homolog (*PTEN*) and tumor

**Fig. 2 | Highly and lowly connected single PDGCs correlate with distinct cell states in 35,822 single PDGCs of the SR101 xenograft scRNA-Seq dataset.** **a** Uniform manifold approximation and projections (UMAPs). Left, colored by the xenografted PDGCL. Middle, colored by SR101 intensity-based sorting. Right, colored by connectivity signature scores. **b** Connectivity signature scores in SR101[high] and SR101[low] groups of each PDGCL. $n = 11,190$ PDGCs (S24, SR101[high]) vs $n = 4439$ PDGCs (S24, SR101[low]), $n = 10,245$ PDGCs (T269, SR101[high]) vs $n = 6933$ PDGCs (T269, SR101[low]) and $n = 2725$ PDGCs (P3XX, SR101[high]) vs $n = 290$ PDGCs (P3XX, SR101[low]), respectively, from $n = 3$ mice per group. Two-sided Mann-Whitney U test. **c** Density plot of connectivity signature scores in SR101[high] and SR101[low] groups. Dotted lines depict medians. **d** UMAP of single PDGCs colored by cell states. **e** Distribution of cell states in SR101[high] and SR101[low] groups. **f** Connectivity signature scores in each cell state. **g** Dot plot of average expression levels of each connectivity gene in each cell state. Dot size indicates the frequency of cells that express the respective gene. Top, 40 upregulated connectivity genes in SR101[high] group. Bottom, 31 downregulated connectivity genes in SR101[high] group. **h** Venn diagram showing the number of overlapping genes between 71-gene connectivity signature and cell-state-defining genes. **i** RNA velocities projected on principal component analysis (PCA) embedding of xenografted S24 PDGCs. Streamline indicates the directional flow. Each dot is a single PDGC colored by cell state. **j** Directed partition-based graph abstraction (PAGA) graphs based on RNA velocity analysis in **i**. Each dot represents one cell state with the dot size indicating the number of PDGCs in the cell state. The width of the arrow indicates the transition possibility between cell states. **a–c**, **f** Connectivity signature scores were Z-score scaled and centered across PDGCs and winsorized to −3 and 3. **b**, **f** Boxes show 25th to 75th percentile, its middle line the median, whiskers the 5th to 95th percentile and individually plotted data points the outliers. Exact $p$-values are shown in the figure. Source data are provided as a Source Data file.

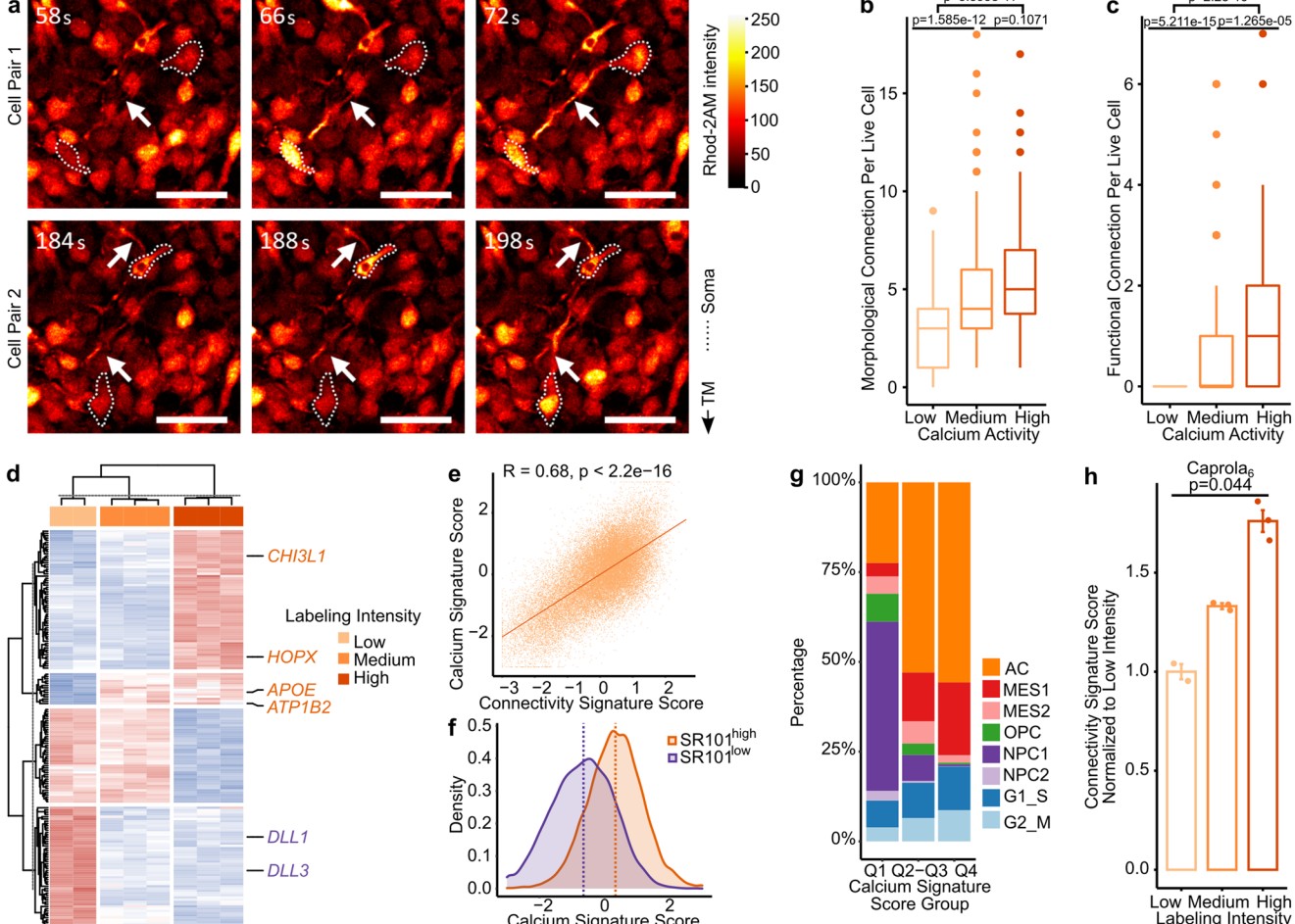

**Fig. 3 | The connectivity signature score reflects functional cell-to-cell connections. a** Time lapse micrographs of $Ca^{2+}$ transients traveling between two S24 PDGCs pairs along a TM in vitro, images representative of $n = 3$ independent experiments. Dotted lines indicate somata of TM-connected PDGCs pairs. Arrowheads indicate intercellular $Ca^{2+}$ transient traveling through TMs. Scale bars depict 50 μm. **b**, **c** Connections per S24-PDGC in low, medium and high $Ca^{2+}$ activity groups. The three groups were the PDGCs of bottom 5% ($n = 159$ PDGCs), middle 5% ($n = 100$ PDGCs) and top 5% ($n = 72$ PDGCs). $n = 3$ recordings. Two-sided Mann-Whitney U test. **b** Number of functional connections. **c** Number of morphological connections. **d** Heatmap of 171 gene calcium signature in low, medium and high labeling intensity groups of S24-Caprola[6] PDGCs. Genes overlapping with the connectivity signature are highlighted in orange (also upregulated in connectivity signature) and blue (also downregulated in connectivity signature). **e–g** Calcium signature score in SR101 xenograft scRNA-Seq dataset. **e** Scatter plot showing correlation of calcium signature score and connectivity signature scores. $n = 35,822$ PDGCs. Two-sided Pearson correlation test. **f** Density plot of calcium signature scores in SR101[high] and SR101[low] groups. Dotted lines depict medians. **g** Distribution of cell states in three groups of calcium signature score separated by first quartile (Q1), two middle quartiles (Q2-Q3) and last quartile (Q4). **h** Connectivity signature scores in RNA-Seq data of S24-Caprola[6] groups with low, medium and high labeling intensities. Shown is the mean and standard error of the mean (SEM, error bars). $n = 2$ replicates (low) vs $n = 3$ replicates (medium) vs $n = 3$ replicates (high). Two-sided Kruskal-Wallis test. **e–g** Connectivity signature scores or gene expression were Z-score scaled and centered across PDGCs/samples and winsorized to −3 and 3. **b**, **c** Boxes show 25th to 75th percentile, its middle line the median, whiskers the 5th to 95th percentile and individually plotted data points the outliers. Exact $p$-values are shown in the figure. Source data are provided as a Source Data file.

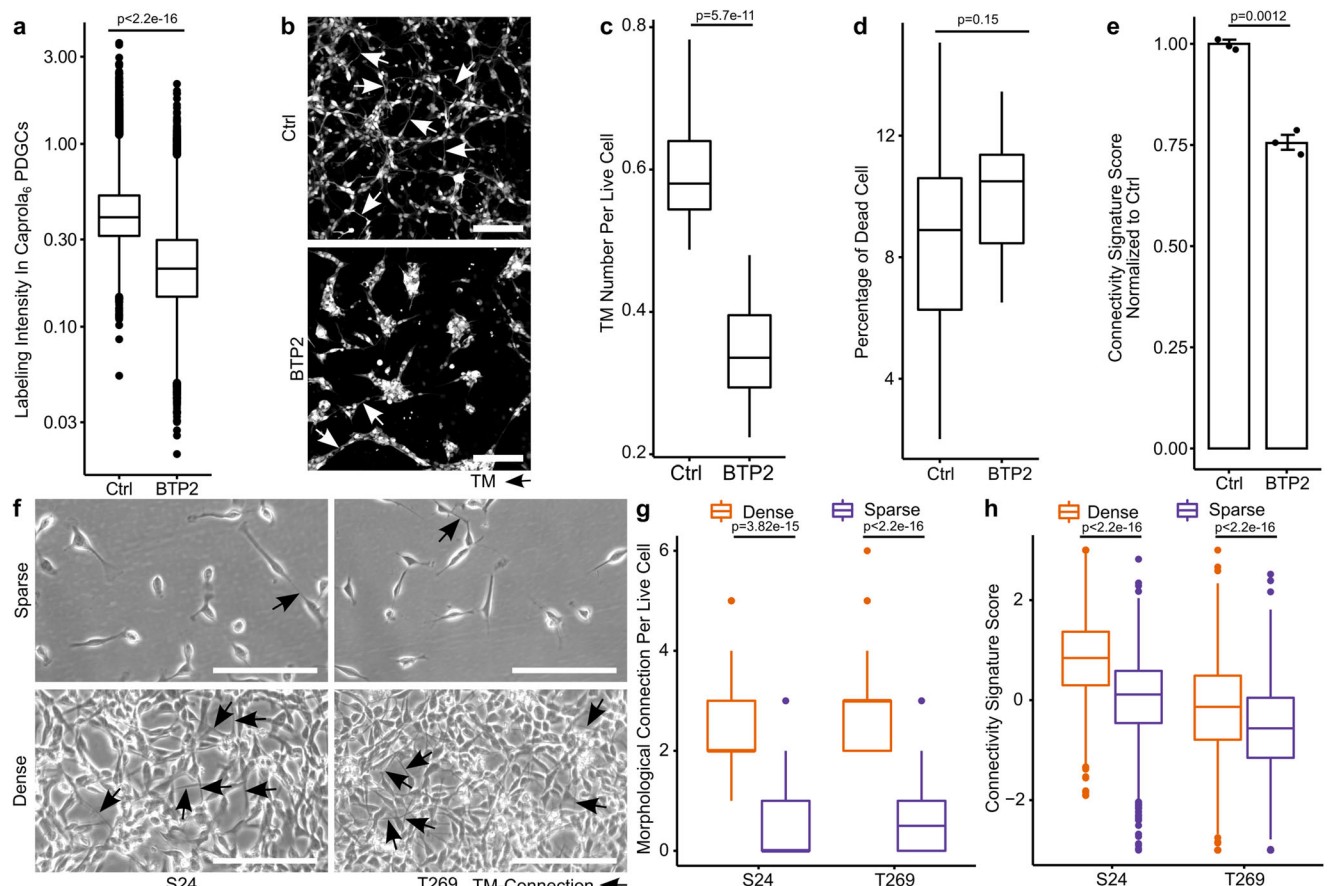

**Fig. 4 | Influences of pharmacologic perturbation and cellular density on TM-networks are reflected by connectivity signature score changes. a** Labeling intensities of FACS analyzed S24-Caprola$_6$ PDGCs after Ctrl or BTP2 treatment. $n = 21{,}792$ PDGCs (Ctrl) vs $n = 20{,}662$ PDGCs (BTP2) of $n = 2$ replicates. Two-sided Mann-Whitney U test. **b** Fluorescence micrographs of S24 PDGCs after Ctrl or BTP2 treatment, images representative of $n = 3$ independent experiments. Scale bars depict 100 μm. **c** TM number per live cell in S24 PDGCs after Ctrl or BTP2 treatment. $n = 19$ ROIs from $n = 2$ independent experiments. Two-sided Mann–Whitney U test. **d** Percentage of death after Ctrl and BTP2 treatment. $n = 19$ ROIs from $n = 2$ independent experiments. Two-sided Mann–Whitney U test. **e** Connectivity signature scores normalized to Ctrl in RNA-Seq of S24 PDGCs after Ctrl or BTP2 treatment. Shown is the mean and standard error of the mean (SEM, error bars). $n = 3$ replicates. Two-sided t-test. **f–h** S24 and T269 PDGCs grown in vitro under stem-like conditions in dense networks or sparse single clones. **f** Phase-contrast micrographs,

representative of $n = 2$ independent experiments. Scale bars depict 100 μm. **g** Number of morphological connections per live cell. $n = 50$ PDGCs in $n = 10$ ROIs from $n = 2$ independent experiments per seeding condition and PDGCL. Arrows depict connections. Two-sided Mann-Whitney U test. **h** Connectivity signature scores in scRNA-Seq data of two conditions and two PDGCLs. $n = 1150$ PDGCs (S24, dense) vs $n = 1541$ PDGCs (S24, sparse) and $n = 1347$ PDGCs (T269, dense) vs 1350 PDGCs (T269, sparse) respectively, from $n = 2$ independent experiments per seeding condition and PDGCL. Two-sided Mann-Whitney U test. Connectivity signature scores or gene expression were Z-score scaled and centered across PDGCs/samples and winsorized to −3 and 3. **a**, **c**, **d**, **g**, **h** Boxes show 25th to 75th percentile, its middle line the median, whiskers the 5th to 95th percentile and individually plotted data points the outliers. Exact p-values are shown in the figure. Source data are provided as a Source Data file.

---

protein p53 (*TP53*) were associated with the connectivity signature in the TCGA cohort (Supplementary Fig. 9a).

**Monitoring the spatiotemporal evolution of TM networks**
As sampling of tumor tissue for sequencing is mainly performed in one spot per tumor in routine analysis, we estimated how different locations in the same tumor impact the connectivity signature score. Both the connectivity signature score and *CHI3L1* expression was lower in the infiltration zone of tumors compared to the tumor core[27] (Fig. 6c). This is in line with the known higher anatomical and functional tumor cell connectivity in more solid established glioma areas[4,10,12].

To further understand longitudinal TM-network development we analyzed temporal GB specimen pairs collected from up to three re-surgeries[28] and found similar connectivity signature scores (Supplementary Fig. 9b). In general, it is recommended that specimens have a minimum tumor content of 75% for reliable molecular-based clinical GB connectivity grading using bulk methods (Supplementary

Fig. 9c). This enhances data accountability through minimizing the contribution of healthy brain cells and especially astrocytes (Supplementary Fig. 9d–f). Due to the high correlation of connectivity signature genes´ RNA and protein expression both RNA and protein-based readouts are feasible (Supplementary Fig. 9g).

**Cell-to-cell connectivity is associated with worse patient survival**
The connectivity signature score proved to be higher in GB compared to astrocytic (1p/19q intact) and to oligodendroglial (1p/19q codeleted) *IDH* mutant (mut) gliomas (Fig. 6d). This reflects previous histology-based morphological TM data[4] and eventually glioma subtype patient survival[29].

We next clarified the impact of tumor cell connectivity, determined in primary GB specimens, on GB patient survival. The shortest survival was found for patients with the highest quartile of connectivity signature score (Fig. 6e). A constant increase in the risk of death correlated with the increase of the continuous connectivity signature score in a cox proportional hazards regression model

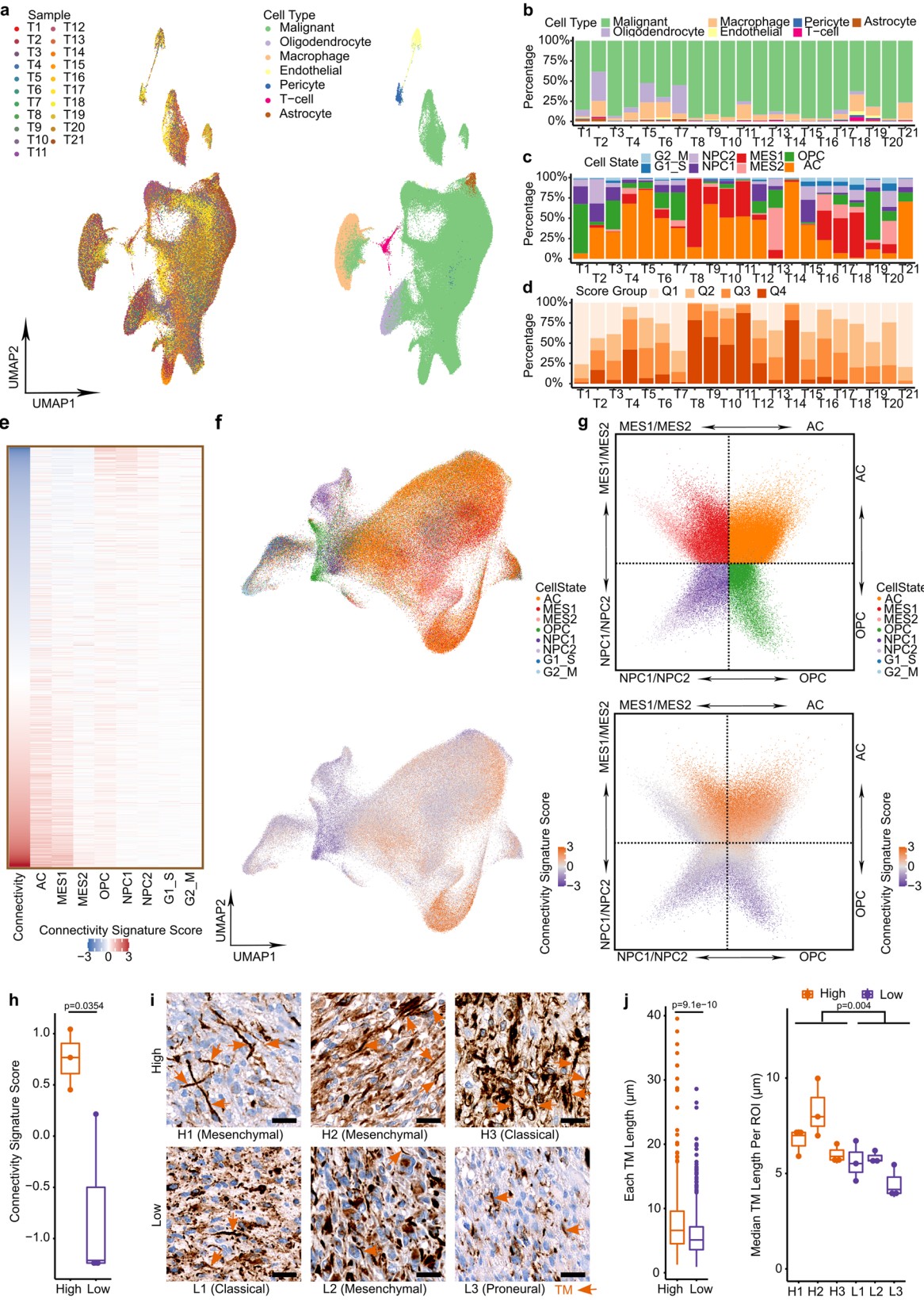

(CoxPH, Fig. 6f, g). This association remained significant after adjusting for age, gender and expression subtype in a multivariate CoxPH analysis (Fig. 6f, g). Corroborating the prognostic value of the connectivity signature, scores were also correlated with progression intervals (Fig. 6g). Likewise, these findings were similar in the recurrent setting (Supplementary Fig. 9h, i).

Together, taking advantage of the established connectivity signature, we here prove a prognostic significance for tumor cell connectivity in glioma and demonstrate its plausible link to gene expression subtypes, mutation profiles, disease subtypes and tumor localization. We show that these tools and insights can be used to discover molecular markers for this crucial hallmark of the disease.

**Fig. 5 | Connectivity signature scores and cell states in snRNA-Seq of patient samples and connectivity signature validation in GB patient sections. a** UMAP of 213,444 single cells from 21 GB patient samples. Left, colored by samples. Right, colored by cell types. **b** Frequency of malignant and non-malignant cell types in each sample. **c** Frequency of malignant cell states in each sample. **d** Frequency of connectivity signature score groups in each sample. A connectivity signature score is calculated for each cell and then assigned to one of the four score quartile groups (lower score quartile [Q1] - highest score quartile [Q4]). **e** Heat map showing connectivity signature scores and cell state signature scores in patient GCs. Each row represents one GC. **f** UMAPs of patient GCs. Top, colored by cell states. Bottom, colored by connectivity signature scores. **g** Two-dimensional representation of patient GCs according to cell state signature scores. Top, colored by cell states. Bottom, colored by connectivity signature scores. **h** Connectivity signature scores

from RNA-Seq of six patients of the N²M² pilot cohort selected for assessment of morphological tumor cell connectivity. $n = 3$ GB patients per group. One-sided t-test **i** Immunohistochemistry (IHC) staining of TMs with anti-nestin in GB patients with high (H1, H2, and H3) or low (L1, L2 and L3) connectivity signature scores, images representative of $n = 9$ ROIs from $n = 3$ patients per group. Arrows indicate TMs. Scale bars depict 20 μm. **j** Box plot of TM lengths (μm) in patients. Left, Per group. Right, Median TM lengths per ROI in each patient. $n = 454$ TMs (high) vs $n = 444$ TMs (low) of $n = 9$ ROIs in $n = 3$ patients per group. Two-sided Mann-Whitney U test. **e, g, h** Signature scores were Z-score scaled and centered across cells and winsorized to −3 and 3. **h, j** Boxes show 25th to 75th percentile, its middle line the median, whiskers the 5th to 95th percentile and individually plotted data points the outliers. Exact p-values are shown in the figure. Source data are provided as a Source Data file.

### CHI3L1 plays a pivotal role in GB and correlates with survival

The analyzes outlined above suggested a particular relevant role for *CHI3L1* in our connectivity signature (Figs. 1g, 3d). Therefore, we investigated the expression pattern and functional impact of *CHI3L1* more deeply.

High *CHI3L1* RNA expression was found to be specific for GB compared to 30 other tumor types and related normal tissues (Fig. 7a). Remarkably, due to the high correlation between mRNA and protein levels (R = 0.85, Supplementary Fig. 10a) and its nature to be a secreted protein, we recently identified CHI3L1 also as a key cerebrospinal fluid (CSF) protein biomarker for GB[30]. High *CHI3L1* RNA expression proved to be prognostic for worse overall survival (Fig. 7b) and this effect was retained in a multivariate CoxPH analysis adjusting for age and gender (Fig. 7c). There was a trend towards worse survival in 45 GB patients with high CHI3L1 protein expression in the CSF (Fig. 7d).

### CHI3L1 is a robust marker for TM network connectivity in GB

*CHI3L1* RNA expression levels were highly correlated with the connectivity signature scores. This was observed in both TCGA and CGGA datasets (R = 0.74 and R = 0.73, Fig. 8a, b). In our snRNA-Seq data of patient samples *CHI3L1* expression was high in the high connectivity score-associated MES1 and AC tumor cell populations, but low in low connectivity score-associated NPC1 and OPC tumor cell populations as well as non-malignant cell types (Fig. 8c). *CHI3L1*, that associates with the MS TCGA expression subtype[26], was highly expressed in SR101[high] compared to SR101[low] cells, particularly in MES1 but also all other cell states (Fig. 8d). This argued that *CHI3L1* is a marker for connectivity independent of the mesenchymal cell state. Moreover, *CHI3L1* expression levels positively correlated with the labeling intensity of Caprola[6] but not Caprola[on] groups (Fig. 8e). To test whether *CHI3L1* expressed areas are directly associated with cell-to-cell connected areas, we used high or low connectivity signature score tumors with particularly long or short TM protrusions, respectively (Fig. 8f–i). Even in heterogenous tumors, there was a positive correlation of CHI3L1 protein staining intensity and TM length (Fig. 8g–i and Supplementary Figs. 8, 10b).

### CHI3L1 drives TM network formation

To address high *CHI3L1* levels found in TM-connected malignant GB cells, we added a blocking antibody against CHI3L1 to PDGCs, which resulted in decreased TM lengths (Fig. 9a, b).

Encouraged by this, we also overexpressed *CHI3L1* in PDGCs (Supplementary Fig. 10c, d) and found increased TM lengths per cell in *CHI3L1* overexpression (OE) PDGCs in vitro (Fig. 9c, d). In vivo *CHI3L1* OE PDGCs retained higher *CHI3L1* expression levels compared to Ctrl PDGCs (Fig. 9e, f and Supplementary Fig. 10e), whereby the CHI3L1 signal co-localized with somata and also TMs (Fig. 9e). Within areas of equivalent GB cell density, TMs of *CHI3L1* OE PDGCs were longer, higher in number and more connected (Fig. 9g–l).

### Higher phenotypic TM-connectivity of CHI3L1 OE PDGCs is accompanied by elevated molecular connectivity

To investigate pathways involved in *CHI3L1* driven connectivity, we conduced RNA-Seq along with proteomic and phosphoproteomic mass spectrometry of Ctrl and *CHI3L1* OE PDGCLs.

Comparative analysis revealed 23 mutual DEGs (Fig. 10a, Supplementary Data 7, 8) and differentially expressed proteins (DEPs, Fig. 10b, Supplementary Fig. 10f, g, Supplementary Data 9, 10) as well as a high correlation of their fold changes (R = 0.77, Fig. 10b). *CST3*, *SPARC* and *SPARCL1*, additionally to the artificially overexpressed *CHI3L1*, were even part of the connectivity signature and their expression was found higher in RNA-Seq of both SR101[high] PDGCs (Fig. 1f) and *CHI3L1* OE PDGCs (Fig. 10a). This and the fact that these DEGs/DEPs are AC cell state defining[3] suggested their particular importance for driving TM-connectivity. Further, the *CHI3L1* regulated DEGs and DEPs increased the connectivity signature score, after exclusion of the artificially altered *CHI3L1* (Fig. 10c), and drove cell states towards AC and away from NPC1 (Supplementary Fig. 10h, i).

Among the 152 differentially phosphorylated sites of proteins (DPPs, Supplementary Data 11, 12), pSer41 and pSer154 of the known TM driver GAP43 were found elevated (Fig. 10d, Supplementary Fig. 11a, b). Interestingly, high GAP43(pSer41) levels were shown before to induce neurite branching and outgrowth[31–36] by stabilizing long actin filaments[32].

To generate a holistic overview about the mechanism of *CHI3L1*-driven TM network induction, we sought to attribute the identified DPP fingerprint to specific kinases. In line with several reports[37–39], MEK/ERK and AKT signaling were found upregulated in *CHI3L1* OE PDGCs (Supplementary Fig. 11c). We next focused on differences in the total gene expression. STAT3 was predicted to be activated and to serve as a signaling hub in *CHI3L1* OE PDGCLs (Supplementary Fig. 11d), which matched previous studies[31,40]. Interestingly, STAT3 signaling was previously associated with elevated GAP43(pS41) levels[41] – pinpointing towards a potential link between the findings made in this study.

Ultimately, the DEG, DEP and DPP-based predictions of biological functions matched the observed more connected phenotype of *CHI3L1* OE PDGCs relative to Ctrl PDGCs: Neurogenesis, Cytoskeleton organization and Cell morphogenesis related GO terms were enriched in all of the omic datasets (Supplementary Fig. 11e–g).

Together, these data identify a functional and upstream role of *CHI3L1* in governing tumor cell connectivity, *CHI3L1* RNA and protein expression as an alternative way to determine overall tumor (cell) connectivity in GB, and finally a therapeutic target for tumor network-disrupting strategies.

## Discussion

While the discovery of communicating, self-repairing and resistant TM-connected tumor cell networks has changed our understanding of incurable gliomas[42], a deeper understanding of the molecular underpinnings of TM-connected GCs has remained elusive and the assessment of tumor cell connectivity in patient samples challenging[4]. Here,

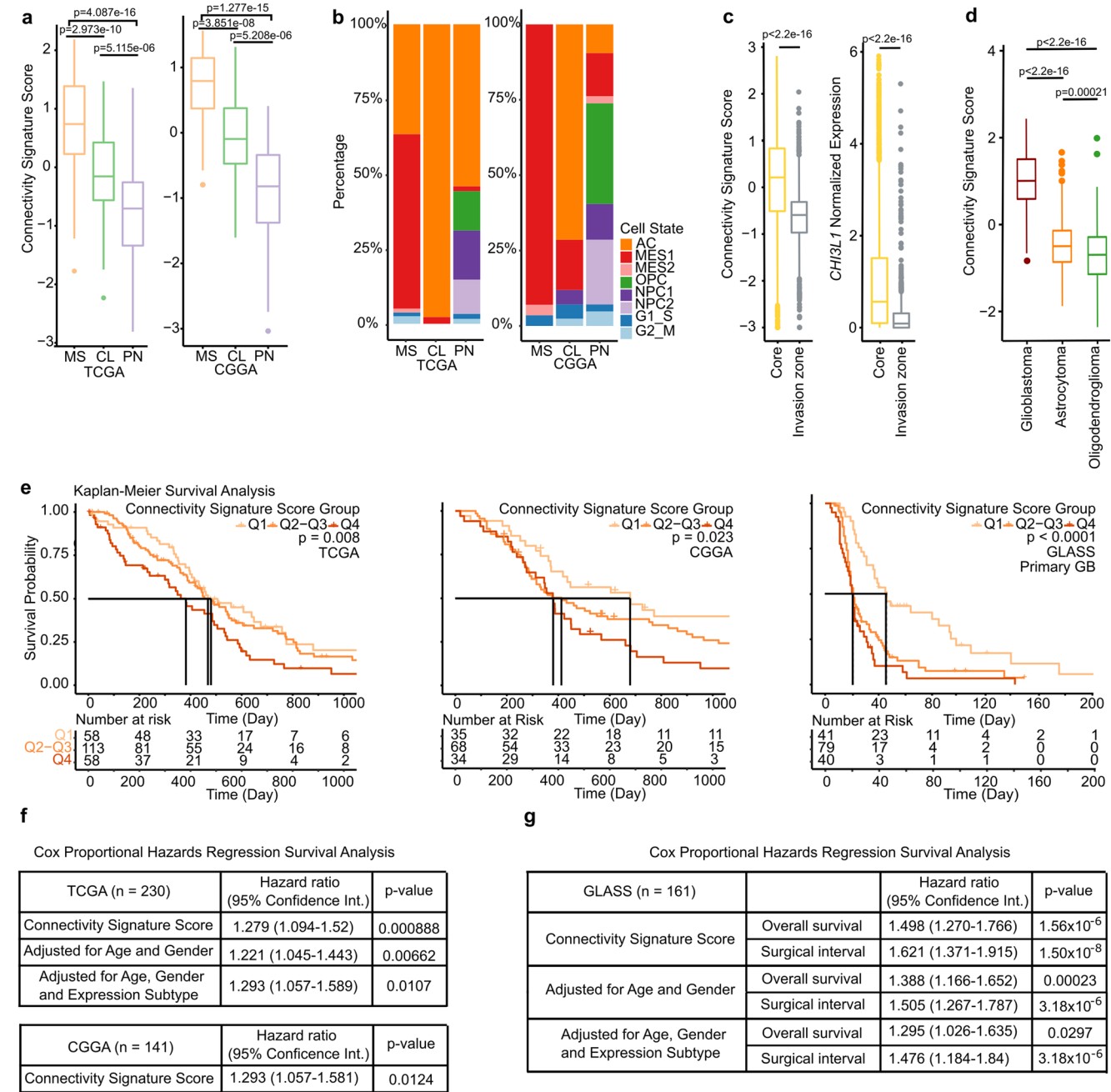

**Fig. 6 | Connectivity signature scores in TCGA and CGGA GB cohorts.**
**a** Connectivity signature scores in TCGA expression subtypes. Left: TCGA cohort tumors of MS ($n = 90$), CL ($n = 76$) and (PN, $n = 61$) subtypes. Right: CGGA cohort tumors of MS ($n = 57$), CL ($n = 42$) and PN ($n = 42$) subtypes. Two-sided Mann-Whitney U test. **b** Frequency of dominant cell states in each expression subtype. Left: TCGA. Right: CGGA. **c** Connectivity signature scores (Top) and normalized *CHI3L1* expression levels (Bottom) in cells of tumor core ($n = 3259$) and invasion zone ($n = 687$) from GB scRNA-Seq dataset[27]. Two-sided Mann–Whitney U test. **d** Connectivity signature scores in GB ($n = 230$), astrocytoma ($n = 241$) and oligodendroglioma ($n = 176$) from TCGA glioma samples. Two-sided Mann–Whitney U test. **e** Kaplan-Meier survival analysis in primary GB cohorts (Left, TCGA; Middle, CGGA; Right, GLASS) stratified into three quartile-based score groups of connectivity signature scores (lower score quartile [Q1] - highest score quartile [Q4]). Log-rank test. **f**, **g** CoxPH regression survival analysis in primary GB cohorts. Univariate analysis with continuous connectivity signature scores and multivariate analysis with connectivity signature scores adjusted for ages, genders and TCGA expression subtypes. **f** TCGA and CCGA datasets. **g** GLASS dataset. Surgical interval turned out to be prognostic. **a**, **c**, **d** Connectivity signature scores were Z-score scaled and centered across samples per cohort and winsorized to −3 and 3. **a**, **c**, **d** Boxes show 25th to 75th percentile, its middle line the median, whiskers the 5th to 95th percentile and individually plotted data points the outliers. Exact p-values are shown in the figure. Source data are provided as a Source Data file.

we established a gene expression-based connectivity signature for GB TM-networks to address these limitations.

The connectivity signature revealed yet unrelated putative TM connectivity-associated genes and also included several known neurogenesis, glioma progression and TM connectivity-associated genes,

validating its biological plausibility. High connectivity signature scores were associated with the AC and MES1 cell states and MS TCGA expression subtype, which reflects the long ago made observation that "mesenchymal development" is characterized by "cells connected via long cellular processes to a functional syncytium"[43]. However, we show

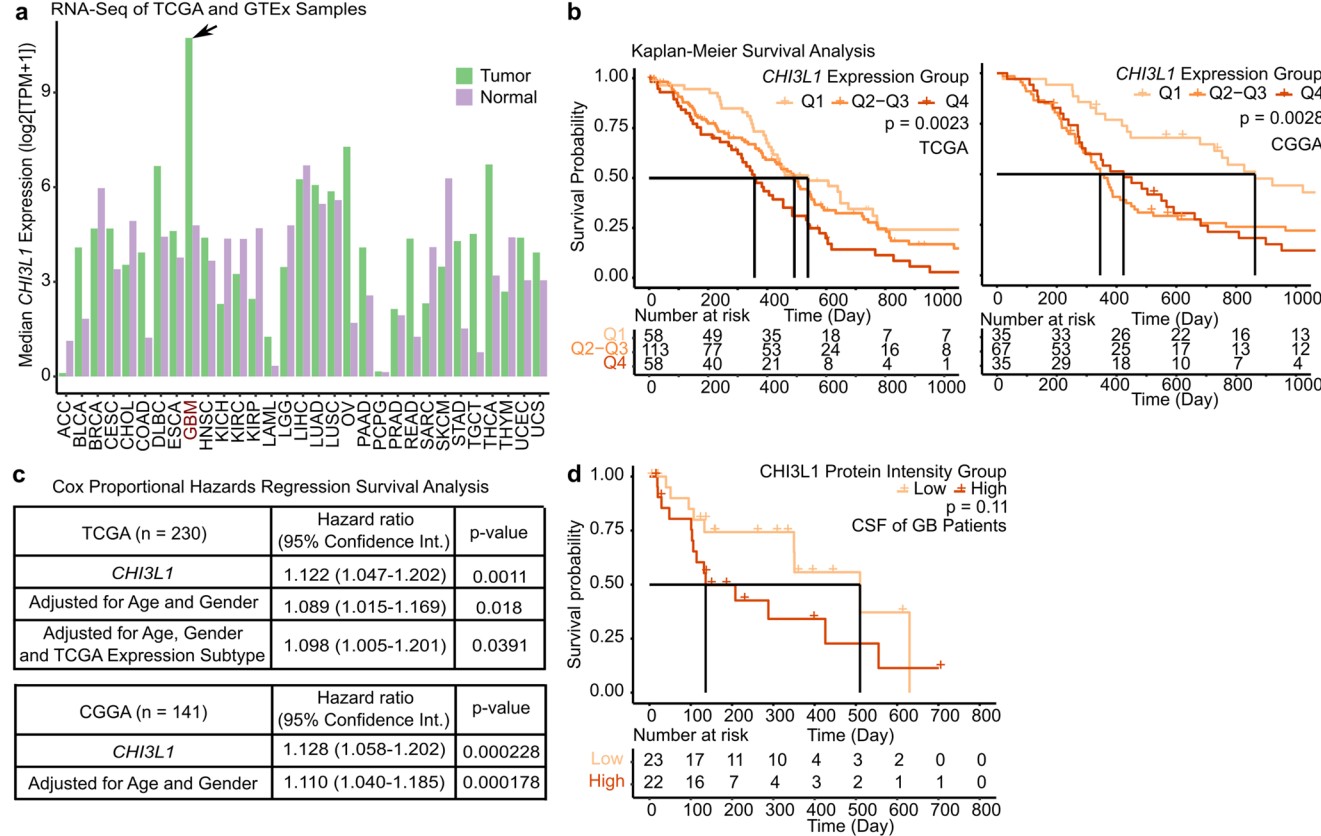

**Fig. 7 | *CHI3L1* plays a pivotal role in GB and correlates with survival. a** Median *CHI3L1* expression levels in 31 tumor types and normal tissues from GEPIA. **b** Kaplan−Meier survival analysis in cohorts stratified into three score groups of *CHI3L1* gene expression (lower score quartile [Q1] - highest score quartile [Q4]). Left, TCGA. *n* = 230. Right, CGGA. *n* = 141. Log-rank test. **c** CoxPH analysis in cohorts. Top, TCGA. Bottom, CGGA. Univariate analysis with *CHI3L1* expression levels (log2[FPKM + 1]) and multivariate analysis adjusted for ages, genders and TCGA expression subtype. **d** Kaplan−Meier survival analysis in CSF proteomics dataset according to upper and lower median-stratified half groups of CHI3L1 protein intensity. Log-rank test. Exact *p*-values are shown in the figure. Source data are provided as a Source Data file.

that the connectivity signature provides a further way to classify GBs and is independent from the MES TCGA expression subtype in terms of impacting survival and *NF1* mutations, and, on single cell level, from each cell state, including MES1 and MES2.

*CHI3L1* expression levels were highly correlated with connectivity signature scores. *CHI3L1* is upregulated in TM-connected GCs in the SR101- and Caprola$_6$ models as well as in GB patient sections. Thus, we suggest *CHI3L1* as a robust TM network marker. *CHI3L1* even functionally influences TM shape, length and number by regulating several connectivity signature genes and inducing cell state shifts towards AC and MES1. Importantly, we proved that therapeutic targeting of CHI3L1 reduces TM length.

Survival analysis showed a clear association of the tumors with the highest connectivity signature scores and worst patient outcome. This is in line with our previous preclinical findings that tumor cell connectivity is a resistance factor to all standard glioma therapies[4,6,8,42]. The connectivity signature can be applied on RNA-Seq, scRNA-Seq and proteomics datasets of various gliomas, enables the rapid and less invasive assessment of the degree of TM connectivity and, thus, serves as a prognostic biomarker in GB.

The SR101 dye transfer model has limitations[9–11]. It does not allow a strict dichotomic discrimination of existing or non-existing cellular connectivity and stress responses or even seizure-like activity in vivo might be caused when SR101 is applied topically[44]. To minimize these effects, we applied SR101 i.v. and confirmingly did not find relevant changes in cell cycling or induction of the stress-related MES2 cell state[3] in the SR101$^{high}$ group. The connectivity signature links the TM-connectivity extent and transcriptional features of all GCs, irrespective

of their spatial distribution. Thus, extremely rare cell populations with similar SR101 intensities but transcriptionally specific profiles, such as pacemaking hub GCs[17] and perivascular GCs[14], might not be accurately represented. However, the provided single-cell resolution might allow the identification of principally any cell type with the known transcriptional profile.

In addition, we implemented several orthogonal methods to assure that changes in the connectivity signature score are accompanied by anatomical and functional changes of TM networks. Most importantly, we confirmed the main connectivity signature genes including *CHI3L1* with a separate Ca$^{2+}$ level-dependent method that serves as a surrogate marker for functional TM-network communication. Additionally, respective controls proved that dye uptake had a neglectable impact on both calcium and connectivity signatures. Of note, in this manuscript we focused on TMs and have not correlated the connectivity signature or underlying readout parameters with features of tunneling nanotubes (TNTs)[15].

Other limitations of this study are related to the tumor heterogeneity which is not accounted for when samples for (sc)RNA sequencing are only collected at one part of the tumor. Experimentally unveiling this aspect, lower connectivity results in the outer part of tumors compared to the core.

The discovered association of connectivity signature expression and specific mutations is exciting and paves the way for follow-up analyzes to shed further light on the functional consequences and relative impact of each of these mutations. Likewise, further functional experiments are warranted to shed light on the suggested key players being mechanistically involved in *CHI3L1*-driven connectivity.

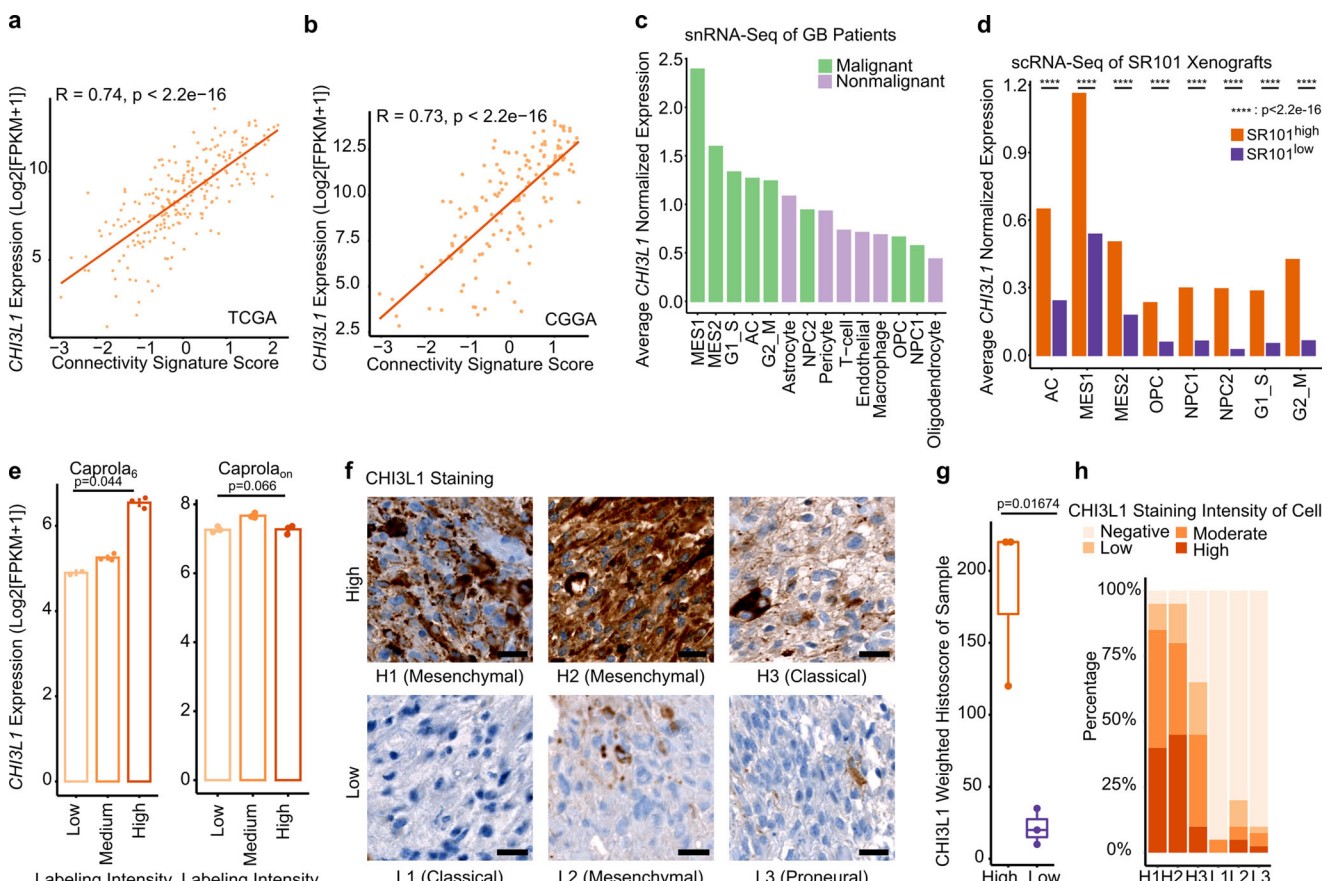

**Fig. 8 | CHI3L1 is a robust marker for connectivity. a, b** Scatter plots showing correlation between *CHI3L1* expression level and connectivity signature scores. Connectivity signature scores were Z-score scaled and centered across samples and winsorized to −3 and 3. Two-sided Pearson correlation test. **a** TCGA **b** CGGA **c** *CHI3L1* expression levels in malignant cell states and non-malignant cell types from our snRNA-Seq dataset of 21 GB patient samples. **d** Average normalized *CHI3L1* expression per cell state in SR101high and SR101low xenografted PDGCs. $n = 12,955$ (AC, SR101high) vs $n = 3542$ (AC, SR101low), $n = 4,234$ (MES1, SR101high) vs 348 (MES1, SR101low), $n = 1038$ (MES2, SR101high) vs $n = 689$ (MES2, SR101low), $n = 421$ (OPC, SR101high) vs $n = 873$ (OPC, SR101low), $n = 1370$ (NPC1, SR101high) vs $n = 4212$ (NPC1, SR101low), $n = 127$ (NPC2, SR101high) vs $n = 195$ (NPC2, SR101low), $n = 2587$ (G1_S, SR101high) vs $n = 936$ (G1_S, SR101low), $n = 1428$ (G2_M, SR101low), $n = 867$ (G2_M, SR101high) vs $n = 936$ (G2_M, SR101low) PDGCs from $n = 3$ mice per group. Two-sided Mann-Whitney U test. **e** *CHI3L1* expression levels in different labeling intensity groups of S24-Caprola6 and S24-CaProLaon. Shown is the mean and standard error of the mean (SEM, error bars). $n = 2$ (Caprola6, low) vs $n = 3$ (Caprola6, medium) vs $n = 3$ (Caprola6, high) replicates. $n = 3$ replicates (Caprolaon). Two-sided Kruskal-Wallis test. **f** IHC staining with anti-CHI3L1 in patients with high (H1, H2, and H3) and low (L1, L2, and L3) connectivity signature scores, images representative of $n = 9$ ROIs from $n = 3$ patients per group. Scale bars depict 20 μm. **g, h** CHI3L1 staining intensities. **g** Per group. Box plot of weighted histoscores. Boxes show 25th to 75th percentile, its middle line the median, whiskers the 5th to 95th percentile, and individually plotted data points the outliers. $n = 3$ patients per group. One-sided t-test. **h** Frequency of CHI3L1 staining intensity of cells. Exact p-values are shown in the figure. Source data are provided as a Source Data file.

In conclusion, we developed a connectivity signature with a respective score calculation for GBs and identified biologically plausible markers for further investigation. *CHI3L1* has emerged as an easy to assess marker of the signature with a functional relevance for TM-formation that can even be determined in standard paraffin sections and has promise to be potentially targetable in GB. This allows to translate the recent fundamental insights into key elements of tumor biology in GB into clinical trials and ultimately into clinical practice.

## Methods

### Ethics statement

This research complies with all relevant ethical regulations.

All animal research experiments conducted in this study were approved by the local authorities (Regierungspräsidium Karlsruhe, Germany, #G110/21, #G132/16, #G210/16) and compliant with the institutional laboratory animal research guidelines. All efforts were made to minimize animal suffering and to reduce the number of animals used according to the 3 R's principles. Mice were maintained in a specific-pathogen-free, standardized environment with $22 \pm 2$ °C temperature, $55 \pm 10\%$ humidity, 12 h light/dark cycles and fed with a

standard diet according to the German Cancer Research Center guidelines. Tumors were grown until the mice showed first symptoms or ≥ 20% weight loss were met. In none of the experiments these limits were exceeded.

All patients from which FFT and FFPE preserved GB specimens were used in this study gave informed consent in written form either prior to inclusion to the NCT Neuro Master Match (N²M²) pilot study[45] or exploratory molecular analyzes. The research is conducted in concordance with the declaration of Helsinki and was approved by the Ethics Committee at the University of Heidelberg, Germany (applications 206/2005 and AFmu-207/2017). The N²M² pilot study included patients with *MGMT* promoter unmethylated tumors, leading to an enrichment of *MGMT* promoter unmethylated samples in our analysis (18/21, 86%, median age is 61 years). Patient selection was solely based on tissue and metadata availability as well as quality parameters of the related tumor specimens.

### Definition for morphological TMs and quantification methods

TM: Cellular protrusions of lengths ≧ 500 nm and thicknesses between 0.5–2.5 μm were defined as TMs as described before[4,46].

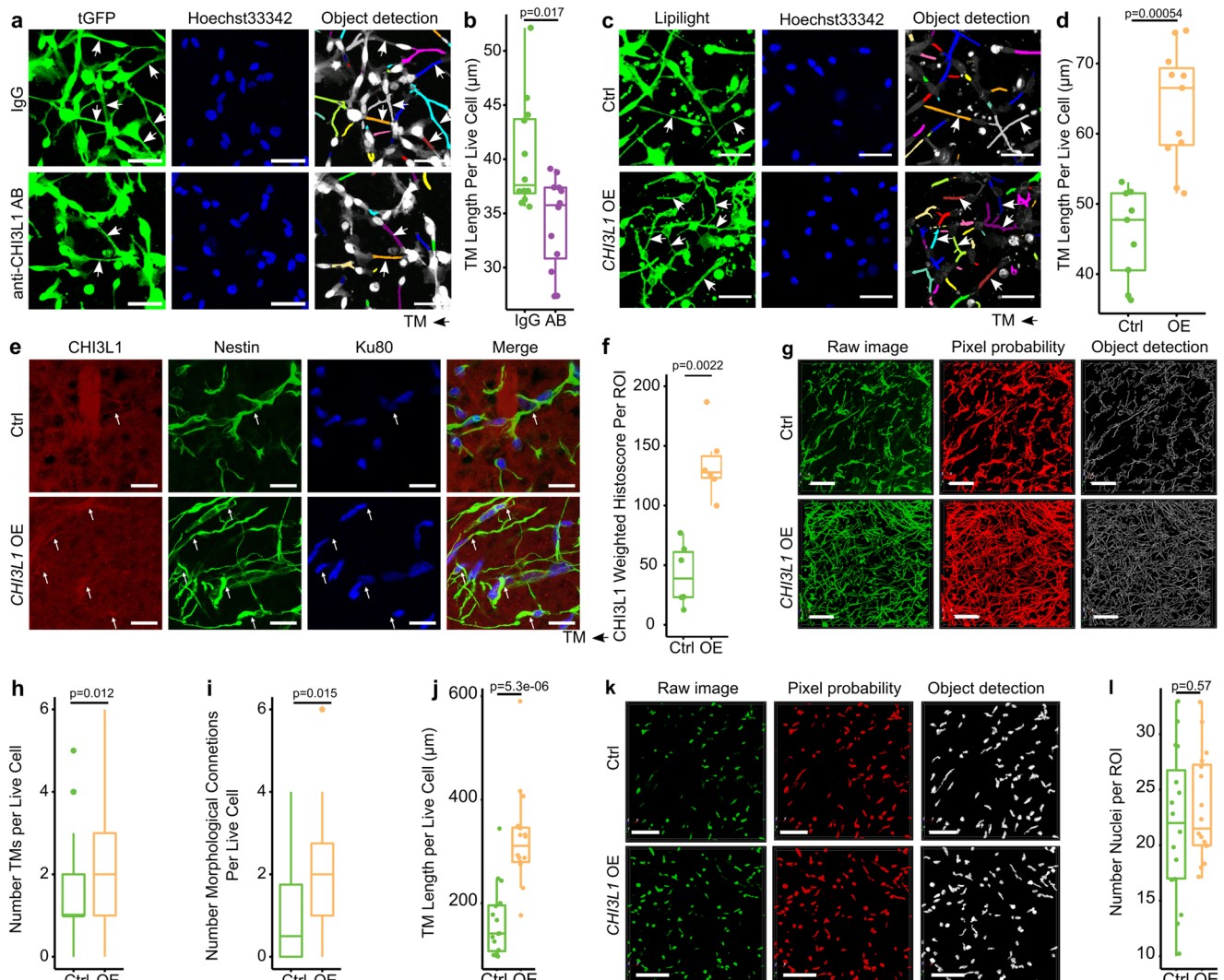

**Fig. 9 | CHI3L1 is a driver gene of TM connectivity. a** Fluorescence micrograph of S24 PDGCs treated with IgG or anti-CHI3L1 antibodies in vitro. tGFP (green) for TM visualization, Hoechst33342 (blue) for nuclei normalization and quantified objects (multicolor). The scale bar depicts 50 μm. **b** TM length per live cell after administration of IgG or anti-CHI3L1 antibodies. *n* = 12 ROIs of *n* = 2 independent experiments. Two-sided Mann-Whitney U test. **c** Fluorescence micrograph of S24 Ctrl and *CHI3L1* OE PDGCs in vitro. Lipilight (green) for TM visualization, Hoechst33342 (blue) for nuclei normalization, and quantified objects (multicolor). Scalebar depicts 50 μm. **d** TM Length per S24 Ctrl and *CHI3L1* OE PDGCs in vitro. *n* = 9 ROIs (Ctrl) vs *n* = 11 ROIs (*CHI3L1* OE) of *n* = 2 independent experiments. Two-sided Mann-Whitney U test. **e** Immunofluorescence micrograph of xenografted S24 Ctrl and *CHI3L1* OE PDGCs with anti-CHI3L1 (red), anti-nestin (green, TM marker) and anti-Ku-80 (blue, nuclear marker). **f** Weighted histoscores of xenografted S24 Ctrl

and *CHI3L1* OE PDGCs. *n* = 6 ROIs in *n* = 3 mice. Two-sided Mann-Whitney U test. **g** 3D micrographs of nestin-stained TMs of xenografted S24 Ctrl or *CHI3L1* OE PDGCs. Scalebars depict 100 μm. **h–j** TM-network parameters of S24 Ctrl and *CHI3L1* OE xenografted PDGCs. *n* = 38 PDGCs in *n* = 16 ROIs in *n* = 4 mice. Two-sided Mann-Whitney U test. **h** TM Number. *n* = 38 PDGCs. **i** Number of TM connections. *n* = 38 PDGCs. **j** TM Length. *n* = 16 ROIs. **k** 3D micrographs of Ku80-stained nuclei of xenografted S24 Ctrl or *CHI3L1* OE PDGCs. Scalebars depict 100 μm. **l** Number of nuclei per ROI. *n* = 16 ROIs in *n* = 4 mice. Two-sided Mann–Whitney U test. **a**, **c**, **e**, **g**, **k** Representative micrographs and the quantifications derived from the images match in terms of the number of independently performed experiments. **b**, **d**, **f**, **h–j**, **l** Boxes show 25th to 75th percentile, its middle line the median, whiskers the 5th to 95th percentile and individually plotted data points the outliers. Exact p-values are shown in the figure. Source data are provided as a Source Data file.

Cell-to-cell connection: Tumor cells extending TMs with at least one terminal end at the bodies or TMs of other tumor cells were classified as connected, and cells without any connection as unconnected[4,12].

## Sample preparation for IHC and confocal microscopy of human tissue

Human biopsy tissue was chemically immersion fixed with 4% paraformaldehyde (PFA; #sc-281692, Santa Cruz, Santa Cruz, California, USA) overnight. Afterwards, the tissue was washed in PBS and cut to 100 μm-thick tissue sections using a VT000S vibratome (Leica, Wetzlar, Germany). Afterwards, sections were stored in PBS (#D8537-500ML, Sigma, part of Merck, Darmstadt, Germany). To visualize tumor cells an antibody tandem panel consisting of a 1:500 diluted mouse anti-nestin

primary (#ab22035, clone 10C2, RRID:AB_446723, Abcam, Cambridge, United Kingdom) and Alexa Fluor™488-conjugated anti-mouse secondary (#A11001, RRID:AB_2534069, Thermo Fisher Scientific, Waltham, Massachusetts, USA) antibody was used. Sections were counterstained with DAPI (#6335.1, Roth, Karlsruhe, Germany). Images were acquired on a TCS SP8 confocal laser-scanning microscope (Leica, Wetzlar, Germany) using a 63x (NA 1.4) oil objective. Images were acquired with a pixel size of 141 nm and 300-nm z steps.

## Cell culture of PDGCLs

PDGCLs S24 and T269 were established from freshly dissected GB tissue from adult patients after informed consent[47] and kindly provided by Dieter Lemke, German Cancer Research Center, Heidelberg

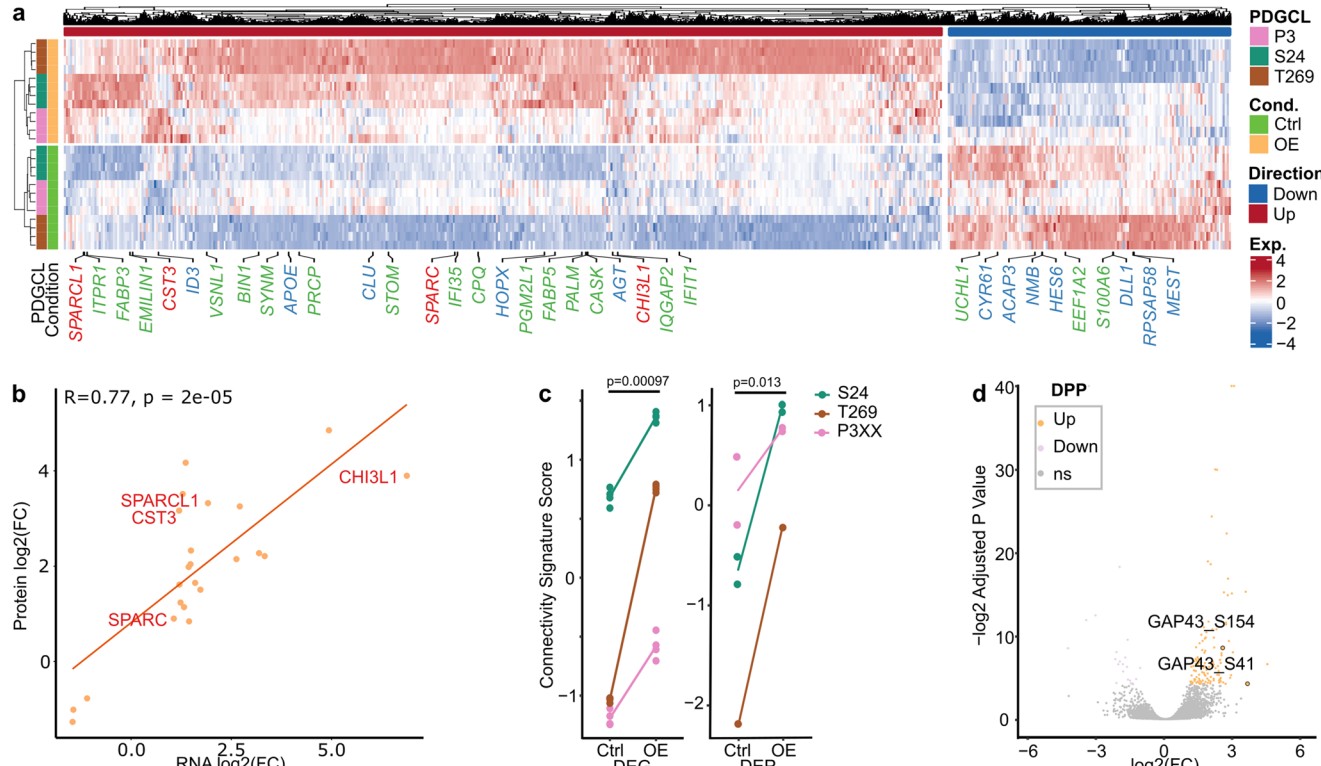

**Fig. 10 | Omics-based molecular fingerprinting of CHI3L1 OE PDGCLs.**
**a** Heatmap showing DEG average expression levels of Ctrl and *CHI3L1* OE PDGCLs. Data were Z-score scaled and centered across samples, and winsorized to −3 and 3. Colors: Purple, downregulated overlapping genes with connectivity signature score; orange, upregulated overlapping genes with connectivity signature score; grey, overlapping genes with DEPs. **b** Scatter plot showing the log2 fold changes of DEGs and DEPs in RNA-Seq and proteome datasets. 23 overlapping genes were found. Overlapping gens with the connectivity signature are depicted red. Two-sided Spearman correlation test. **c** Connectivity signature scores derived from RNA-Seq and proteomics datasets of Ctrl and *CHI3L1* OE PDGCLs. Lines indicate the average in each PDGCL. *n* = 4 independent replicates per PDGCL in RNA-Seq. *n* = 2 independent replicates (S24 and T269) and *n* = 1 independent replicate (P3XX) in proteomics. Two-sided paired t-test. **d** Volcano plot comparing DPPs of Ctrl and *CHI3L1* OE in PDGCLs P3XX, S24 and T269. Phosphosites shown in orange or purple have adj. *p*-value < 0.05. Phosphorylation sites of GAP43 are depicted. $P_{adj}$ is the adjusted value statistically corrected for multiple testing. Exact p-values are shown in the figure. Source data are provided as a Source Data file.

University. PDGCLs P3XX and BG5 were kindly provided by Hrvoje Miletic, K. G. Jebsen Brain Tumour Research Centre, University of Bergen[48]. All four tumors have been diagnosed as GB, *IDH* wt.

All PDGCLs were cultured as neurospheres under serum-free, non-adherent, stem-like conditions in PDGCL media, consisting of DMEM/F-12 (#11330-032, Life Technologies, part of Thermo Fisher Scientific, Waltham, Massachusetts, USA), B27 supplement (#17504044, Life Technologies, part of Thermo Fisher Scientific, Waltham, Massachusetts, USA), 5 μg/ml insulin (#I9278, Sigma, part of Merck, Darmstadt, Germany), 5 μg/ml heparin (#H4784, Sigma, part of Merck, Darmstadt, Germany) 20 ng/ml epidermal growth factor (EGF; #PHG0311, Life Technologies, part of Thermo Fisher Scientific, Waltham, Massachusetts, USA) and 20 ng/ml fibroblast growth factor (FGF; #PHG0021, Life Technologies part from Thermo Fisher Scientific, Waltham, Massachusetts, USA).

Methylation profiling with the methylation EPIC array (#WG-317-1003, Illumina, San Diego, California, USA) was employed to confirm GB origin and the brain_classifier_v12.8 workflow (https://www.molecularneuropathology.org/) used for further data analysis. Briefly, S24 is characterized by a GB receptor tyrosine kinase (RTK) I, whereas BG5, P3XX, and T269 exhibit a GB RTK II methylation subtype (Supplementary Table 1[49]). Mutational fingerprints derived from whole-exome sequencing (Supplementary Fig. 1b) as well as the highest scoring TCGA expression subtype[25] and cell state[3] derived from RNA-Seq differed between PDGCLs (Supplementary Table 1).

In order to allow identification and re-isolation after tumor resection, PDGCs were lentiviral transduced with the MISSION® shRNA

pLKO.1-puro-CMV-tGFP_shnon-target (#SHC016, Sigma, part of Merck, Darmstadt, Germany) vector for cytosolic tGFP expression.

HEK293FT cells (#R70007, Thermo Fisher Scientific, Waltham, Massachusetts, USA) were co-transfected with *tGFP* lentiviral expression constructs and 2nd generation viral packaging plasmids VSV.G (kind gift from Tannishtha Reya, Addgene plasmid # 14888, RRID:Addgene_14888, http://n2t.net/addgene:14888) and psPAX2 (kind gift from Didier Trono, Addgene plasmid #12260, RRID:Addgene_12260, http://n2t.net/addgene:12260). 48 h after transfection, virus-containing supernatant was removed and cleared by centrifugation (5 min/500 g). The supernatant was passed through a 0.45 μm filter (#760517, Ahlstrom, Helsinki, Finland). PDGCLs were transduced with lentiviral particles at 70% confluency in the presence of 10 μg/ml polybrene (TR-1003-G, Merck, Darmstadt, Germany). 24 h after transduction successfully transduced PDGCs were selected with 1 μg/ml puromycin (#A2856.0100, Applichem, Darmstadt, Germany) and subjected to FACS sorting.

All PDGCLs were regularly checked for authenticity and absence of infections, such as mycoplasms and non-human cell contamination, as part of the multiplex cell contamination test (Multiplexion GmbH, Heidelberg, Germany).

## RNA-Seq
For RNA isolation, cells were washed with ice-cold PBS (#D8537-500ML, Sigma, part of Merck, Darmstadt, Germany) and lysed in the dish by addition of 1% beta-Mercaptoethanol (#M3148-100ml, Sigma, part of Merck, Darmstadt, Germany)-supplemented RLT lysis buffer as part of the QIAGEN RNeasy Mini Kit (#74004, Qiagen, Hilden,

Germany). All downstream steps for RNA isolation were conducted according to the manufacturer´s recommendation. On-column DNAse digestion was performed with the RNAse free DNAse set (#79254, Qiagen, Hilden, Germany). RNA was eluted into RNAse-free water (#4387936, Thermo Fisher Scientific, Waltham, Massachusetts, USA) and RNA integrity validated using the RNA Screen Tape System (#5067-5576, #5067-5577, #5067-5581, Agilent, Santa Clara, USA) and the 4150 Tapestation System (#G2992AA, Agilent, Santa Clara, California, USA) according to the provider´s recommendations. Library preparation and RNA sequencing on a NovaSeq6000 device (Illumina) was carried out by the GPCF at the DKFZ.

## Whole exome sequencing of PDGCLs

DNA and RNA extraction. Genomic DNA was extracted using the DNA/RNA/Protein Mini Kit (#80004, Qiagen, Hilden, Germany) according to the manufacturer's instructions. 4150 Tapestation (#G2992AA, Agilent, Santa Clara, California, USA) was used to determine DNA integrity numbers.

Whole exome sequencing (WES). WES was carried out by the GPCF. Briefly, Exome capture was performed with the Agilent SureSelect XT HS + Human All Exon V7 kit (#5191-4028, Agilent, Santa Clara, California, USA) and the libraries were paired-end sequenced on a NovaSeq6000 device (Illumina, Santa Clara, California, USA).

Data processing. The WES data from cell lines were aligned using BWA (v.0.7.15) against the human genome (1KGRef_PhiX). Reads duplication was marked using Sambamba (v.0.6.5). The parameter settings were based on the DKFZ ODCF workflow (https://github.com/DKFZ-ODCF/AlignmentAndQCWorkflows). Somatic short variants (SNVs + Indels) were identified using the GATK (v.4.2.0.0) pipeline: Variants were called using Mutect2, with the omission of variants in germline and a panel of normals (germline resource and panel of normals were downloaded from gatk-best-practices/somatic-b37). Cross-sample contamination was estimated using GetPileupSummaries and CalculateContamination. Orientation bias artifacts were learned using LearnReadOrientationModel. Then, variants were filtered using FilterMutectCalls and annotated using Funcotator. Non-silent somatic variants in neuro-oncology relevant genes[50] were retained.

## Two-photon microscopy of astrocytes, PDGCLs and correlation of SR101 staining with tumor cell connections

For the preparation of two-photon microscopy experiments, a chronic cranial window was implanted into 8–10 week old male Crl:NMRI-*Foxn1^nu* nude (RRID:MGI:5653040; Charles River, Wilmington, Massachusetts, USA) mice as described[4,9]. At least 10 days after implantation of the chronic cranial window, the glass was temporarily removed to cortically inject $5 \times 10^4$ viable PDGCs and establishment of solid tumors allowed. $n = 3$ mice were used per PDGCL and injections were performed independently. For correlation of SR101 intensities with PDGC connections (Fig. 1c, d) SR101 (#S359, Invitrogen, part of Thermo Fisher Scientific, Waltham, Massachusetts, USA) was dissolved in sterile saline solution (#2350748, B. Braun Melsungen AG, Melsungen, Germany) and intravenous injected to reach 0.12 mg /g body weight. Alternatively, no SR101 was administered for morphological assessment of PDGCL tumors (Fig. 1b). Repetitive intravital two-photon microscopy was performed 4–10 h after SR101 injection during which the body temperature of the mice was kept constant using a heat pad with a fixed temperature of 37 °C. A Zeiss 7MP microscope (Zeiss, Oberkochen, Germany), equipped with a Ti:Sapphire Chameleon UltraII laser (Coherent, Santa Clara, California, USA) and Zeiss Zen 2012 black edition software (Zeiss, Oberkochen, Germany) for excitation of SR101 (900 nm) and tGFP (950 nm), and band-pass 500–550 and 575–610 nm filters was used for imaging. A 20x (NA 1.0) water immersion objective (Zeiss, Oberkochen, Germany) was used.

Astrocytes and PDGCs were analyzed in $n = 5$ regions in $n = 3$ animals on D64+/−9 days, whereby tGFP^low, SR101^high cells were defined as astrocytes[13] and tGFP^high cells as PDGCs. Mean SR101 signal intensities of individual PDGCs were measured in the cell bodies. To compensate for different signal intensities in different tumor areas and depths the SR101 signal intensity of each PDGC was normalized by the mean value of the highest 10% of intensities of all PDGCs in the respective region. Connected cells were defined by a direct cell-cell connection via a TM. The number of astrocytes and PDGCs in different tumor areas was counted manually after segmentation with Aivia (Leica, Wetzlar, Germany).

## Separation of SR101^high and SR101^low PDGCL xenograft groups

Animal work. PDGCL spheroids were dissociated into a single cell suspension using Stem-Pro Accutase™ (#1110501, Thermo Fisher Scientific, Waltham, Massachusetts, USA). $5 \times 10^4$ viable PDGCs were slowly injected into the right striatum of 8–10 week old male Crl:NMRI-*Foxn1^nu* nude mice (RRID:MGI:5653040; Charles River, Wilmington, Massachusetts, USA) using a 10 μl micro-syringe (#80308, Hamilton, Reno, Nevada, USA) driven by a stereotactic device (Stoelting, Wood Dale, Illinois, USA). $n = 3$ mice were used per PDGCL and injections were performed independently. Mice were intraperitoneally injected with SR101 (#S359, Invitrogen, S359, Invitrogen, part of Thermo Fisher Scientific, Waltham, Massachusetts, USA) in sterilized saline solution (#2350748, B. Braun Melsungen AG, Melsungen, Germany) at a dose of 0.12 mg per g body weight. After an incubation period of 8 h to ensure maximum SR101 uptake from PDGCs, mice were deeply anesthetized with ketamine/Ketaset® (#794-523, Zoetis, Berlin, Germany) and xylazine/Rompun® (#770-081, Bayer, Leverkusen, Germany) and transcardially perfused with sterilized phosphate buffer saline (PBS, #D8537, Sigma, part of Merck, Darmstadt, Germany). Single cell suspensions were generated from the whole brains utilizing a combination of gentleMACS™ Dissociator (#130-093-235, Miltenyi Biotec, Bergisch Gladbach, Germany) and brain tumor dissociation kit (#130-095-942, Miltenyi Biotec, Bergisch Gladbach, Germany) according to the manufacturer´s recommendations. The obtained suspension was passed through 100 μm (#542000, Greiner Bio-one, Kremsmünster, Austria) and 70 μm (542070, Greiner Bio-one, Kremsmünster, Austria) strainer meshes. After subsequent centrifugation at 500 g for 5 min, the cell pellet was resuspended in FACS buffer, consisting of 1% fetal calf serum (FCS; #S0615, Sigma, part of Merck, Darmstadt, Germany) in PBS.

FACS. The single cell suspension was incubated with eBioscience™ Calcein Violet 450 AM (#65-0854-39, Invitrogen, part of Thermo Fisher Scientific, Waltham, Massachusetts, USA) and TO-PRO™-3 Iodide (#T3605, Invitrogen, part of Thermo Fisher Scientific, Waltham, Massachusetts, USA) for 10 min on ice prior to sorting. Standard gating techniques were used to discriminate doublets and dead cells. The viable fraction was defined by TO-PRO™-3 Iodide negativity and Calcein Violet 450 AM positivity. To further allow discrimination of the non-malignant cells, the tGFP^high population was selected for separation of highly connected tumor cells (SR101^high) and lowly connected tumor cells (SR101^low) using the FACSAria™ cell sorter (BD Biosystems, Franklin Lakes, New Jersey, USA) and FACSDiva® v.8.0.2 software (RRID:SCR_001456, BD Biosystems, Franklin Lakes, New Jersey, USA). The following excitations and filters were used: V405-450/50 (Calcein Violet), B488-530/30 (tGFP), YG561-586/15 (SR101) and R640-650/17 (TO-PRO™-3).

## RNA-Seq data generation and preprocessing from SR101 xenograft models

Sorted PDGCs from at least 3 mice per replicate were resuspended in lysis buffer included as a part of the RNeasy® Micro Kit (#74004, Qiagen, Hilden, Germany). mRNA was then isolated and purified in accordance with the manufacturer's instructions. The conversion of RNA to

DNA was done with the SMARTer® Ultra® Low Input RNA for Illumina Sequencing (#634940, TakaraBio, Kusatsu, Japan). The libraries were then prepared using NEBNext® ChIP-Seq Library Prep Master Mix Set for Illumina (#E6240, New England Biolabs, Ipswich, Massachusetts, USA) and sequenced on an Illumina HiSeq 2000 sequencer (RRID:SCR_020132, v.4, Illumina, San Diego, California, USA) in 50 bp single-end mode by the Genomics and Proteomics Core facility, DKFZ. The bioinformatics tools for gene expression quantification from RNA-Seq were used with default parameters: The quality of bases was evaluated and controlled using FASTX-Toolkit (RRID:SCR_005534). HOMER (RRID:SCR_010881, v.4.7) was applied for PolyA-tail trimming; reads with a length of < 17 bp were removed. The filtered reads were mapped with STAR (RRID:SCR_004463, v.2.3) against the human reference genome (GRCh38) and Picard (RRID:SCR_006525, v.1.78) with CollectRNASeqMetrics were used for quality checking. Count data was generated by htseq-count (RRID:SCR_011867, v.0.9.1) using the GENCODE (RRID:SCR_014966, v.26) for annotation. Genes with less than 10 a total counts in all samples were discarded.

### scRNA-Seq data generation from SR101 xenograft models
A total of $5 \times 10^4$ tGFP^high, SR101^high and tGFP^high, SR101^low PDGCs from at least 3 mice/replicates per PDGCL were sorted into 10% bovine serum albumin (BSA; #8076.4, Roth, Karlsruhe, Germany) coated tubes containing 500 ul PBS (#D8537, Sigma, part of Merck, Darmstadt, Germany) supplemented with 10% PDGCL media, 3 mM ethylenediaminetetraacetic acid (EDTA, #AM9260G, Invitrogen, part of Thermo Fisher Scientific, Waltham, Massachusetts, USA) and 0.1% BSA. Cells were centrifuged at 400 g for 5 min and the volume adjusted to allow loading of the maximum number of cells for scRNA-Seq. scRNA-Seq was done using the Chromium Next GEM Single Cell 3′ GEM, Library & Gel Bead Kit v2 (PN-120237, 10x Genomic, Pleasonton, California, USA) according to the manufacturer´s instructions and sequencing carried out on a HiSeq 4000 sequencer (SY-401-4001, Illumina, San Diego, California, USA) or on a NovaSeq 6000 sequencer (20012850, Illumina, San Diego, California, USA) to obtain approximately $2 \times 350$ million reads per sample.

### Single cell data preprocessing and quality control
Gene expression count matrices of scRNA-Seq data from SR101 xenograft models were generated using Cell Ranger (RRID:SCR_017344, v.2.1.1, 10X Genomics, Pleasanton, California, USA) with default parameters, against the pre-built hg19 human reference genome (Cell Ranger reference, v.1.2.0). We discarded PDGCs by uniform exclusion criteria: (1) cells which had fewer than 200 or more than 8000 genes detected. (2) cells which had fewer than 500 or more than 80,000 counts detected. (3) cells which had more than 10% of counts that came from mitochondrial genes.

After the uniform exclusion, sample-wise outlier cells were detected and removed if the number of genes or counts were more than three median absolute deviations (MADs) above the median using the isOutliers function in the scater package (RRID:SCR_015954, v.1.10.1). In each sample, per-cell doublet scores and per-sample doublet score thresholds were estimated by Scrublet (RRID:SCR_018098, v.0.2.1) with default parameters. If one doublet score threshold was located between two peaks of a doublet score histogram, this threshold was accepted and the cells with a doublet score higher than this threshold were removed. Shared nearest neighbor unsupervised clusters were identified and visualized in UMAP with Seurat (RRID:SCR_007322, v.3.1.5) to further detect clusters located exclusively far away from the majority of clusters. In the end, we obtained 35,822 cells from six samples of three PDGCL xenografted mouse models.

### Single cell data processing and integration
Data processing. After data preprocessing and quality control, scRNA-Seq data of SR101 xenograft models were further processed using Seurat (RRID:SCR_007322, v.3.1.5) with default parameters: The gene expression counts were normalized using the NormalizeData function. Then 2000 highly variable genes were identified using the FindVariableFeatures function. The variation of number of counts among cells was regressed out, and the resulting residuals were scaled and centered by the ScaleData function. Next, we reduced dimensionality of the data by principal component analysis using the RunPCA function. The number of principal components (PCs) used for further analyzes was determined using the ElbowPlot function. The data was visualized in UMAP using RunUMAP function with determined PCs.

Data integration. To remove the differences of individuals and perform batch correction, an integration method based on identification of shared anchors between pairs of samples was applied using Seurat (RRID:SCR_007322, v.3.1.5) with default parameters: The gene expression count of each PDGCL was normalized and highly variable genes were selected using the NormalizeData and FindVariableFeatures functions. Then the normalized data was integrated with the FindIntegrationAnchors function (dims = 1:30) and the IntegrateData function (dims = 1:30). The integrated data was then further processed using the ScaleData, RunPCA, ElbowPlot, RunUMAP functions as described before.

### Development of the connectivity signatures
The computational development of connectivity signatures is illustrated in Fig. 1e. In scRNA-Seq data of SR101 xenograft models, DEGs between SR101^high and SR101^low groups were identified in each PDGCL xenografted model using the FindMarkers function with default parameters in Seurat (RRID:SCR_007322, v.3.1.5). We then aggregated the significant DEGs (adjusted $p$ value < 0.05) from all three PDGCLs. Among the aggregated DEGs, the DEGs with the same direction of regulation and large fold change (absolute log fold-change ≥ 0.4) in two PDGCLs, or DEGs with the same direction of regulation in all three PDGCLs were kept. In total, 71 DEGs were derived from the scRNA-Seq dataset and served as the connectivity signature.

In RNA-Seq of SR101 xenograft models, DEGs between SR101^high and SR101^low groups were identified using DESeq2 (RRID:SCR_015687, v.1.22.2): To obtain consistent DEGs across two PDGCL xenografted models, ~ PDGCL + Group was included in the design formula of the DESeqDataSet function. Differential expression analysis was performed using the DESeq function. Then the log fold changes were shrunken using the apeglm method in the lfcShrink function. Other parameters were by default. The significant DEGs (adjusted $p$ value < 0.05) with an absolute log2 fold-change ≥1 were kept. Finally, 245 DEGs were derived from the RNA-Seq dataset.

### Heatmap visualization of the connectivity signatures
For each connectivity gene derived from scRNA-Seq, the gene expression level of the gene in cells of each sample were averaged using the AverageExpression function in Seurat (RRID:SCR_007322, v.3.1.5). The average expression levels were Z-score scaled, centered, winsorized at −3 and 3, and then visualized as heatmap using ComplexHeatmap (RRID:SCR_017270, v.2.5.4).

The bulk count matrix was transformed with variance stabilizing transformation using the vst function in DESeq2 (RRID:SCR_015687, v.1.22.2), and the batch effects between the PDGCL xenografted models were corrected with the removeBatchEffect function of the LIMMA package (RRID:SCR_010943, v.3.36.5). Finally, the expression levels of connectivity genes derived from RNA-Seq were Z-score scaled, centered, winsorized at −3 and 3, and then visualized as heatmap using ComplexHeatmap (RRID:SCR_017270, v.2.5.4).

### GO enrichment analysis
GO enrichment analysis of the 71-gene connectivity signature derived from scRNA-Seq or 245-gene connectivity signature derived from RNA-

Seq was performed by the compareCluster function using cluster-Profiler (RRID:SCR_016884, v.3.18.1) against GO Biological Process, which was obtained with the settings fun = enrichGO and ont = BP. The most enriched GOs were visualized with the emapplot function using enrichplot (v.1.10.2).

There are 16,759 genes commonly expressed in both scRNA-seq and RNA-Seq datasets of SR101 xenograft models. GSEA of these genes preranked by the fold change between highly and lowly connected groups in the scRNA-Seq dataset or RNA-Seq dataset was calculated by gene set enrichment analysis (RRID:SCR_003199, v.4.1.0, Broad Institute, Boston, Massachusetts, USA) against the neurogenesis gene set.

GO enrichment analysis of 123 DEPs, 152 DPPs or 184 calcium signature genes was performed by ShinyGO (RRID:SCR_019213, v.0.741) against the GO Biological Process gene set of ShinyGO.

## Connectivity signature score

The connectivity signature derived from scRNA-Seq data contains 71 genes, among which, 40 genes are upregulated in highly connected cells and 31 genes are downregulated. The 40 upregulated genes were used as a gene set to calculate a score (connectivity-upregulated signature score) in each cell using the AddModuleScore function in Seurat (RRID:SCR_007322, v.3.1.5). The score represents the relative expression of a gene set. Similarly, a second score (connectivity-downregulated signature score) based on the 31 downregulated genes was calculated. Finally, the connectivity signature score was defined as the connectivity-upregulated signature score minus the connectivity-downregulated signature score. Another connectivity signature score based on 245 genes (57 upregulated genes and 188 downregulated genes) derived from the RNA-Seq data was generated accordingly.

## The performance of the connectivity signatures for prediction of SR101-sorted labels

In each cell of the scRNA-Seq data from SR101 xenograft models, the connectivity-upregulated signature score based on 40 scRNA-Seq-derived upregulated connectivity genes and the connectivity-downregulated signature score based on 31 scRNA-Seq-derived downregulated connectivity genes were calculated. If the connectivity-upregulated signature score was higher than the connectivity-downregulated signature score, the cell was predicted as highly connected, otherwise, the cell was predicted as lowly connected. Confusion matrix and prediction metrics (i.e accuracy, sensitivity, specificity, positive predictive value and negative predictive value) were obtained between the number of cells predicted as highly connected or lowly connected based on the calculated scores and the number of cells labeled as highly connected or lowly connected after SR101-based cell sorting, using the R package caret (RRID:SCR_021138, v.6.0-80). Another prediction based on 57 RNA-Seq-derived upregulated connectivity genes and 188 RNA-Seq-derived downregulated connectivity genes was calculated in the same way.

Negative control. 100 random gene sets, each gene set including 71 randomly selected genes (40 gene as an upregulated gene set and 31 as a downregulated gene set, same size as the scRNA-Seq-derived connectivity signature), were utilized to calculate scores and obtained the average prediction metrics. Another 100 random gene sets, each gene set including 245 randomly selected genes (57 gene as an upregulated gene set and 188 as a downregulated gene set, the same as RNA-Seq-derived connectivity signature), were utilized to calculate scores and obtained the average prediction metrics.

## Malignant cell state assignment

Cell state-defining markers from a GB scRNA-Seq study[3] were utilized to calculate cell state signature scores in each malignant cell in our SR101 xenograft scRNA-Seq dataset using the AddModuleScore function in Seurat (RRID:SCR_007322, v.3.1.5). Malignant cells were assigned to this cell state that gained the highest signature score among all cell state signature scores.

## RNA velocity in scRNA-Seq data of PDGCL xenografted mouse models

The pre-mature and mature mRNA count matrices were obtained by velocyto (RRID:SCR_018167, v.0.17.15) with default setting for 10X scRNA-Seq. Then the count matrices were processed by scVelo (RRID:SCR_018168, v.0.2.4): the data was filtered, normalized, reduced in PCA space, and nearest neighbors obtained using default parameters. Then the RNA velocities were estimated and projected in PCA embedding as streamlines. Directed PAGA graphs were calculated based on cell state transition possibilities inferred from velocities. The PAGA graphs were visualized by Cytoscape (RRID:SCR_003032, v.3.9.0).

## In vitro Ca²⁺ imaging assay

S24 PDGCs were cultured in vitro on Matrigel® in high-glucose medium (HGM) under serum-free stem-like conditions. These conditions preserve both gene expression and biological properties, such as the diffuse growth and network formation of the original tumor[51], and are referred to as Matrigel® monolayer assay in the following.

Experimentally, HGM was produced by supplementing DMEM-F12 medium (#11330-032, Invitrogen) with glucose to 50 mM final concentration (G7021-1KG, Sigma, part of Merck, Darmstadt, Germany), B27 supplement (#12587-010, Invitrogen, part of Thermo Fisher Scientific, Waltham, Massachusetts, USA), 5 µg/ml insulin (#I9278, Sigma, part of Merck, Darmstadt, Germany) and 5 µg/ml heparin (#H4784, Sigma, part of Merck, Darmstadt, Germany). After dilution, HGM was sterile filtered. A 96-well plate (#655090, Greiner Bio-One, Kremsmünster, Austria) was coated with growth-factor reduced Matrigel® (#G356231, Corning Inc., Corning, New York, USA) diluted 1:50 in DMEM-F12 medium and allowed to solidify at 37 °C for 1 h. S24 PDGCL was singularized using Accutase (#1110501, Thermo Fisher Scientific, Waltham, Massachusetts, USA), washed with PBS (#D8537, Sigma, part of Merck, Darmstadt, Germany), resuspended in HGM and 30,000 cells seeded per well.

After 7 days incubation at 37 °C, 5% $CO_2$ the Ca²⁺-sensitive fluorescent Rhod-2AM (#R1244, Thermo Fisher Scientific, Waltham, Massachusetts, USA) was added to 1 µM final concentration and incubation at 37 °C for 30 min allowed. Ca²⁺-Imaging was performed in medium at 37 °C and 5% $CO_2$ in a heat-controlled chamber using a Zeiss LSM 710 ConfoCor 3 confocal microscope with Zeiss Zen 2012 black edition v.8.1.0.484 software (Zeiss, Oberkochen, Germany) at 561 nm ex. with a 20x (NA 0.8) dry objective. For time series a scanning speed of 1.52 seconds per frame was used. Every time series was recorded over 30 min. At least 3 recordings of S24 PDGCs were conducted.

## Analysis of Ca²⁺ communication

Ca²⁺ activity and cross-correlation analysis was performed on $n = 1357$ cells from 3 recordings à 30 min using a customized pipeline. Using Fiji 2.0.0[52] (RRID:SCR_002285), single-cell mean-intensity traces over time were acquired. In MATLAB (#R2020b, RRID:SCR_001622., MathWorks Inc., Natick, Massachusetts, USA) single-cell traces were smoothed using the gaussian filter (sigma = 10 seconds) and peaks were detected using the peak finder function.

Cross-correlation analysis was performed using the following functions implemented in MATLAB[53,54]:

To determine the coactivity between active cells (≥4 Ca²⁺ peaks), pairs of single-cell traces a and b were shifted forwards and backwards in time relative to one another to adjust for the time needed by the Ca²⁺ transient traveling from one cell to the other. For every pair of traces the best correlation over a total segment length of 10 min was found using the Pearson's correlation rho(a, b) and defined as the coactivity

of the two cells:

$$\text{rho}\,(a,b) = \max_\tau \left( \frac{\sum_{i=1}^{n}(a_{i-\tau} - \bar{a})\,(b_i - \bar{b})}{\left\{ \sum_{i=1}^{n}(a_{i-\tau} - \bar{a})^2 \sum_{j=1}^{n}\left(b_j - \bar{b}\right)^2 \right\}^{1/2}} \right) \quad (1)$$

where n is the length of each single-cell trace measured in number of frames.

Following data was excluded from analysis as a predefined criterion to avoid random correlations: Cells with less than four peaks, cell pairs that were more than 100 μm apart and correlations where the signal would have traveled <4 μm/s or >25 μm/s. The distance cut-off was chosen as the maximum value at which the detected functional connections optimally corresponded to the visually observed $Ca^{2+}$ transients traveling between two cells. Higher distance cut-off values resulted in decreasing concordance as increasingly more functional connections were found randomly. The range in speed was determined by manually measuring the speed of several $Ca^{2+}$ signals traveling between two cells and was defined by calculating two standard deviations below and above the mean for all groups and choosing the lowest and highest values, respectively. The distance cut-off and the range in speed limited the range of possible time shifts of the respective trace to a maximum of 25 s (= 100 μm / 4 μm/s).

A pair of cells was defined as coactive if the maximum correlation coefficient ($\text{rho}(a, b)_{max}$) was above 0.49 for S24 in vitro corresponding to the 95th percentile of the null control[54]. As the null control, a dataset was generated using the linear shift method[55]. Here, the same calculations as described above were performed, but when calculating the best correlation between pairs of single-cell traces a and b over a total segment length of 10 min, the paired 10-min-long segment was randomly chosen from trace b with a linear time shift $\Delta T > 5$ min to ensure that the segments of trace a and b cannot contain biologically meaningful correlations (For the empirical data, the paired 10-min-long segment would have been chosen from trace b with $\Delta T = 0$). All subsequent calculations were then performed identically to the empirical data. Thereby, the same traces and the same number of segments were correlated but without the possibility of detecting true $Ca^{2+}$ communication, while preserving the temporal autocorrelation of each trace.

## Quantification of anatomical connections in in vitro $Ca^{2+}$ imaging data
Anatomical cell-cell-connections were determined manually using Fiji 2.0.0.

## Recording of cellular calcium epochs using Caprola
The concept behind the Caprola constructs and experimental procedure was previously described[18]. Briefly, Caprola$_6$ consists of the self-labeling HaloTag7 protein whose labeling reaction was rendered strictly dependent to the presence of calcium. An additional control construct, CaProLa$_{on}$, consists of a constitutively active Caprola whose labeling reaction is calcium-independent. We harnessed this feature of Caprola$_{on}$ to eliminate the dye permeability contribution in the conducted experiment, therefore ensuring the calcium-specificity of the observations.

Generation of stable cell lines. Plasmids encoding for Caprola$_6$ and Caprola$_{on}$ were provided by M.C.H and J.H. Caprola encoding lentiviruses were produced as described in the section Cell culture of PDGCLs. Clones co-overexpressing EGFP and Caprola were enriched by exposure to Puromycin (#A2856.0100, Applichem, Darmstadt, Germany) for 72 h and, after 7 days, by subsequent separation based on their EGFP intensity using FACS (FACSAria™, BD Biosystems, Franklin Lakes, New Jersey, USA).

RNA-Seq. Technically, Caprola$_6$ or Caprola$_{on}$ expressing PDGCs were cultured under Matrigel® monolayer assay conditions in T75 flasks. After 48 h, chemical labeling was performed by the addition of CPY-CA dye to reach 125 nM final concentration. After an incubation at 37 °C, unbound excess dye was quenched through the exposure to 1 uM recombinant HaloTag protein for 5 min. PDGCs were subsequently rinsed with ice-cold media and subsequently PBS (#D8537, Sigma, part of Merck, Darmstadt, Germany), singularized using Accutase (#A1110501, Thermo Fisher Scientific, Waltham, Massachusetts, USA) and recovered in PBS. Next, PDGCs were subjected to FACS sorting in order to separate groups based on their labeling intensity. Briefly, the single cell population was defined using standard gating techniques and the viable fraction was characterized by high EGFP signals. This population was further selected for separation of high, medium and low CPY-CA/EGFP (aka labeling intensity) ratios, whereas the EGFP signal was similar between these groups, using the FACSAria™ cell sorter (BD Biosystems, Franklin Lakes, New Jersey, USA) and FACS-Diva® v.8.0.2 software (RRID:SCR_001456, BD Biosystems, Franklin Lakes, New Jersey, USA). The following excitations and filters were used: B488-530/30 (EGFP) and RL640-670/14 (CPY-CA). RNA from the collected PDGCs was isolated with the Arcturus Pico Pure Kit (#Kit0204, Thermo Fisher Scientific, Waltham, Massachusetts, USA). Library preparation and RNA sequencing on a NovaSeq6000 device (Illumina) was carried out by the GPCF at the DKFZ.

## BTP2 treatment of S24-Caprola cells
S24-Caprola$_6$ PDGCs were seeded, cultured and analyzed by FACS as described before, but in 96-well plate format. 10 μM BTP2 (#Y4895, Sigma, part of Merck, Darmstadt, Germany) or DMSO as a control were added 24 h after cell seeding and cells cultured for another 48 h under 37 °C, 5% $CO_2$ and atmospheric pressure. Flow cytometry analysis was performed with FlowJo™ v.10.8.1 (RRID:SCR_008520, BD Biosystems, Franklin Lakes, New Jersey, USA).

## BTP2 treatment of S24 PDGCL cells
Cell culture. S24-tGFP PDGCs were cultured under Matrigel® monolayer assay conditions either in 96-well plates (#655090, Greiner Bio One, Kremsmünster, Austria) for image acquisition or 6-well plates (#92006, TPP, Trasadingen, Switzerland) for RNA isolation. 10 μM DMSO as a control or BTP2 (#Y4895, Sigma, part of Merck, Darmstadt, Germany) were added 24 h after the cell seeding and cells cultured for another 48 h under 37 °C, 5% $CO_2$ and atmospheric pressure.

Cells were sequentially washed with each HGM and PBS once and incubated for 1 h with Hoechst33342 (#H3570, Invitrogen, part of Thermo Fisher Scientific, Waltham, Massachusetts, USA) and EthD-2 (#E3599, Thermo Fisher Scientific, Waltham, Massachusetts, USA) in HGM.

Image acquisition and analysis. Images were acquired on a LSM710 confocal microscope using 405 nm (Hoechst33342) and 514 nm excitation (EthD-2) with a 20× (NA 0.8) dry objective. The percentage of dead cells was assessed based on nuclei (Hoechst33342) and dead cell (EthD2) counts using Fiji 2.0.0. TMs per cell were manually counted in Fiji 2.0.0.

RNA-Seq. RNA-Seq was carried out similar to as described for the wt PDGCLs.

## RNA-Seq data processing of wt and BTP2 treated PDGCL as well as Caprola$_6$, CaProLa$_{on}$ PDGCL datasets
The sequencing quality of samples was assessed by a standard quality control. One sample (Caprola$_6$, low labeling intensity group, replicate 2) was excluded from further analysis due to a low quality as indicated by a low number of reads. Sequencing reads were aligned using STAR (RRID:SCR_004463, v.2.5.3a) against the human reference genome GRCh38, and gene counts were generated and annotated using GENCODE (RRID:SCR_014966, v.32) by featureCounts function of Subread package (RRID:SCR_009803, v.1.5.3). Gene counts were normalized to fragments per kilobase million (FPKM)

values and log2 transformed. Then, the connectivity signature scores were calculated.

Caprola$_6$ and Caprola$_{on}$ signatures were generated with the same methods illustrated in Supplementary Fig. 5c. The genes expressed in only one sample were excluded. DEGs were identified using edgeR (RRID:SCR_012802, v.3.34.1). Data was normalized using calcNormFactors function and estimated dispersions using estimateDisp function. Then data was fitted to negative binomial GLM using glmQLFit function. DEGs in pairwise groups (three comparisons, i.e., High vs. Medium, Medium vs. Low and High vs. Low) were identified by quasi-likelihood (QL) F-test using glmQLFTest function. The DEGs of three comparisons were merged and the following filters applied: (1) DEGs with adjusted $p$ value (FDR) > 0.05 were removed; (2) DEGs only detected in one comparison were removed; (3) DEGs with no consistent direction of regulation among comparisons were removed; (4) DEGs with low expression that log count-per-million (CPM) < 2 were removed; (5) ordering the remaining DEGs according to fold change, the DEGs not located in the top 50 or bottom 50 most regulated genes in each comparison were removed. At last, 184-gene Caprola$_6$ signature and 57-gene Caprola$_{on}$ signature were obtained.

### scRNA-Seq from a stem-like culture in vitro model of connectivity

Cell seeding and culture. S24-tGFP and T269-tGFP PDGCs were cultured adherently under stem-like conditions in neural stem cell media. Solely the seeding density dictated if the cells formed either a highly connected network or remained mainly unconnected.

In brief, T25 cell culture flasks (#690175, Greiner Bio-one, Kremsmünster, Austria) were coated at 4 °C overnight with 10 ug/ml Poly-ornithin (#P3655, Sigma, part of Merck, Darmstadt, Germany) in ddH$_2$O and washed twice with ice-cold PBS (#D8537, Sigma, part of Merck, Darmstadt, Germany). Subsequently, laminin coating using a 10 ug/ml solution in PBS (#L2020 1 mg, Sigma, part of Merck, Darmstadt, Germany) was performed for 3 h. S24 and T269 PDGCs were singularized with Accutase (#A1110501, Thermo Fisher Scientific, Waltham, Massachusetts, USA), washed twice with PBS and resuspended in neural stem cell media consisting of a 1:1 mixture of Neurobasal-A media (#21103-049, Thermo Fisher Scientific, Waltham, Massachusetts, USA) and DMEM/F-12, GlutaMAX (#31331-093, Thermo Fisher Scientific, Waltham, Massachusetts, USA), supplemented with B27 w/o vit. A (#12587-010, Thermo Fisher Scientific, Waltham, Massachusetts, USA), N2 supplement (#17502-048, Thermo Fisher Scientific, Waltham, Massachusetts, USA), 20 ng/ml EGF (#PHG0311, Life Technologies, part of Thermo Fisher Scientific, Waltham, Massachusetts, USA) and 20 ng/ml FGF (FGF; #PHG0021, Life Technologies, part of Thermo Fisher Scientific, Waltham, Massachusetts, USA). 36,000 and 862,000 S24 PDGCs or 50,000 and 978,000 T269 PDGCs, respectively, were planted per flask and growth allowed for 3 days under 37 °C, 5% CO$_2$ standard culture conditions.

Image acquisition and quantification of TMs per cell. After 3 days, phase contrast images were acquired using a 20x (NA, 0.4) dry objective (#MRP46202, Nikon, Miyato, Japan) on an Eclipse Ts2-FL (Nikon, Miyato, Japan) microscope equipped with a 1920×1200 pixel monochromatic camera (#IS-DMK33UX174, Nikon, Miyato, Japan). Fiji 2.0.0 was used for manual quantification of the TM-number per cell. A total of $n = 50$ cells from $n = 10$ ROIs from $n = 2$ independent experiments were quantified per condition and PDGCL.

Single cell sequencing. Cells were detached using Accutase, washed twice with PBS and labeled with cholesterol modified oligos (CMO,[56]). Briefly, 500,000 PDGCs per line and condition were resuspended in PBS and incubated with CMO solution (Integrated DNA technologies, Coralville, Iowa, USA). After 3 wash cycles using 0.1% BSA (#0163.4, Roth, Karlsruhe, Germany) in PBS and 300 g, 3 min, 4 °C centrifugation, DAPI was added to 100 ng/ul final concentration. Using standard gating strategies to discriminate multiplets and dead cells,

around 8000 live cells per condition were sorted on a FACSAria™ Fusion Special Order System (BD Biosystems, Franklin Lakes, New Jersey, USA) with FACSDiva® v.8.0.2 software (RRID:SCR_001456, BD Biosystems, Franklin Lakes, New Jersey, USA) using the following excitation and filter settings: V405-450/50 (DAPI) and B488-530/30 (tGFP). PDGCs were collected in a 10% BSA coated 1.5 ml reaction tube and further processed according to the manufacturer´s instructions with the Chromium Next GEM Single Cell 3′ GEM, Library & Gel Bead Kit v3.1 (#PN-1000121, 10x Genomics, Pleasanton, California, USA). Sequencing was carried out on a NovaSeq 6000 (#20012850, Illumina, San Diego, California, USA).

Bioinformatic preprocessing. The procedure of the bioinformatic preprocessing was similar to the one for SR101 xenograft dataset with the following exceptions: Count matrices of stem-like in vitro model of connectivity scRNA-Seq were generated using "cellranger multi" function in Cell Ranger software with default parameters, against the pre-built hg19 human reference genome (Cell Ranger reference, v.1.2.0, 10x Genomics, Pleasanton, California, USA).

In the end, we obtained 5388 PDGCs from two stem-like in vitro models of connectivity.

### Assessing TM parameters from a serum-based in vitro model of connectivity

Quantification of TMs. Cytosolic tGFP-overexpressing PDGCLs BG5-tGFP, S24-tGFP, and T269-tGFP were cultured under two different culture conditions. For neurosphere conditions, referred to as TM-, cells were cultured in PDGCL media as described in section Cell culture of PDGCLs. In order to induce the formation of TMs, referred to as TM + , cells were kept in DMEM (#11965-118, Life Technologies, part of Thermo Fisher Scientific, Waltham, Massachusetts, USA) supplemented with 10% FCS (#S12595H, R&D Systems, Minneapolis, Minnesota, USA) and 6-well plates.

PDGCs were seeded in a density of $2 \times 10^5$ cells per well. Tissue culture-treated 6-well plates were used for the TM+ condition (#353224, Corning Inc., Corning, New York, USA) in contrast to the 6-well plates used for TM- conditions (#83.3920.500, Sarstedt, Nürnbrecht, Germany). On day 20 cultures were prepared for imaging.

Preparation, image acquisition and Ilastik-based TM-length quantification. Wells were rinsed once with PBS to get rid of floating dead cells. Media containing 1.25 μM Ethidium-homodimer 2 (EthD2, #E3599, Invitrogen, part of Thermo Fisher Scientific, Waltham, Massachusetts, USA) and 1 μg/ml Hoechst33342 (#H3570, Invitrogen, part of Thermo Fisher Scientific, Waltham, Massachusetts, USA) were added to allow dead and total cell quantification. Dyes were allowed to bind for 30 min at 37 °C, 5% CO$_2$ before imaging. Images were acquired either on a LSM710 or LSM780 confocal microscope (Zeiss, Oberkochen, Germany) and a EC plan Neofluar® 10x (NA 0.3) dry DICI or 20x (NA 0.8) dry objective (Zeiss, Oberkochen, Germany) and Zeiss Zen 2012 black edition v.8.1.0.484 software (Zeiss, Oberkochen, Germany). The following excitation wavelengths were used: 405 (Hoechst33342), 488 (tGFP) and 561 (EthD2). $Z$ intervals of 5 μm and gains between 620 and 750 were used. Laser power and maximum imaging time were tuned as low as possible to avoid phototoxicity. Images with a pixel size of 0.89 μm and an imaging frequency of 0.3 Hz were used for quantification.

Image processing and Ilastik-based quantification. Images were transferred to Fiji 2.0.0 for channel splitting and generating of orthogonal projections. Machine-learning-based semi-automatic image analysis of TM lengths and nuclei numbers was performed with the open-source software Ilastik[57,58] (RRID:SCR_015246) after appropriate training, using a previously validated pipeline[59]. TMs/cell were calculated based on the sum of all objects lengths per ROI divided by the number of cells per ROI. Only ROIs with a comparable number of cells were selected for analysis. Mann-Whitney U test was used for comparison of groups. Figures show maximum intensity projections with

Hoechst and tGFP signal being oversaturated to allow better visibility of nuclei and TMs.

### scRNA-Seq from a serum-based in vitro model of connectivity

**Cell lines and cell culture.** PDGCLs BG5, S24, and T269 were cultured under TM- and TM+ conditions as described in section Assessing TM parameters from a serum-based in vitro model of connectivity, with the only exception of using a comparable North America approved FCS from the same manufacturer (#S1155OH, R&DSystems, Minneapolis, Minnesota, USA) and T25 flasks instead of 6-well plates.

**FACS.** After cultivation under both culture conditions, cells were blocked with 1% BSA in PBS. PDGCs were washed with PBS and subsequently resuspended in 1.5 ml of PBS/1%BSA containing 1 μM calcein AM (#C1430, Life Technologies, part of Thermo Fisher Scientific, Waltham, Massachusetts, USA) and 0.33 μM TO-PRO™-3 (#T3605, Invitrogen, part of Thermo Fisher Scientific, Waltham, Massachusetts, USA) to co-stain before sorting. Sorting was performed with FACSAria™ Fusion Special Order System (BD Biosystems, Franklin Lakes, New Jersey, USA) with FACSDiva® v.8.0.2 software (RRID:SCR_001456, BD Biosystems, Franklin Lakes, New Jersey, USA) using B488-530/30 nm (Calcein AM) and R640-670/14 nm (TO-PRO-3™) excitations and filters. An unstained control was included with every sample. Standard, strict forward scatter height versus area criteria were applied to discriminate doublets. Viable cells were detected as staining positive for calcein AM and negative for TO-PRO™-3.

**scRNA-Seq.** PDGCs were sorted into 96 well plates (#0030128.648, Eppendorf, Hamburg, Germany) containing cold TCL Buffer (#1070498, Qiagen, part of Thermo Fisher Scientific, Waltham, Massachusetts, USA) including 1% beta-mercaptoethanol (#M7522, Sigma, part of Merck, Darmstadt, Germany), snap frozen on dry ice and stored at −80 °C. Whole transcriptome amplification, library preparation and sequencing were performed according to the SmartSeq2 protocol[60] with the following modifications as previously published[3]: RNA purification from single cells was performed with Agencourt RNAClean XP beads (#A63987, Beckmann Coulter, Brea, California, USA) prior to olio-dT primed reverse transcription with Maxima reverse transcriptase (#EP0753, Life Technologies, part of Thermo Fisher Scientific, Waltham, Massachusetts, USA) and locked template switch oligonucleotide (#339413, Qiagen, part of Thermo Fisher Scientific, Waltham, Massachusetts, USA). This was followed by 20 cycles of polymerase chain reaction (PCR) amplification using KAPA HiFi HotStart ReadyMix (#KK2602, Roche, Basel, Switzerland) and subsequent purification with Agencourt AMPure XP beads. Library construction was performed using the Nextera XT Library Prep kit (#FC-131-1024, Illumina, San Diego, California, USA) and custom barcode adapters (sequences available upon request). Libraries from 864 cells with unique barcodes were combined and sequenced with a NextSeq 500 sequencer (#SY-415-1001, Illumina, San Diego, California, USA).

### scRNA-Seq data processing of a serum-based PDGCL in vitro model of connectivity

Sequencing reads were aligned using STAR (RRID:SCR_004463, v.2.5.3a) against the human reference genome hg19, and gene counts were generated and annotated using GENCODE (RRID:SCR_014966, v.19) by featureCounts function of Subread package (RRID:SCR_009803, v.1.5.3). Gene counts were normalized to FPKM values and log2 transformed. We identified low quality cells by the number of expressed genes lower than 2000 or higher than 8000. We obtained 566 cells from three PDCGLs.

### Single nuclei (sn)RNA-Seq data generation from patient samples

Frozen resected tumor material was retrieved from the Department of Neuropathology in Heidelberg and reviewed by a board-certified neuropathologist. Diagnoses were molecularly confirmed according to the recent WHO classification and methylation profiles were confirmed with methylation EPIC array (#WG-317-1003, Illumina, San Diego, California, USA).

Due to the frozen nature of the obtained tissue, we needed to employ snRNA-Seq instead of scRNA-Seq. For single nuclei isolation, resected tumor material underwent the following quality control. Only material with a tumor content ≥ 70% and a low percentage of necrosis, as determined on hematoxylin and eosin-stained sections by a board-certified neuropathologist (Department of Neuropathology, University Hospital Heidelberg, Germany) was considered for further processing. Clinical and pathological characterization of patients are summarized in Supplementary Table 4. Human patient samples were manually anonymized.

### Single nuclei preparation

Nuclei isolation was accomplished as described[61]. Briefly, 10–20 mg of the tumor sections were roughly chopped on ice and resuspended in 1.5 ml lysis buffer consisting of 320 mM sucrose (#84097, Sigma, part of Merck, Darmstadt, Germany), 5 mM $CaCl_2$ (#21115, Sigma, part of Merck, Darmstadt, Germany), 3 mM $Mg(CH_3COO)_2$ (#63052, Sigma, part of Merck, Darmstadt, Germany), 2 mM EDTA (#AM9260G, Invitrogen, part of Thermo Fisher Scientific, Waltham, Massachusetts, USA), 0.5 mM ethylene glycol tetraacetic acid (EGTA, #J61721, Alfa Aesar, part of Thermo Fisher Scientific, Waltham, Massachusetts, USA), 1 mM dithiothreitol (DTT; #43816, Sigma, part of Merck, Darmstadt, Germany), 0.1% Triton X-100 (#A4975, AppliChem, Darmstadt, Germany) and 10 mM Tris(hydroxymethyl)aminomethan (Tris) pH 8.0 (#15568025, Life Technologies, part of Thermo Fisher Scientific, Waltham, Massachusetts, USA). The suspension was transferred to a dounce homogenizer (#9651617, Th. Geyer, Renningen, Germany) that was pre-coated with 0.1% Triton X-100 and nuclei were isolated applying 10 strokes with each pestle A and B. Large debris was removed by 100 μm (#542000, Greiner Bio-one, Kremsmünster, Austria) and 70 μm (#542070, Greiner Bio-one, Kremsmünster, Austria) strainer meshes and the suspension collected in separate 50 ml tubes (#227261, Greiner Bio-one, Kremsmünster, Austria). After each transfer, tubes were rinsed with 1 ml washing buffer (lysis buffer without DTT and Triton X-100) that was pooled with the collected suspension. Next, nuclei were subjected to three repeated wash cycles consisting of centrifugation (550 g, 5 min, 4 °C), supernatant removal and resuspension in 1.5 ml washing buffer. After the first cycle, suspension was transferred to microcentrifuge tubes (#0030.120.086, Eppendorf, Hamburg, Germany). Adaptions for the last cycle included addition of 500 μl homogenization buffer (320 mM Sucrose, 30 mM $CaCl_2$, 18 mM Mg(Ac)$_2$, 0.1 mM EDTA, 0.1% Nonidet P40 [#APA1694.0250, Applichem, Darmstadt, Germany], 0.1 mM phenylmethylsulfonyl fluoride [PMSF, #6367.2, Roth, Karlsruhe, Germany], 1 mM beta-Mercaptoethanol [#M7522, Sigma, part of Merck, Darmstadt, Germany], 60 mM Tris pH 8.0) to the nuclei pellet and a resting time of 5 min before resuspension in another 1 ml homogenization buffer. In rare cases (<10%), we opted for further purification using a iodixanol (#07820, Stem Cell Technologies, Vancouver, Canada) gradient. Briefly, the pellet was resuspended in 200 μl gradient buffer consisting of 30 mM $CaCl_2$, 18 mM $Mg(CH_3COO)_2$, 0.1 mM PMSF, 1 mM beta-Mercaptoethanol and 60 mM Tris pH 8.0. After transfer to a new microcentrifuge tube, 200 μl of 50% iodixanol in gradient buffer was used to generate a final concentration of 25% iodixanol. The nuclei suspension was carefully layered onto a gradient consisting of equi-voluminous 300 μl layers of 29% and 35% iodixanol in gradient buffer supplemented with 160 mM sucrose. Separation was performed on a swinging-bucket centrifuge Hereus™ Multifuge™ 40 (Thermo Fisher Scientific, Waltham, Massachusetts, USA) at 4 °C for 20 min with 3000 g. 200 μl of the nuclei-containing interphase was collected and

passed through a 20 μm filter (#130-101-812, Miltenyi Biotec, Bergisch Gladbach, Germany). Partially, trituration using wide-bore tips (#10089010, Thermo Fisher Scientific, Waltham, Massachusetts, USA) was necessary to facilitate disaggregation of the nuclei.

All aforementioned steps were performed on ice and all plastic consumables having contact with nuclei were pre-coated with 0.1% Triton X-100 prior to use to prevent sample loss.

Finally, integrity and purity of the nuclei was confirmed using Trypan Blue (#15250-061, LifeTechnologies, part of Thermo Fisher Scientific, Waltham, Massachusetts, USA) staining and the nuclei sequenced according to the 10x Genomics protocol (see section scRNA-Seq data generation from PDGCL xenografted models).

### Bioinformatic preprocessing, processing and integration and malignant cell state assignment

The count matrices of patient sample snRNA-Seq were generated using Cell Ranger software with standard parameters, against a custom pre-mRNA hg19 human reference genome generated by mkref function following the official guideline (https://support.10xgenomics.com/single-cell-gene-expression/software/pipelines/3.1/advanced/references). Single cell data processing and integration as well as malignant cell state assignment was accomplished the same way as for the SR101 xenograft datasets. In snRNA-Seq dataset, one unsupervised cluster that expressed markers of two different cell types was further removed. In the end, we obtained 213,444 cells from 21 patient samples.

### Identification of malignant and non-malignant cell types in snRNA-Seq of patient samples

**Cell type marker collections.** The top 100 upregulated markers per cell types (i.e., malignant cells, macrophages, T-cells and oligodendrocytes) were identified from a GB snRNA-Seq dataset[3] using the FindAllMarkers function with default parameters in Seurat (RRID:SCR_007322, v.3.1.5). The top 100 upregulated markers of endothelial cells were obtained from a healthy brain RNA-Seq dataset[22]. The top 100 enriched markers in pericytes were obtained from brain mural cells RNA-Seq dataset[23]. The upregulated markers of healthy astrocytes compared to malignant astrocytes were obtained from a human brain RNA-Seq dataset[22].

**Cell type signature scores.** In the patient integrated snRNA-Seq dataset, cell type signature scores (i.e., malignant signature score, macrophage signature score, T-cell signature score, oligodendrocyte signature score, endothelial signature score, pericyte signature score, and astrocyte signature score) based on cell type markers were calculated in each cell using the AddModuleScore function in Seurat (RRID:SCR_007322, v.3.1.5).

**Cell type assignment.** In the patient integrated dataset SNN unsupervised clustering was performed using the FindNeighbors function and the FindClusters function (resolution = 0.7), and 24 clusters were obtained. In each cluster, the medians of each cell type signature score were calculated and represented as $S_{ij}$, with $i$ being one cell type and $j$ being one cluster. Then the non-malignant scores $NMS_{ij}$ were defined as $S_{ij}$ minus malignant signature score $S_{mj}$ (m indicates malignant cells): $NMS_{ij} = S_{ij} - S_{mj}$. The clusters were assigned to non-malignant cell types if $NMS_{ij}$ more than MAD above the median of all $NMS_{ij}$: cluster 8, 9, and 23 as macrophages, cluster 5 as oligodendrocytes, cluster 19 as T-cells, cluster 22 as pericytes and cluster 17 as endothelial cells. The remaining clusters were assigned as malignant clusters and were validated based on CNV estimation using the infercnv (RRID:SCR_021140, v.1.2.1) with recommended parameters for 10x Genomics data (cutoff = 0.1, cluster_by_groups = TRUE, denoise = TRUE, HMM = TRUE). The assigned macrophages, oligodendrocytes, T-cells, pericytes and endothelial cells were used as reference non-malignant cells. Each non-malignant cell type and

malignant clusters were downsampled to 500 cells. We found that the malignant clusters contained large-scale CNVs except cluster 21. Cluster 21 showed the highest astrocyte signature score and, accordingly, cluster 21 was reassigned as astrocyte cluster.

### Two-dimensional projection of patient malignant cells by cell state

Similar to[3], we obtained signature scores for each cell state in single cells and projected the cells according to the cell state signature scores. Y axis values represent the maximum score from the AC/MES1/MES2 states from which the maximum score from the OPC/NPC1/NPC2 states have been subtracted. If Y > 0, the X axis values represent AC minus the maximum of MES1 and MES2. If Y ≤ 0, the X axis values represent OPC minus the maximum of NPC1 and NPC2. Cells were colored by connectivity scores and plotted by ggplot2 (RRID:SCR_014601, v.3.3.2).

$$Y = \max(S_{AC}, S_{MES1}, S_{MES2}) - \max(S_{OPC}, S_{NPC1}, S_{NPC2})$$

$$if\ Y > 0, X = S_{AC} - \max(S_{MES1}, S_{MES2})$$

$$if\ Y \leq 0, X = S_{OPC} - \max(S_{NPC1}, S_{NPC2}) \tag{2}$$

### Interactive web app

The interactive web app (https://connectivity-glioma.dkfz.de/) was implemented using R. The graphical user interface of the web app was constructed using the Shiny framework. It integrates metadata and normalized gene expression matrices from the SR101 scRNA-Seq dataset and the patient tumor scRNA-Seq dataset. UMAPs were generated utilizing the ggplot2 package (RRID:SCR_014601, v.3.3.2). Additionally, boxplots including statistical tests were created using the ggpubr package (RRID:SCR_021139, v.0.4.0) and scatterplots with correlation coefficients were created using the ggpubr package.

### Quantification of TMs in FFPE patient samples

Making use of the fact that tumors of patients enrolled in the $N^2M^2$ pilot study had been previously characterized with RNA-Seq[45] we selected the three patients with highest and lowest connectivity scores, respectively and available FFPE tissue. Manual quantification of TM number and length was done on nestin-stained FFPE sections as described before[4].

### General preparation of slides

FFPE blocks containing fresh fixed paraffin embedded patient resection specimens were obtained from the Department of Neuropathology in accordance with local ethical approval. 3 μm sections were generated using the HM 355 S automated microtom (#905200, Thermo Fisher Scientific, Waltham, Massachusetts, USA) and mounted on Superfrost® slides (#J1800AMNZ, Thermo Fisher Scientific, Waltham, Massachusetts, USA). Subsequent drying was allowed for 30 min on a 37 °C hot plate followed by baking for 10 min in a 75 °C oven.

**Nestin staining.** Nestin protein levels was detected using the ultraView DAB protocol on the automated VENTANA® BenchMark ULTRA platform (Roche, Basel, Switzerland).

To detect nestin expression slides were incubated with 1:200 diluted anti-nestin antibody, clone 10C2 (#MAB5326, RRID:AB_11211837, MerckMillipore, Burlington, Massachusetts, USA), for 32 min. VENTANA® standard signal amplification and ultra-wash was followed by counterstaining with Hematoxylin II (#790-2208, Roche, Basel, Switzerland) and blueing reagent (#760-2037, Roche, Basel, Switzerland) for

4 min each. Slides were removed from the staining platform, washed with tap water containing a drop of dishwashing detergent and rinsed with deionized water. After staining, all specimens were immersed in a series of ethanol (EtOH) solutions (#20821.330, VWR, part of Aventor, Radnor, Pennsylvania, USA) of increasing concentrations until 100% and Xylol (#534056-4L, Sigma, part of Merck, Darmstadt, Germany). Eukitt® (#6.00.01.0001.06.01.01, ORSAtec GmbH, Bobingen, Germany) was used for mounting.

**Hematoxylin-Eosin (HE) staining.** For dewaxing and rehydration sections were passed through xylol (#9713.3, Roth, Karlsruhe, Germany) and decreasing concentrations of EtOH (#200-678-6; Fisher Scientific, Waltham, Massachusetts, USA) until the solution evenly flowed across the slide. Staining with Mayer´s hematoxylin solution consisting of 0.1% hematoxylin (#1.04302.0100, Merck, Darmstadt, Germany), 0.02% sodium iodate (#6525; Merck, Darmstadt, Germany), 5% potassium aluminum sulfate (#8896.1; Roth, Karlsruhe, Germany), 5% chloralhydrate (#K318.1; Roth, Karlsruhe, Germany) and 0.1% citric acid (#3958.1; Roth, Karlsruhe, Germany) for 1 min was followed by blueing in running tap water for 3 min. Slides were incubated in eosin solution consisting of 10% Eosin G (#7089.2, Roth, Karlsruhe, Germany) and 2 drops of glacial acetic acid (#3738.1; Roth, Karlsruhe, Germany) in 70% EtOH (#200-678-6, Fisher Scientific, Waltham, Massachusetts, USA) for 30 s and subsequently rinsed in ddH$_2$O before mounting.

**Image analysis of patient tissue.** All slides were scanned at 20x resolution using an Axioscan Z1 slide scanner (RRID:SCR_020927, Zeiss, Jena, Germany). Zen 2.6 Blue Edition® software (RRID:SCR_013672, Zeiss, Jena, Germany) was used to globally adjust the copies of original photomicrographs for white and black balance. Photomicrographs were additionally cropped, rotated and resampled to allow alignment with other stainings. For image analysis three 500 × 500 pixel regions in each patient sample were selected based on number of nuclei (100 ± 20), nestin positivity and adjacency to denser tumor tissue. Then TMs were measured manually in these regions using Fiji 2.0.0. There were 20–84 TMs measured per image with a total of $n = 898$.

**Alignment of nestin and HE staining.** Zen 2.6 Blue Edition was used to globally adjust the copies of original photomicrographs for white and black balance. Photomicrographs were additionally cropped, rotated and resampled to allow alignment with other stainings. Subsequent removal of background shadows at the tile edges of no-sample containing tiles was done using Zen 2.6 Blue Edition.

**Target staining and TM quantification in patient tumor tissues**
CHI3L1 staining. Detection of CHI3L1 protein expression in sections adjacent to the ones used for nestin and HE stainings was carried out as described in section Quantification of TMs in FFPE patient samples, with the exception of an additional heat induced epitope retrieval step with CC1 solution (#05279801001, Roche, Basel, Switzerland) for 32 min. Slides were subsequently incubated with 1:1250 diluted anti-CHI3L1 antibody (#ab77528, RRID: AB_2040911, Abcam, Cambridge, United Kingdom), for 32 min.

Histoscoring of CHI3L1. A histoscore was used to assess the quantity of the CHI3L1 staining intensities of both global tumor tissue level but also of 500 × 500 pixel CHI3L1 ROIs aligned with the nestin ROIs, which had been independently selected before by a blinded person.

Histoscoring is a widely used semiquantitative classification of the staining intensity of heterogeneously stained tissues. Technically, the staining intensity of each individual cell is assigned to a scaled rating: 0 (negative), 1 (low), 2 (moderate), and 3 (high). A weighted histoscore is

calculated by the formula:

$$Weighted\ histoscore = \sum_{r=0}^{3} SI_r * P_r \qquad (3)$$

where $r$ represents the rating of staining intensity; $SI_r$ represents the staining intensity of cell with $r$; $P_r$ represents the percentage of cells with $r$ in the whole sample.

Based on this, the maximum score being reached is 300 (if 100% of cells have a high intensity) and the minimum score is 0 (if 100% of cells do not stain). All ratings were performed by a board-certified neuropathologist (Department of Neuropathology, University Hospital Heidelberg, Germany).

**Patient cohorts for validation of connectivity signature**
TCGA[62] cohort (RRID:SCR_003193, https://www.cancer.gov/tcga). The RNA-Seq gene expression matrix, somatic mutation information, CNV information and clinical data of TCGA diffuse glioma samples were downloaded from UCSC Xena (RRID:SCR_018938, http://xena.ucsc.edu). We obtained 146 samples from TCGA GB cohort and 502 samples from TCGA lower grade glioma cohort. We further investigated *IDH* mutation status and chromosome 1p/19q co-deletion status in all samples. Finally, we obtained 230 *IDH* wt samples, 176 *IDH* mut with 1p/19q co-deletion samples, 241 *IDH* mut without 1p/19q co-deletion samples and one sample without clear classification. The 230 *IDH* wt samples were derived from 90 female, 139 male und 1 unclear donor and subjected for connectivity signature validation and survival analysis.

CGGA[63] cohort (RRID:SCR_018802, http://www.cgga.org.cn). We downloaded clinical data and RNA-Seq gene expression matrix of 325 GB samples from the CGGA webpage, of which 141 samples had *IDH* wt and intact 1p/19q status. These 141 samples were subjected for connectivity signature validation and survival analysis.

Gene Expression Profiling Interactive Analysis (GEPIA[64], RRID:SCR_018294, http://gepia.cancer-pku.cn). We downloaded the median *CHI3L1* gene expression level (transcripts per million [TPM]) of RNA-Seq data from GEPIA, which contains 31 tumor types from TCGA and related normal tissue samples from the genotype-tissue expression (GTEx, RRID:SCR_013042, https://www.gtexportal.org/home/).

The glioma longitudinal analysis[28] (GLASS cohort). We downloaded clinical information and RNA-Seq gene expression matrix of 425 primary and recurrent samples from Synapse (RRID:SCR_005918, https://www.synapse.org/glass). These 425 samples were subjected for connectivity signature validation and survival analysis. 161 primary GB specimen and 133 specimen collected at first recurrence were analyzed.

The GBMap harmonized GB scRNA-Seq dataset[23]: The Seurat object containing the gene count matrix of 338,564 cells from 110 GB patients and the cell annotation metadata was retrieved from the CELLxGENE data portal (RRID:SCR_021059, https://cellxgene.cziscience.com/collections/999f2a15-3d7e-440b-96ae-2c806799c08c). 125,486 malignant cells from 74 donors were analyzed. Donors that had less than 20 malignant cells and/or 20 nonmalignant cells were removed.

GB proteogenomic[65] cohort. Of the 93 GB patients, derived from 42 female and 51 male donors, we downloaded proteomics data from the CPTAC Assay Portal (https://cptac-data-portal.georgetown.edu/cptac/s/S048) and paired RNA-Seq data from the GDC Cancer Portal (RRID:SCR_014514, https://portal.gdc.cancer.gov/projects/CPTAC-3). These data were subjected for the correlation analysis between RNA and protein expression levels of *CHI3L1* and connectivity signature scores.

## Molecular classification of TCGA RNA-Seq

The TCGA and CGGA *IDH* wt GB samples were classified into three expression subtypes (i.e., MS, CL and PN) by single sample GSEA analysis-based classification as described in[25] (ssGSEA, R codes from[25]). The FPKM expression matrix was used as input for ssGSEA and 100,000 permutations was performed to obtain p values for each subtype. Each sample was assigned to the subtype with the smallest *p* value.

## Patient survival analyzes

Patient survival analyzes were performed by the survival (RRID:SCR_021137, v.3.1-12) and survminer (RRID:SCR_021094, v.0.4.2): Kaplan-Meier survival analysis was performed in the category groups (e.g. three patient groups basing on quartiles of connectivity signature scores, or three patient groups basing on quartiles of gene expression levels) with overall survival times. CoxPH was performed in continuous values (e.g. connectivity signature scores, or gene expression levels) with age, gender, overall survival times and surgical interval times. TCGA expression subtype was further considered. Exponents of the coefficients (Exp. coef.) with 95% confidence intervals (CI, 95% int.) indicated the hazard ratio of higher connectivity signature scores and *CHI3L1* gene expression levels.

## Antibody and recombinant blocking in vitro experiments

15,000 S24-tGFP PDGCs per well were cultured in Matrigel® monolayer assay conditions as described in section In vitro Ca$^{2+}$ imaging assay. Either an anti-CHI3L1 blocking antibody (#MABC196, clone mAY, RRID:AB_2891310, Merck, Darmstadt, Germany) or IgG1 antibody (#401402, clone MG1-45, RRID:AB_2801451, Biolegend, San Diego, California, USA) was added to reach 19 nM final concentration. Preparation of cells, image acquisition and analysis was carried out as described in section In vitro Ca$^{2+}$ imaging assay. $n \geq 15$ ROIs were analyzed per condition in each of $n = 2$ independent experiments.

## *CHI3L1* OE

**Cloning.** For functional characterization of *CHI3L1* overexpression, the gene was placed under control of a human phosphoglycerate kinase 1 promoter (PGK1) promotor. For detection of the overexpressing cells a N-terminal monomeric green fluorescent protein Tag (TagGFP) followed be a 2 A self-cleaving peptide was added. For unlabeled cell lines, the N-terminal GFP-tag was replaced by a noncoding adapter sequence. The mRNA expression was stabilized by the addition of a C-terminal SV40 polyadenylation sequence. For virus generation the complete expression cassette was inserted in a lentiviral expression vector (rwpLENTI-PGK-R4-GW-R3-SV40-Puro/Core Facility Cellular Tools DKFZ, Heidelberg, Germany) by gateway multisite recombination technology (Thermo Fisher Scientific, Waltham, Massachusetts, USA). The expression vector contains a resistance maker for positive selection with puromycin.

**Virus production and infection.** S24wt, T269wt and P3XXwt were transduced with the Ctrl or *CHI3L1* overexpression plasmids as described in Section Cell Culture of PDGCLs, and 24 h after transduction virus containing medium was replaced by 1 µg/ml Puromycin (#A2856.0100, Applichem, Darmstadt, Germany) containing selection media. FACS was additionally used to select positive clones.

## Quantitative real-time polymerase chain reaction (qPCR)

**RNA extraction and cDNA synthesis.** Harvested PDGCs were washed with ice-cold PBS (#D8537-500ML, Sigma, part of Merck, Darmstadt, Germany). Afterwards, PDGCs were resuspended in 1% beta-Mercaptoethanol (#M3148-100ml, Sigma, part of Merck, Darmstadt, Germany)-supplemented RLT lysis buffer, which is part of the QIAGEN RNeasy MicroKit (#79216, Qiagen, Hilden, Germany) or QIAGEN RNeasy Mini Kit (#74004, Qiagen, Hilden, Germany). Lysate was homogenized with QiaShredder columns (#79654, Qiagen, Hilden, Germany).

The kit type for subsequent RNA extraction was tailored to the absolute cell numbers. Lysates containing up to 500,000 cells were processed with the QIAGEN RNeasy® Micro Kit whereas samples with 500,000 to one million cells were processed with the QIAGEN RNeasy® Mini Kit. All steps were carried out according to the manual. On-column DNAse digestion was performed with the RNAse free DNAse set (#79254, Qiagen, Hilden, Germany). RNA was eluted in RNAse-free water (#4387936, Thermo Fisher Scientific, Waltham, Massachusetts, USA). Reverse transcription was performed according to the manufacturer´s recommendations using the High-Capacity cDNA Reverse Transcription Kit with RNAse Inhibitor (#4374967, Applied Biosciences Applied Biosciences, Foster City, California, USA) and a total of 1 µg RNA per 20 µl reaction.

**Amplification.** qPCR was performed with 9 ng cDNA, Taqman™ Gene Expression Master Mix (#4369016, Thermo Fisher Scientific, Waltham, Massachusetts, USA) and the respective TaqMan™ probes (Applied Biosystems, Foster City, California, USA). The following probes were used: Hypoxanthine Phosphoribosyltransferase 1 (HPRT1; Hs002800695_m1) and CHI3L1 (Hs01072228_m1). All reactions were carried out in a 96-well reaction plate (#N8010560, Applied Biosciences), covered with MicroAmp™ optical adhesion film (#4311971, Applied Biosciences, Foster City, California, USA) and analyzed on a QuantStudio™ 3 Real Time PCR System operated with the QuantStudio Design & Analysis software v1.5.2 (RRID:SCR_018712, Thermo Fisher Scientific, Waltham, Massachusetts, USA).

$n \geq 2$ independent experiments with each having $\geq 2$ technical replicates were performed. Standard curves were generated for each gene and the amplification was 85–115% efficient. Relative quantification of gene expression was determined by the delta-delta $C_T$ method and included normalization to GAPDH, HPRT1 and TBP using the Bestkeeper© software[66].

## Western blot

Cell lysis was performed with ice-cold RIPA buffer consisting of 50 mM Tris, pH 7.4 (#A1086,1000, AppliChem Panreac, Darmstadt, Germany), 150 mM NaCl (#31434-1KG, Sigma, part of Merck, Darmstadt, Germany), 1% Triton X-100 (#A4975, AppliChem Panreac, Darmstadt, Germany), 0.5% sodium deoxycholate (#A1531-0100, AppliChem Panreac, Darmstadt, Germany), 1 mM EDTA (#AM9260G, Thermo Fisher Scientific, Waltham, Massachusetts, USA), 1 mM EGTA (#15425795, Alfa Aesar, part of Thermo Fisher Scientific, Waltham, Massachusetts, USA), 0.5 mM PMSF (#6367.2, Roth, Karlsruhe, Germany), Complete protease inhibitor (#4693132001, Roche, Risch, Switzerland) and HALT Phosphatase inhibitor (#78420, Thermo Fisher Scientific, Waltham, Massachusetts, USA). Protein concentrations were measured using the BCA kit (#23225, Thermo Fisher Scientific, Waltham, Massachusetts, USA). 10 µg protein diluted with NuPAGE LDS 4x sample buffer (#NP007, Thermo Fisher Scientific, Waltham, Massachusetts, USA) and NuPAGE 10x sample reducing agent (#NP004, Thermo Fisher Scientific, Waltham, Massachusetts, USA) was denatured and electrophoretically separated on NuPAGE 4–12% Bis-Tris Mini Gel (#NP0321BOX, Thermo Fisher Scientific, Waltham, Massachusetts, USA) in Novex™ NuPAGE™ MOPS SDS running buffer (#NP001, Thermo Fisher Scientific, Waltham, Massachusetts, USA) with NuPAGE antioxidant (#NP005, Thermo Fisher Scientific, Waltham, Massachusetts, USA). Proteins were blotted onto a PVDF membrane (#IPVH00010, Merck, Darmstadt, Germany) and blocked for 1 h with 5% milk (#T145.2, Roth, Karlsruhe, Germany) or 5% BSA in TBS-T (5 mM Tris [#A1086,1000, AppliChem Panreac, Darmstadt, Germany], 16 mM Tris-HCl [#9090.3, Roth, Karlsruhe, Germany], 150 mM NaCl [#3134-1KG-M, Sigma, part of Merck, Darmstadt, Germany], 0.15% Tween 20 [#500-018-3. MPBio, Santa Ana, California, USA]). The following antibodies were used for subsequent probing overnight: anti-CHI3L1, 1:2000 (#MABC196, clone mAY, RRID:AB_2891310, Merck, Darmstadt, Germany), anti-GAP43, 1:1000 (#8945 S, clone D9C8, RRID:AB_10860076, Cell signaling technologies,

Danvers, Massachusetts, USA), anti-GAP43(pS41) 1:1000 (#ab167162, clone EPR1854(2), Abcam, Cambridge, United Kingdom) and anti-GAPDH, 1:1000 (#97166 S, clone D4C6R, RRID:AB_2756824, Cell signaling technologies, Danvers, Massachusetts, USA) as loading control. After three wash steps with TBS-T, 1 h incubation with 1:10,000 diluted HRP-coupled anti-mouse (NA931V, RRID:AB_772210, GE Healthcare, Chicago, Illinois, USA), anti-rabbit (NA9340V, GE Healthcare, Chicago, Illinois, USA) or Dy800-conjugated anti-mouse (#SA5-10172, RRID:AB_2556752, 1:10,000 dilution, Thermo-Fisher Scientific, Waltham, Massachusetts, USA) secondary antibody in 5% BSA or milk and another three wash steps with TBS-T, chemiluminescent signal induced by Clarity Western ECL Substrate (#170-5060, Bio-Rad, Hercules, California, USA) was visualized on a ChemiDoc MP Imaging System equipped with the Image Lab Touch Software v.2.0.0.27 (Bio-Rad, Hercules, California, USA). Image analysis was performed with Image Lab v.6.0.0 software (Bio-Rad, Hercules, California, USA).

### In vitro Matrigel® monolayer assay of *CHI3L1* OE cells

A total of 15,000 S24-Ctrl or S24-*CHI3L1* OE PDGCs were seeded per well in a 96-well plate and cultured under Matrigel® monolayer assay as described in section In vitro $Ca^{2+}$ imaging assay. Preparation, image acquisition and Ilastik-based TM-length quantification was done as described in section In vitro $Ca^{2+}$ imaging assay, with the only exception of addition of 4 nM MemGlow488/Lipilight (#MG01-02, Cytoskeleton Inc., Denver, Colorado, USA) before imaging to facilitate TM visibility. Representative images are maximum intensity projections, Gaussian filter with Kernel size 3 applied and objects detected in Ilastik.

### In vivo examination of *CHI3L1* OE cells

**PDGC implantation and tumor harvest.** A total of 50,000 S24-Ctrl or S24 *CHI3L1* OE PDGCs were injected into the striatum of 8-12 week old female Crl:CD1-*Foxn1^{nu}* nude mice (RRID:IMSR_CRL:086, Charles River, Wilmington, Massachusetts, USA) as previously described in section Separation of SR101^{high} and SR101^{low} PDGCL xenograft groups. $n = 4$ mice per group were injected in $n = 2$ independent experiments. Brains were harvested after a solid tumor had manifested, post-fixed in 4% PFA (#sc-281692, Santa Cruz, Santa Cruz, California, USA), incubated in 20% sucrose (#S0389-500G; Sigma, part of Merck, Darmstadt, Germany) for 24 h and embedded into Tissue Tek O.C.T compound (#4583, Sakura Finetek, Alphen aan den Rijn, Netherlands).

**Histological processing.** Serial sections of 50 μm were generated using a cryostat (#CM3050S, Leica, Wetzlar, Germany) and transferred to PBS (#PBS-1A, Capricorn Scientific, Ebsdorfergrund, Germany) to allow free floating immune stainings. Sections were rinsed twice with each TBS and TBS-T consisting of 11,5 mM Tris (#A1086,1000, AppliChem Panreac, Darmstadt, Germany), 38,45 mM Tris-HCl (#9090.3, Roth, Karlsruhe, Germany), 0.9% (w/v) NaCl (#3134-1KG-M, Sigma, part of Merck, Darmstadt, Germany) and, in the latter case, addition of 0.5% Triton X-100 (#X100-1L, Sigma, part of Merck, Darmstadt, Germany). After 90 min of blocking in 0.25% BSA (#0163.4, Roth, Karlsruhe, Germany) and 10% donkey or normal goat serum (#5425, Cell signaling technologies, Danvers, Massachusetts, USA) in TBS-T, the following primary antibodies were added for 72 h: Nestin, 1:500 (#ab6320, clone 196908, RRID:AB_308832, Abcam, Cambridge, United Kingdom), to allow TM visualization, CD31, 1:100 (#AF3628, RRID:AB_2161028, R&D Systems, Minneapolis, Minnesota, USA) to allow vessel visualization, and Ku-80, 1:400 (#2180 S, clone C48E7, RRID:AB_2218736, Cell signaling technologies, Danvers, Massachusetts, USA) to allow normalization for S24 PDGCL cell number. Four washes with TBS-T were followed by a 12 h incubation with 1:500 diluted goat anti-mouse and anti-rabbit secondary antibodies conjugated to Alexa Fluor 546 and AlexaFluor 633, respectively (#A11010, RRID:AB_2534077; #A11003, RRID:AB_141370; #A21070, RRID:AB_2535731; #A21050, RRID:AB_141431; Thermo Fisher Scientific, Waltham, Massachusetts, USA). After two consecutive 15 min washes with TBS-T and one with TBS, sections were stained with DAPI (10 ug/ml; #6335.1, Roth, Karlsruhe, Germany) for 15 min followed by one 15 min wash cycle with TBS and three cycles with TB. Sections were mounted onto standard microscope slides (#AA00000112E01MNZ10, Epredia, Portsmouth, New Hampshire, USA) and covered in Aqua Poly-Mount (#F4680-25ML, Sigma, part of Merck, Darmstadt, Germany).

**Image acquisition and quantification.** All ROIs selected for TM quantification were located in the caudoputamen and had a similar PDGCL density. Images from at least 4 mice per group were acquired on a LSM780 confocal microscope (Zeiss, Oberkochen, Germany) equipped with a 63x (NA1.4) Oil DICIII objective (Zeiss, Oberkochen, Germany) and Zeiss Zen 2012 black edition v.8.1.0.484 software (Zeiss, Oberkochen, Germany). The following excitations and detection wavelengths were used for sequential image acquisition: 405/410-585 (DAPI), 561/569-631 (AlexaFluor546) and 633/638-747 (AlexaFluor633). Image dimensions were set to 285x285x32 μm with a z-stack distance of 400 nm and a pixel size of 72 nm.

Quantification of cell number and TM length/cell within a ROI ($n = 4$ ROIs per mouse, $n = 4$ mice per group) was accomplished using Aivia (Leica, Wetzlar, Germany). A customized training set was used to threshold the pixel probabilities of two classes (nuclei and TM) and allow automatic segmentation. Features such as number and length of the detected nuclei and TMs were exported for further analysis. For performance quality control of the algorithm TM length/cell and number of nuclei were also assessed manually in Fiji 2.0.0 based on the raw images and confirmed the results obtained with Aivia. Number of nuclei per ROI and TM length/cell per ROI were validated manually from raw images using Fiji 2.0.0. Moreover, Fiji was used for the assessment of the number of TMs/cell and number of connections/cell ($n = 40$ cells, 4 mice per group). Perivascular cells were defined as such if cell bodies had contact with vessels.

A histoscore from S24 cells in $n = 16$ ROIs of $n = 4$ mice per condition was calculated to quantify CHI3L1 intensity in situ. The same methodology as for CHI3L1 in situ quantification in human patient samples was used.

### RNA-Seq of *CHI3L1* OE cell lines

**RNA-Seq.** RNA isolation and RNA-Seq were conducted as described before.

**Bioinformatic processing.** RNA-Seq data was aligned using STAR (v.2.5.3a) against the human genome (1KGRef_PhiX). Reads duplication was marked using Sambamba (v.0.6.5). A gene-count matrix was generated using Featurecounts (Subread v.1.6.5). The parameter settings wer based on the DKFZ ODCF workflow (https://github.com/DKFZ-ODCF/RNAseqWorkflow). Differential expression analysis was performed between the *CHI3L1* OE and Ctrl groups in each cell line, utilizing DESeq2 (v.1.32.0). Genes with adjusted $p$-values < 0.05, normalized count levels >10, and absolute fold-change ≥3 were retained as DEGs. The remaining DEGs that were identified in at least two cell lines and ranked within the top 100 upregulated or top 100 downregulated genes in each cell line were merged. Ultimately, a total of 353 DEGs were derived from the RNA-Seq dataset.

### Proteomics and phosphoproteomics of *CHI3L1* OE cell lines

**Mass spectrometry sample processing and data analysis.** Three PDGCL experiments with S24 ($n = 2$), T269 ($n = 1$) and P3XX ($n = 2$) overexpressing *CHI3L1* were compared with their empty vector counterparts, resulting in 10 samples for label free relative protein mass spectrometry (MS) quantification and 10 samples for phosphopeptide quantification.

**Protein extraction.** For label free proteome and phosphoproteome analysis, cell pellets containing 500 μg protein were lysed with 200 μl

8 M urea lysis buffer containing: Tetraethylammonium bromide (TEAB) 50 mM pH 8.5, 8 M urea, 1 mM NaCl, 1% Benzonase (Sigma), one Protease inhibitor tablet (Complete Tablets Mini, EDTA-free EASY pack) and PhosSTOP protease inhibitor buffer (Sigma-Aldrich) per 10 mL urea buffer. Residual cell debris was removed by centrifugation at 14,000 g for 10 min at 4 °C. The protein concentration was determined by the micro BCA protein assay kit (#23235, Thermo Fisher Scientific, Waltham, Massachusetts, USA) according to the manufacturer's instructions.

**Protein digestion.** Protein reduction and alkylation was performed with 10 mM DTT at 27 °C and 30 mM IAA for 1 h at RT. A Wessel-Flügge protein cleanup was conducted according to[67]. Protein lysates were diluted 1:5 with 50 mM TEAB (pH 8.5). 500 μg protein were digested with Trypsin/Lys-C (Promega) at 1:50 enzyme:protein ratio overnight at 37 °C. To stop the reaction, 10% formic acid (FA) was added to a final concentration of 2% FA.

**Peptide desalting and enrichment.** Prior to phosphopeptide enrichment, samples were desalted using SepPak tC18 100 mg 1cc solid Phase extraction cartridges (Waters) following the manufacturer's instructions. In brief, protein digests were equilibrated on-column washed with 2.5% FA and eluted with 80%/0.6% acetonitrile (MeCN)/FA in water. Peptide yields were quantified by nanoESI-LC-MS/MS. Desalted peptide samples were directed to speed vac to dryness.

**Sequential phosphopeptide enrichment.** A consecutive phosphopeptide SMOAC enrichment protocol based on metal affinity chromatography using High-Select™ TiO$_2$ combined with High-Select Fe-NTA phosphopeptide enrichment kits (Thermo Fisher Scientific Waltham, Massachusetts, USA) were applied to peptide samples according to the manufacturer's instructions. For SMOAC, the wash fractions of the HiSelect TiO$_2$ phosphopeptide enrichment were combined and applied to the Fe-NTA FT enrichment. After eluting the phosphopeptides from TiO$_2$ and Fe-NTA, both eluates were combined and analyzed by nanoLC-MS/MS.

**nanoLC MS/MS analysis.** Peptide samples prepared for global proteome and phosphoproteome analysis were separated and analyzed by nanoflow LC-MS/MS using a Dionex 3000 nanoUHPLC (Thermo Fisher Scientific, Waltham, Massachusetts, USA) attached to an Orbitrap Exploris (Thermo Fisher Scientific, Waltham, Massachusetts, USA) mass spectrometer. Samples for the proteome or phosphoproteome analysis were resuspended in MS loading buffer containing 2.5% 1,1,1,3,3,3-hexafluoro-2-propanol (HFIP), 0.1% trifluoroacetic acid (TFA) in water. Peptide loading and washing was performed for 3 min with 0.1% TFA in water at a flow rate of 30 μl/min. using a trapping cartridge Acclaim PepMap300 C18, 5μm, 300 Å (Thermo Fisher Scientific Waltham, Massachusetts, USA). Peptides were separated on a nanoEase, 1.7 μm, 300 Å, 75 μm x 200 mm analytical column (Waters) at a flow rate of 300 nl/min. For whole proteome analysis a three step 150 min gradient was applied for chromatography: 2–4% solvent B (99.9 % MeCN, 0.1% FA) in 4 min, 4–30% in 132 min and 30–80% in 3 min followed by a washing and an equilibration step with solvent A being 0.1% FA in water. For phosphopeptide separation, the nanoUHPLC method was adjusted as follows: 2–4% solvent B in 4 min, 4–28% in 132 min and 28–78% in 3 min followed by a washing and an equilibration step. The spray voltage was 2.2 kV for nanoESI ionization and the ion transfer tube temperature was set to 275 °C. The MS instrument operated in the data-dependent (DDA) mode. Full scan MS spectra (m/z 375–1400) were acquired with a maximum injection time of 45 ms at 60,000 resolution for full proteome and 120,000 for phosphoproteome analysis. The automatic gain control (AGC) target value was set to 200% and the isolation window was set to 1.2 m/z. The normalized MS collision energy was set to 28. MS/MS scans cycles were triggered for 2 sec. A maximum injection time of 54 ms at 15,000 resolution was set

for high-resolution MSMS spectra. Dynamic exclusion was set to 10 sec. Undetermined charge states and single charged signals were excluded from fragmentation.

**Protein and phosphopeptide identification and quantification.** MS raw data was processed by MaxQuant v.2.0.1.0 (RRID:SCR_014485) software package including Perseus v.1.6.15.0 (RRID:SCR_015753) for statistical analysis. Protein as well as phosphopeptides were identified applying the UniProt database UP000000589 (Homo sapiens; 01, 2020; 20367 sequences, RRID:SCR_002380). Carbamidomethylation of cysteines was set as fixed modification. Phosphorylation of serine, threonine or tyrosine as well as oxidation of methionine, N-terminal acetylation, glutamine and asparagine deamidation were set as variable modifications. The 'match-between-runs' function and LFQ option was enabled for label free quantification. Identification FDR cutoffs were 0.01 on the protein level and peptide level. Phosphopeptides with phosphosite localization probabilities x ≥ 0.75 were selected for further analysis. The proteins or phosphosites only detected in one sample were removed for further processing. 5022 proteins were kept and a median of 4286 proteins per sample were obtained in proteomics dataset. 12,799 phosphosites were kept and a median of 8520 phosphosites per sample were obtained in the phosphoproteomics dataset.

**DEP and DPP identification.** The LFQ data was normalized by 'vsn' method using DEP (v.1.14.0). The intensity distribution and cumulative fraction of proteins in the proteomics and phosphoproteomics datasets indicated that proteins with missing values had lower intensities. This observation suggests that the proteins with missing values were below the detection limit. To handle the missing values, we employed the deterministic minimum (MinDet) method for imputation. This method replaced each missing value with the smallest detectable intensity (0.01 quantile) observed within each sample. 152 DPPs were identified in *CHI3L1* OE samples against control samples with adjust *p* value < 0.05 and absolute log2 fold change >1.5 by test_diff function using DEP (v.1.14.0). Due to high differences among cell lines in proteomics dataset, we further corrected the normalized and imputed proteomics data by removeBatchEffect function using limma (RRID:SCR_010943, v.3.36.5). Then, the corrected data was fitted to linear model using lmFit function, and calculated the empirical Bayes statistics using eBayes function. 123 DEPs were identified in *CHI3L1* OE samples against control samples with adjust *p* value < 0.05 and absolute log2 fold change >0.5.

### Pathway analysis of Ctrl and *CHI3L1* OE PDGCLs

Ingenuity Pathway analysis (Qiagen, Hilden, Germany, RRID:SCR_008653) was used to predict pathways activated in *CHI3L1* OE PDGCLs based on DEGs. Kinase enrichment analysis was conducted with KEA3 as described[68].

### Statistics & reproducibility

All in vitro and in vivo experiments were performed in a randomized fashion and conducted independently at least two times unless otherwise specified. No statistical method was used to predetermine sample size in all experiments. Our sample sizes were selected based on those reported to generate statistically meaningful data in similar studies from our group. No data were excluded from the analyzes except specified otherwise in the respective methods sections. The exact number of independent replicate samples (*n*) and statistical parameters are provided in the legends corresponding to the specific experiments.

The investigators were blinded to allocation during experiments and outcome assessment whenever possible. Further information can be found in the respective methods sections and the Reporting Summary.

For normally distributed data and/or datasets with n ≥ 5 data points the statistical significance was assessed by t-test and one-way ANOVA,

respectively. Among the recurrent non-synonymous mutated genes in at least 5% TCGA GB patients (27 genes), the connectivity signature score related mutated genes were identified using wilcox.test function in R.

Mean comparisons between two groups without an equal distribution and/or $n \geq 5$ was obtained by Mann-Whitney U test or among three groups by Kruskal-Wallis test. All statistical tests were two-sided if not stated otherwise and conducted with ggpubr (RRID:SCR_021139, v.0.4.0). Error bars show standard error of the mean (SEM) and all boxplots are according to Tukey. Boxes show 25th to 75th percentile, its middle line the median, whiskers the 5th to 95th percentile and individually plotted data points the outliers.

Multiple testing was adjusted and obtained FDR using p.adjust function in R. Pearson or Spearman correlation coefficients were calculated using ggpubr (RRID:SCR_021139, v.0.4.0). Statistical significance for the overlap between two gene sets was calculated by hypergeometric probability via http://nemates.org/MA/progs/overlap_stats.html. Multivariate analysis was conducted to correct for age and gender. A $p$ value of $p < 0.05$ was generally considered significant.

### Visual illustration
Parts of the illustrations were drawn by using pictures from Servier Medical Art. Servier Medical Art by Servier is licensed under a Creative Commons Attribution 3.0 Unported License (https://creativecommons.org/licenses/by/3.0/).

Additionally, BioRender.com was used for the creation of the illustrations.

### Reporting summary
Further information on research design is available in the Nature Portfolio Reporting Summary linked to this article.

## Data availability
Raw data files of all WES, bulk RNA-Seq and scRNA-Seq data generated in this study have been deposited in the European Genome-Phenome Archive database (EGA) under accession number EGAS00001007611 [https://ega-archive.org/studies/EGAS00001007611, RRID:SCR_004944]. These patients' sensitive genetic data, including raw and processed data files, are available in line with the EGA policy and access controlled by the Data Access Committee to ensure patient privacy. All of WES, bulk RNA-Seq and scRNA-Seq data generated and analyzed in this study are provided in the Supplementary Information, Supplementary Data and Source Data. Raw MS data generated in this study have been deposited in the ProteomeXchange Consortium (http://proteomecentral.proteomexchange.org, RRID:SCR_004055) via the PRIDE partner repository (RRID:SCR_003411) with the dataset identifier PXD044001 [https://proteomecentral.proteomexchange.org/cgi/GetDataset?ID=PXD044001]. The MS data generated in this study are provided in the Supplementary Information, Supplementary Data and Source Data. scRNA-Seq data from the SR101 dataset can be additionally explored by an interactive web app (https://connectivity-glioma.dkfz.de). RNA-Seq data of the TCGA cohort was obtained from the UCSC Xena platform [https://xenabrowser.net/datapages/?cohort=GDC%20TCGA%20Lower%20Grade%20Glioma%20(LGG)&removeHub=https%3A%2F%2Fxena.treehouse.gi.ucsc.edu%3A443, https://xenabrowser.net/datapages/?cohort=GDC%20TCGA%20Glioblastoma%20(GBM)&removeHub=https%3A%2F%2Fxena.treehouse.gi.ucsc.edu%3A443, RRID:SCR_018938, https://www.cancer.gov/tcga]. RNA-Seq data of the CGGA cohort was obtained from the CGGA webpage [http://www.cgga.org.cn/download.jsp, RRID:SCR_018802]. RNA-Seq data from 31 tumor types and related healthy tissues was obtained from the GEPIA server [http://gepia.cancer-pku.cn/detail.php?gene=chi3l1, RRID:SCR_018294]. RNA-Seq data from the GLASS cohort was obtained from the Synapse platform [https://www.synapse.org/glass, RRID:SCR_005918]. scRNA-Seq data from GBmap was obtained from the CELLxGENE data portal [https://cellxgene.cziscience.com/collections/999f2a15-3d7e-440b-96ae-2c806799c08c,

RRID:SCR_021059]. GB proteogenomic cohort was obtained from the CPTAC Assay Portal [https://cptac-data-portal.georgetown.edu/cptac/s/S048] and the GDC Cancer Portal [https://portal.gdc.cancer.gov/projects/CPTAC-3, RRID:SCR_014514]. Gene sets for GSEA were obtained from gsea-msigdb.org (RRID:SCR_003199, v.4.1.0, Broad Institute). All of these datasets were accessible without any restriction. Source data is available. Source data are provided with this paper.

## Code availability
All codes supporting the current study, including the ones underlying the interactive web app (http://connectivity-glioma.dkfz.de) have been deposited in Zenodo under https://zenodo.org/records/10481241.

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

## Acknowledgements

We thank Hrvoje Miletic for providing the P3XX and BG5 cell line. We are grateful to the Light Microscopy Facility and FACS Facility at the DKFZ for support in all kinds of various aspects. We thank Jan-Philipp Mallm and the scOpen lab of the DKFZ, Heidelberg, Germany for providing support with the 10x Chromium Controller for scRNA-Seq. Sequencing capacity was kindly provided by the Genomics and Proteomics Core Facility of the DKFZ. We are grateful to the IT core facility of the DKFZ for providing the URL and hosting the web application (http://connectivity-glioma.dkfz.de) and the Omics IT and Data Management Core Facility of the DKFZ for providing data storage capacity. We thank Katja Bauer and Birgit Kaiser, Laura Doerner, Lea Hofmann, Moritz Schalles and Hai-Yen Nguyen for expert technical support. Tim Holland-Letz provided expert statistical support. We are grateful for Nils Hebach´s expert consulting. The sequencing analyzes were supported by a DKFZ-HIPO grant (Heidelberg Center for Personalized Oncology, H057) to W.W., a DKFZ-HIPO grant (K25) to W.W. and T.K. and a DKFZ-HIPO grant for scRNA-Seq specifically dedicated for this work (K32) to F.W., W.W., M.S. and T.K. The work was supported by the SFB grant UNITE Glioblastoma (SFB1398, WP A03 to W.W. and T.K., WP A01 to F.W. and E.J. and WP D02 to M.S.) of the German Research foundation (DFG). The proteomic analyzes were supported by the DKFZ and the NCT Heidelberg (NCT 3.0. G840). The Caprola work was supported by the Max Planck Society. M.-C.H. was supported by the Heidelberg Biosciences International Graduate School (HBIGS) and the Boehringer Ingelheim Foundation.

## Author contributions

L.H. performed bioinformatic analysis of RNA-Seq, scRNA-Seq, snRNA-Seq and mass spectrometry-based proteomic and phosphoproteomic data, analyzed pathways as well as TCGA and CCGA patient survival, developed the interactive web app and contributed to all aspects of the study. D.C.H. designed, conducted, analyzed and interpreted experiments and contributed to all aspects of the study, in particular establishment and execution of single cell-and nucleus RNA sequencing workflows, widefield and confocal imaging of live-cell, immunofluorescence and IHC GB network activity and morphology, qPCR and Western Blot. Additionally, IPA and GO term-based pathway analysis was carried out. R.J.W. performed tumor injections, immunofluorescence sample preparation and provided methodological and conceptual input. D.H. conducted and analyzed Ca2+ imaging data, Caprola and BTP2 experiments, provided conceptual and methodological input and interpreted data. D.D.A. conducted analysis of TM-network parameters. R.X. provided PDGCS from SR101 dye models. M.-C.H. and J.H. generated the Caprola vectors, established labeling and sorting strategies for Caprola PDGCs and provided conceptual input. P.S. provided staining of human paraffin sections under the supervision of F.S. S.He. conducted TM network analysis of human paraffin sections. J.I. provided methodological and conceptual input for PDGCL cultivation. G.C. performed sample preparation for mass spectrometry-based proteomics and phosphoproteomics. A.K. generated Caprola PDGCLs and performed sequencing. L.D.K. provided methodological and conceptional input on various aspects of the study. M.R. performed two-photon microscopy experiments, image analysis and provided conceptional and intellectual input. H.M. conducted serum-model experiments, Smart-Seq2 scRNA-Seq and provided conceptual input under the supervision of M.L.S. E.J. performed two-photon microscopy of SR101 in vivo experiments, image analysis and conceptual input. A.J. and S.Ho. provided methodological and conceptual input for immunofluorescence. K.E. provided methodological and conceptual input for snRNA-Seq. D.R. performed snRNA-Seq, cell culture, qPCR, Western Blot, immunofluorescence and confocal imaging. U.W. analyzed mass spectrometry-based proteomic and phosphoproteomic data. V.V. performed two-photon microscopy of PDGC TM-networks in vivo and immunofluorescence of patient samples. R.W. provided Ctrl and CHI3L1 OE PDGCLs. C.H.M. provided patient tissue. F.W. provided conceptual input. M.S. provided conceptual input, performed data interpretation and supervised RNA-expression data analysis. W.W. provided conceptual input and supervised all aspects of the study. T.K. conceptualized and supervised all aspects of the study and performed data interpretation. L.H., D.C.H. and T.K. wrote the manuscript with the input of all co-authors.

## Funding

## Competing interests

E.J., W.W. and F.W. report the patent (WO2017020982A1) entitled Agents for use in the treatment of glioma filed by the Ruprecht-Karls-Universität Heidelberg and Deutsches Krebsforschungszentrum Stiftung des öffentlichen Rechts. F.W. is co-founder of DC Europa Ltd (a company trading under the name Divide & Conquer) that is developing new medicines for the treatment of glioma. M-C.H. and J.H. report the patent (WO2020212537A1) entitled Circular permuted haloalkane transferase fusion molecules filed by the Max Planck Society. The remaining authors declare no competing interests.

## Additional information

[1]Bioinformatics and Omics Data Analytics, German Cancer Research Center (DKFZ), Heidelberg, Germany. [2]Clinical Cooperation Unit Neurooncology, German Cancer Consortium (DKTK), German Cancer Research Center (DKFZ), Heidelberg, Germany. [3]Department of Neurology and Neurooncology Program, National Center for Tumor Diseases, Heidelberg University Hospital, Heidelberg, Germany. [4]Faculty of Biosciences, Heidelberg University, Heidelberg, Germany. [5]Department of Chemical Biology, Max Planck Institute for Medical Research, Heidelberg, Germany. [6]Department of Neuropathology, Institute of Pathology, University Hospital Heidelberg Heidelberg, Germany. [7]Clinical Cooperation Unit Neuropathology, DKTK, DKFZ, Heidelberg, Germany. [8]Neurosurgery Clinic, University Hospital Mannheim, Mannheim, Germany. [9]Medical Faculty Mannheim, Central Institute of Mental Health, Heidelberg University, Mannheim, Germany. [10]Hector Institute for Translational Brain Research, Mannheim, Germany. [11]German Cancer Research Center (DKFZ), Heidelberg, Germany. [12]Institute of Reconstructive Neurobiology, School of Medicine & University Hospital Bonn, University of Bonn, Bonn, Germany. [13]Pediatric Glioma Research Group, DKTK, DKFZ, Heidelberg, Germany. [14]Hopp Children's Cancer Center at the NCT Heidelberg (KiTZ), Heidelberg, Germany. [15]Department of Neuroanatomy, Institute for Anatomy and Cell Biology, Heidelberg University, Heidelberg, Germany. [16]Genomics and Proteomics Core Facility, DKTK, DKFZ, Heidelberg, Germany. [17]Broad Institute of Harvard and MIT, Cambridge, MA, USA. [18]Department of Pathology and Center for Cancer Research, Massachusetts General Hospital and Harvard Medical School, Boston, MA, USA. [19]Department of Neurosurgery, Heidelberg University Hospital, Heidelberg, Germany. [20]Biomedical Informatics, Data Mining and Data Analytics, Faculty of Applied Computer Science and Medical Faculty, University of Augsburg, Augsburg, Germany. [21]These authors contributed equally: Ling Hai, Dirk C. Hoffmann. ✉e-mail: t.kessler@dkfz.de

