## [Peer Review File · Nature Communications]

REVIEWER COMMENTS

Reviewer #1 (Remarks to the Author): Expert in glioblastoma tumour microenvironment, intravital microscopy, and in vivo models

The manuscript by Hai et al. describes a novel “connectivity signature” for GB and the role of CHI3L1 in the formation of tumor microtubes.

The authors use a novel dye uptake methodology, scRNA-Seq and orthotopically implanted patient-derived GB to isolate the highly connected GB cells within the tumor to then investigate the geneset signature of this novel cell state (“located” between AC and MES-like cell states). Using data mining the authors demonstrate that this gene signature is correlated with poor patient survival. Moreover, overexpression of CHI3L1 in GB cells allowed them to mechanistically demonstrate targetability of the connectivity GB cell state.

The clinical and pre-clinical relevance of tumor microtubes is suggested by a rich recent literature on this topic, but still a solid geneset was missing. The abstract and method part are well written and the results are highly interesting.

Overall, the experiments were really well performed and the data support the conclusions the authors made. There are, however, a small number of issues that need to be addressed or clarified.

Major issues:

1. Although I think that the SR101 method to isolate highly connected GB cells is smart, I must confess that there is a potential issue in it. With it, the authors are enriching for the GB cells that connect the most with astrocytes (since SR101 needs to be uptaken by them) and this may be a different subpopulation of the wanted “connected GB cells”. What about the GB cells that are connected among tumor cells but far from astrocytes? Which is the percentage of them? Astrocyte distribution strongly varies in different mouse models, so this may be a confounding mechanism that needs to be taken in account.
2. In normal vasculature, SR101 needs to cross the endothelial BBB to reach astrocyte. This means that also endothelial cells are “labeled” by SR101 (<https://link.springer.com/article/10.1007/s00429-013-0645-0>). Then, potentially, the SR101 labeling may also isolate vessel co-opting GB cells, that is potentially a different mechanism, as previously presented by the authors (<https://www.nature.com/articles/s41467-021-21117-3>). How is this reconciled with the technical method proposed by the authors?
3. The 3 cell lines appear to have differential percentage of SR101high cells (maybe not, but this quantitative information is missing), is this related to their mutational and transcriptomic landscape as suggested later in the clinical database analyses?
4. I find the validation of the SR101 method with Caprola elegant in principle. Anyway, it is still correlative. Is it possible to show SR101+ and Caprola+ GB cells in the very same image/experiment?

5. How transcriptionally and functionally stable is this connectivity “cell state”?
6. The mechanism shown can be theoretically achieved in the “connected GB cells” by tumor microtubules (TM) or tumor nanotubes (TNT). Can you clearly say which is relevant here?

Minor issues:

1. The authors may analyze the distribution of the “connectivity” geneset within the recently published public harmonized database (<https://www.biorxiv.org/content/10.1101/2022.08.27.505439v1>). This may further illustrate the features of the “connected” GB cells in much larger database.
2. A clear description of the mutational and transcriptomic landscape of the patient-derived cell lines used should be provided. Are they PN or MES? Which type of mutations are present?
3. A kinase enrichment analysis may be interesting additional information to be extracted from the proteome/phosphoproteome dataset to understand possible additional targets and signaling for CHI3L1 OE.
4. Figure 4h and 6k: statistical significance is missing

Reviewer #2 (Remarks to the Author): Expert in glioblastoma genomics and scRNA-seq / snRNA-seq

Hai et al. identify a connectivity metric to score cells from adult glioblastomas using scRNA-seq. They show that their score correlates with the mesenchymal glioma phenotype (as exemplified by CHI3L1) and inferior survival. The authors seek to develop this score as a biomarker for clinical trials. The study is interesting and contributes a significant advance to our understanding of glioma connectivity networks. The manuscript is largely complete, including: rigorous single-cell analysis of patient-derived models and relative controls, rigorous survival analysis demonstrating significant differences beyond known covariates, as well as some validation activities in situ and in vitro and via correlations with public data. However, there are a few moderate concerns:

1. The authors state in their abstract that the connectivity signature provides a “robust biomarker” for future clinical trials. However, as it stands the biomarker is the output of a single-cell assay taken from resected tumor tissue, typically of the primary disease. Thus, it couldn’t serve as a biomarker for any adjuvant therapy at diagnosis. Moreover, the authors have not shown that their connectivity score predicts features of recurrence. There are several recent single-cell and bulk RNA-seq longitudinal studies which can be used for this analysis. Otherwise, the authors should clarify this point.
2. CHI3L1 has been described as a marker of GBM cells with a mesenchymal phenotype. Can the authors clarify if the mesenchymal phenotype is a consequence of TM-connectivity?
3. There is limited overlap between the PDX-derived connectivity signature genes and the in vitro models.

4. The mesenchymal signature can be induced via AP1 signaling due to cellular stress. Have the authors considered that dye uptake may induce a stress response and drive mesenchymal transition? Along these lines is there preferential dye uptake in certain cell types?

Reviewer #3 (Remarks to the Author): Expert in glioblastoma genomics and scRNA-seq / snRNA-seq

The manuscript represents a substantial body of work aiming to understand the mechanisms underlying treatment resistant in glioblastoma tumors, with focus on resistance related to the development of tumor microtubes (TMs). The investigations are based on pioneer work by the authors who identified functional, multicellular network microtube structures in astrocytomas, allowing cells to communicate via Ca²⁺ transients, linked to treatment resistance. Therefore, targeting of microtubes has emerged as a possible new therapeutic approach.

The authors establish a gene expression signature of this connectivity by performing a series of logically laid out experiments using animal models, cell lines and patient samples, as well as by utilizing large publicly available datasets. The results and methods are described at a high level of detail, the methodologies used are sound and conclusions supported by the experimental data.

The results will lead to improvements in the molecular understanding of the Glioblastoma (GB) TM network and identify prognostic biomarkers, as well as possible therapeutic candidates. The identification of CHI3L1 as a single, GB specific marker for increased connectivity further underlines the significance of this work for GB research. The work is highly significant for the field and of interest to a wider audience.

Minor Points:

1. As a first step the authors aimed to develop a connectivity signature for GB. In order to establish a suitable experimental system the authors used tGFP overexpressing patient derived glioblastoma cells (PDGCs) xenografts, allowed to develop TM networks and separated according to high or low SR101 intensities, indicative of high or low TM mediated interconnection. scRNA-seq and bulk RNA-seq was performed to identify DE genes between the two categories. A schematic representation of the experimental scheme is shown in Figure 1. The scheme seems muddled in point 6 whereby both scRNA-seq and RNA-seq experiments seem to go through 10x GEMs. In fact, separate methods were used, the RNA-seq step utilizing SMARTER low input reagents. A modification of the figure is required to make this part clear.

2. The way the two bulk and scRNA-seq approaches were used for the development of the signature needs to be shown a bit better, in particular the overlap between the DE genes in both experiments and their respective direction of change. The reviewer feels that showing only the 13 gene overlap between the genes in the 71 and 245 gene signatures derived from the scRNA-seq and bulk RNA-seq data is fine, but are there DE genes found in bulk for which one or two scRNA-seq datasets were in agreement? It is possible that the criteria used (as shown in Fig1e) were too stringent, therefore a more comprehensive Venn diagram (extended version of Fig1e) to include the bulk DE genes would be informative. This can be shown as a supplementary figure.

3. Since the scRNA-seq derived signature was selected as “connectivity signature”, based on its performance in distinguishing the SR101 high from the SR101 low group, it makes sense to move Fig1g to supplementary and thus allow the rest of the figures in Figure 1 to gain space and easier to follow by indicating in Fig1f that in the 71 gene signature, 40 genes are up and 31 are down. Since this is mentioned in the text that follows (page 9) and in Fig2 it will make reading the manuscript easier.

4. The web app for connectivity was not accessible, requiring a password.

5. As a next step the authors extended their work using Caprola6 to identify a calcium signature based on sorting cells cells according to labelling intensities followed by RNA-seq. A positive correlation was found between the calcium and connectivity signatures, but only six genes overlapped. The authors should comment on this, is the correlation driven only by these genes?

6. The connectivity signature is tested on 21 clinical samples which were used to perform snRNA-seq in order to identify cell types and cell states in each tumor, and assess if application of the signature will identify tumors with higher connectivity scores. Results were validated by IHC. Since the authors demonstrate that the 71 gene signature can indeed identify tumors with high connectivity, they should define the sensitivity of high connectivity detection from bulk RNA-seq data depending on tumor cell content and presence of cell states associated with high connectivity (AC and MES1). This does not seem clear from the results presented in this section, but would be useful in order to mine bulk RNA-seq datasets. This applies also to the next step of the work utilizing TCGA and CGGA data and the sampling area association (lower in infiltration, higher in core areas).

7. The authors analyzed TCGA and CGGA data to assess the predictive value of the signature in terms of disease outcome, as well as identify gene mutations associated with the connectivity score. Indeed NF1, PTEN and TP53 were identified as associated. Would these mutations predict connectivity in any way? Or can they be used in combination with the expression signature?

8. The authors show that the CHI3L1 gene is a good marker for TM network connectivity, specifically expressed in GBs and perform a series of functional experiments involving both blocking using antibodies and overexpression, showing that it is driver of network formation. Proteomic and phosphoproteomic analysis of overexpressing cells identified differentially expressed and phosphorylated proteins, identifying GAP43, which is associated with filament stabilizations. However, an expression analysis would also indicate whether the overexpression of CHI3L1 drives the connectivity expression signature and to what extent. Such information would be very useful, considering the thoroughness of the work far and it is not clear why this was not carried out (or this reviewer missed it).

9. Overall the main figures tend to be overloaded and difficult to read in an A4 format. Some sub figures can be moved to supplementary. For example Figure 5h is a main figure on its own and hard to read in the current format.

Reviewer #4 (Remarks to the Author): Expert in MS-based proteomics and phosphoproteomics, and cancer proteogenomics

Reviewer Comments:

In this present manuscript, Hai et al. established a gene expression signature of tumor network connectivity utilizing single-cell RNA-sequencing xenografted primary glioblastoma cells, that could serve to determine the interconnectivity in individual tumors. Indeed, it is quite interesting work, I have my specific comments as below:

1. The genetic background of the PDGCL should be clarified, otherwise we cannot determine whether the genomic alterations of the tumors impact the connection.
2. The FACS data should be presented, how did the author obtain the SR101 high or low cells, the cell purity should be clarified.
3. Did the author perform any biological or technical repeats?
4. Missing value handling needs to be carefully described and justified.
5. The gene connective signatures were inferred based on transcriptomic data. However, since the proteins are the final executors, could the author perform further analysis and illustrate how the connective signatures expressed at proteomic level?
6. How many legends and receptors were defined as connective signatures?
7. P-values seem to be unadjusted. It is required to adjust them for multiple hypothesis testing.
8. The author performed proteomic and phosphoproteomic analysis to investigate the pathways that involved in CH13L1 driven connectivity. However, the exact mechanism is still unclear. Further functional experiments should be conducted.
9. The web app link was not accessible to me.

Revision of the manuscript “A clinically applicable connectivity signature for glioblastoma includes the tumor network driver CHI3L1” by Hai, Hoffmann et al.

- Point-to-point reply to reviewer’s comments -

We have indicated newly added data in green in all main, extended and supplementary files as well as the point-to-point responses.

Reviewer’s Comments:

Reviewer #1 (Remarks to the Author)

The manuscript by Hai et al. describes a novel “connectivity signature” for GB and the role of CHI3L1 in the formation of tumor microtubes.

The authors use a novel dye uptake methodology, scRNA-Seq and orthotopically implanted patient-derived GB to isolate the highly connected GB cells within the tumor to then investigate the geneset signature of this novel cell state (“located” between AC and MES-like cell states). Using data mining the authors demonstrate that this gene signature is correlated with poor patient survival. Moreover, overexpression of CHI3L1 in GB cells allowed them to mechanistically demonstrate targetability of the connectivity GB cell state.

The clinical and pre-clinical relevance of tumor microtubes is suggested by a rich recent literature on this topic, but still a solid geneset was missing. The abstract and method part are well written and the results are highly interesting.

Overall, the experiments were really well performed and the data support the conclusions the authors made. There are, however, a small number of issues that need to be addressed or clarified.

Reply: We are grateful for this reviewer's assessments. We now in detail outline new experiments and insights in the point-to-point responses below. Additionally, we have indicated newly added data in green in all main, extended and supplementary files.

Major issues:

1. Although I think that the SR101 method to isolate highly connected GB cells is smart, I must confess that there is a potential issue in it. With it, the authors are enriching for the GB cells that connect the most with astrocytes (since SR101 needs to be uptaken by them) and this may be a different subpopulation of the wanted "connected GB cells". What about the GB cells that are connected among tumor cells but far from astrocytes? Which is the percentage of them? Astrocyte distribution strongly varies in different mouse models, so this may be a confounding mechanism that needs to be taken in account.

Reply: Our group has recently shown that patient-derived glioblastoma cells (PDGCs) are cytoplasmatically connected with astrocytes via tumor microtubes (TMs)¹. To delineate the contribution of astrocytes to PDGC intracellular SR101 levels, we correlated SR101 intensities of individual PDGCs, PDGC network parameters and astrocyte distribution.

Precisely, we defined brain areas with different PDGCs densities (new Extended Data Fig. 1c,f) and found that SR101 intensities (new Extended Data Fig. 1d) positively correlate with both PDGC density (new Extended Data Fig. 1f) and the extent of inter-PDGC TM-connections (new Extended Data Fig. 1g).

In contrast, astrocytes were found to be homogeneously distributed between the different brain areas (new Extended Data Fig. 1c,e). In fact, we did not discover PDGCs far from astrocytes.

Based on this data, we conclude that

1) PDGC density, but not astrocyte density, is a surrogate marker for morphological and molecular TM-connectivity – as previously shown *in vitro* (Fig. 3n-p), *in vivo* ⁽²⁾ and in patient specimens (Fig. 5d, ³).

2) The extent of TM-connections between PDGCs plays a pivotal role for intracellular SR101 staining intensities.

Thus, we conclude that the SR101 approach is suitable for quantifying PDGC-PDGC TM-connections.

2. In normal vasculature, SR101 needs to cross the endothelial BBB to reach astrocyte. This means that also endothelial cells are “labeled” by SR101 (<https://link.springer.com/article/10.1007/s00429-013-0645-0>). Then, potentially, the SR101 labeling may also isolate vessel co-opting GB cells, that is potentially a different mechanism, as previously presented by the authors (<https://www.nature.com/articles/s41467-021-21117-3>). How is this reconciled with the technical method proposed by the authors?

Reply: We thank the reviewer for inquiring about the spatiotemporal distribution of SR101 and the implication of vessel co-opting PDGCs. As the reviewer points out, PDGCs obtained from our xenograft model were sorted and sequenced irrespective of their spatial distribution (new Extended Data Fig. 1j-l). However, only a minority (mean of 4%) of all PDGCs is spatially associated with blood vessels in the xenograft model (new Extended Data Fig. 1h-i), suggesting a minor role of this population. Of note, the xenograft model is representative of GB patient specimens⁶.

As previously shown and raised by the reviewer, transcriptomic fingerprints of vessel-coopting GCs differ from parenchymal PDGCs^{4,5}. Specifically, Jung et al.⁶ showed that NOTCH1 pathway genes are upregulated in perivascular PDGCs but downregulated in parenchymal cells. In agreement with our approach not having enriched vessel co-opting GB cells, NOTCH1 pathway

genes DLL1, DLL3 and HES6 were all downregulated in the SR101 scRNA-Seq and RNA-Seq datasets (Fig. 1f, g). Taken together these facts, the effect of the rare perivascular cell population on the developed connectivity signature is neglectable.

To incorporate the reviewer's excellent comment, we now discuss the contribution of rare populations' transcriptomes (page 34, lines 9-13).

3. The 3 cell lines appear to have differential percentage of SR101^{high} cells (maybe not, but this quantitative information is missing), ...

Reply: As correctly pointed out by the reviewer the absolute number of QC-passed single-cell sequenced cells differed between PDGCLs (S24, T269, P3XX), and also SR101 groups (SR101^{high} and SR101^{low}, Supplementary table 2). The reasons for this were mainly technical, e.g. P3XX tumors were relatively smaller compared to S24 and T269 tumors and we therefore retrieved a relatively lower amount of P3XX PDGCs for scRNA-Seq.

Most importantly, we would like to clarify that the absolute numbers of scRNA sequenced SR101^{high}/SR101^{low} PDGCs differ from the numbers of FACS-analyzed SR101^{high}/SR101^{low} PDGCs and are therefore not indicative for the respective tumors' overall connectivity.

To assess and compare the overall connectivity of individual PDGCL tumors, which we did similarly in the "clinical database" analysis, we now analyzed mean SR101 fluorescence intensities and connectivity signature scores from the fast *in vivo* growing S24 and P3XX PDGCLs. SR101 intensities and connectivity signature scores were both elevated in the S24 compared to the P3XX PDGCL (Reviewer Fig. 1). These findings imply that intracellular SR101 levels might be a suitable surrogate marker for inter-tumoral connectivity *in vivo*. However, further validation is warranted, given the small number of PDGCLs used in this experiment.

Reviewer Fig. 1: Relative SR101 intensities of xenografted PDGCLs correlate with connectivity signature scores. **a**, Histogram showing SR101 intensities of single, live, tGFP^{high} S24 (left) and P3XX (right) PDGCs. Backgating was used for visualization of gating strategies. **b-c**, Comparison of SR101 intensities between S24 and P3XX PDGCs. n = 2526 (P3XX) and 1696 (S24) PDGCs. **b**, Histogram. **c**, Violin plot. Two sided t-test. **d**, Connectivity signature scores of xenografted S24 and P3XX PDGCs. d, Scores were scaled, centered and winsorized at -3 and 3 across cells.****, p < 0.0001.

... is this related to their mutational and transcriptomic landscape as suggested later in the clinical database analyses?

Reply: We believe that the given number of xenografted PDGCLs included in the SR101 experiment (n=3) hampers a direct conclusion about the impact of individual mutations and transcriptome subtypes for statistical reasons. Nonetheless, we clarified the genetic and

transcriptomic landscape of all PDGCLs (new Supplementary Table 1, new Extend Data Fig. 1a, new Extend Data Fig. 1b). Transcriptome-wise, P3XX PDGCL is characterized by a CL/MES2 (TCGA subtype/cell state) profile and S24 by a MES/AC (TCGA subtype/cell state) profile. These data collaboratively add to the reported correlation of TCGA expression subtype, cell state, connectivity signature scores (Fig. 4, Fig. 5a,b) and directly link them with SR101 levels. We have now added to the discussion that further investigations are necessary to shed light on the relative impact of individual mutations (page 35, line 1-5).

4. I find the validation of the SR101 method with Caprola elegant in principle. Anyway, it is still correlative. Is it possible to show SR101+ and Caprola+ GB cells in the very same image/experiment?

Reply: We agree with the reviewer that such a hybrid experiment would be very interesting and extremely insightful. However, there are some technical issues that hamper such an experiment. On the one hand, the mechanism by which SR101 is preferentially accumulated in highly connected PDGCs *in vivo* is fascinatingly complex and requires the presence of a blood-brain barrier to prevent unspecific penetration of the brain, vessels for dye transport and astrocytes that connect to PDCCs - just to name some of the critical pre-requirements. *In vitro* experiments just cannot account for this. On the other hand, Caprola is not yet setup in rodents and the optimization for such a system, although currently in preparation⁷, is extremely cumbersome and involves the determination and optimization of, amongst others, administration route, toxicity management and labeling kinetics. Therefore, although desirable it is unfortunately not yet possible to combine both methods in a useful manner.

5. How transcriptionally and functionally stable is this connectivity “cell state”?

Reply: We agree with the reviewer that it is of interest to investigate the attributes of GB TM networks longitudinally. We therefore analyzed the transcriptional stability of the connectivity signature state in temporal patient pairs. Specifically, we harnessed the GLASS cohort, in which GB patient specimen were obtained from up to three re-surgeries⁸, and found connectivity signature scores to be retained over time (new Extended Data Fig. 8a). This interesting finding needs to be further discussed, e.g. if this analysis misses transient treatment effects on the TM network because of a potential time gap between treatment end and relapsed tumor sampling. An ideal setting to investigate drug-related effects on TM-networks would sample during or directly after therapy. However, this was not given in the GLASS cohort. Of note, the upcoming Persurge trial investigating the effect of a TM-inhibitor was designed to include a more TM-connectivity tailored readout scheme (see Reviewer 2 question 1).

We now investigated the functional stability of the connectivity cell state. Connectivity signature scores in both primary and relapse situations were prognostic for the overall survival and frequency of relapses (new Fig. 5f-h, new Extended Data Fig. 8g,h).

Taken together, these findings point towards a high transcriptional and functional stability of the connectivity “cell state”.

6. The mechanism shown can be theoretically achieved in the “connected GB cells” by tumor microtubes (TM) or tumor nanotubes (TNT). Can you clearly say which is relevant here?

Reply: Multicellular tumor networks in GB are driven by two different types of long intercellular cell-to-cell connections: TMs and thinner and shorter lived tunneling tumor nanotubes (TNTs,⁹). In this manuscript, we solely aim at tumor microtubes (TM,^{10,11}). We have not elaborated on a correlation of morphological and functional features of TNTs with *in vivo* SR101 intensities,

Caprola labeling intensities and CHI3L1 levels - just to name some of the performed experiments for determination of the molecular underpinnings of GB TM-networks.

However, we correlated the connectivity signature with all known molecular markers⁹ of TMs (i.e. *CTNND1*, *TTYH1*, *GAP43*, *THBS1*, *TGFBI*, *SMAD3*) and TNTs (i.e. *CDC42*, *RAC1*, *CSPG4*, *LST1*, *TNFAIP2*, *RASSF1*, *S100A4*, *PIK3CA*, *MTOR*, *KRAS*, *TP53*). We found a good overlap with TM markers but not with TNT marker genes (Reviewer Fig. 2).

Reviewer Fig. 2: The connectivity signature score is a good surrogate marker for TMs. SR101 xenograft scRNA-Seq dataset. n = 35,822 PDGCs. **a**, Density plots showing the distribution of scores based on known TM or TNT markers gene sets⁹ in SR101^{high} and SR101^{low} groups. Left, TM marker gene set score. Right, TNT marker gene set score. **b**, Scatter plot showing correlation of the connectivity signature score with TM or TNT scores. Left, TM marker gene set score. Right, TNT marker gene set score. Pearson correlation test. **a,b**, Values were Z-score scaled and centered across PDGCs and winsorized to -3 and 3.

Together with our in-depth wet-lab correlation of the connectivity signature with TMs, this implies that the connectivity signature is suitable as a surrogate marker of TM-connected PDGCs.

We have clarified in the manuscript that we have not correlated the connectivity signature with features of TNTs (page 4 line 17, page 33 lines 2-6, page 34, line 20-22).

Minor issues:

1. The authors may analyze the distribution of the “connectivity” geneset within the recently published public harmonized database (<https://www.biorxiv.org/content/10.1101/2022.08.27.505439v1>). This may further illustrate the features of the “connected” GB cells in much larger database.

Reply: We thank the reviewer for this suggestion. We have now included a comprehensive map of connectivity signature scores in different populations of the healthy brain and compared them to different GB cell populations (new Extended Data Fig. 8c-e). The results in GCs were comparable to the analysis in our patient cohort and illustrate heterogeneity of the score based on cell subtype composition in different patient samples (new Extended Data Fig. 6i-n).

2. A clear description of the mutational and transcriptomic landscape of the patient-derived cell lines used should be provided. Are they PN or MES? Which type of mutations are present?

Reply: More detailed information about methylation subtype (new Supplementary Table 1), copy number profile (new Extend Data Fig. 1a), mutations (new Extend Data Fig. 1b), and transcriptional subtypes (new Supplementary Table 1) were added.

3. A kinase enrichment analysis may be interesting additional information to be extracted from the proteome/phosphoproteome dataset to understand possible additional targets and signaling for CHI3L1 OE.

Reply: We are grateful for the reviewer’s suggestion. We now predicted kinases putatively responsible for the identified DPPs. MEK/ERK and AKT signaling pathways were predicted to be upregulated in CHI3L1 OE PDGCs (Extended Data Fig. 10c). This was in line with previous

reports: PDGC-stimulation with CHI3L1 resulted in elevated activation of these pathways¹², whereas inhibition of CHI3L1 with a small molecule drug conferred the opposite effect¹³. Moreover, a phosphoproteomic characterization of neuronal growth cones independently identified elevated ERK activation¹⁴.

4. Figure 4h and 6k: statistical significance is missing

Reply: Statistical tests were performed and significance is now indicated in Fig. 4h and Fig. 6k.

Reviewer #2 (Remarks to the Author)

Hai et al. identify a connectivity metric to score cells from adult glioblastomas using scRNA-seq. They show that their score correlates with the mesenchymal glioma phenotype (as exemplified by CHI3L1) and inferior survival. The authors seek to develop this score as a biomarker for clinical trials. The study is interesting and contributes a significant advance to our understanding of glioma connectivity networks. The manuscript is largely complete, including: rigorous single-cell analysis of patient-derived models and relative controls, rigorous survival analysis demonstrating significant differences beyond known covariates, as well as some validation activities in situ and in vitro and via correlations with public data. However, there are a few moderate concerns:

We thank the reviewer for the overall positive feedback and valuable suggestions. We now addressed these few concerns and discussed into more detail in the point-to-point responses below. Additionally, we indicated newly added data in green in all main, extended and supplementary files.

1. The authors state in their abstract that the connectivity signature provides a “robust biomarker” for future clinical trials. However, as it stands the biomarker is the output of a single-cell assay taken from resected tumor tissue, typically of the primary disease. Thus, it couldn’t serve as a biomarker for any adjuvant therapy at diagnosis.

Reply: In the present manuscript we provide evidence for the connectivity signature as a prognostic biomarker. The usefulness as a predictive biomarker has not been analyzed here.

Harnessing three independent cohorts, we showed that the connectivity signature, determined from RNA-Seq of primary GB, is prognostic for survival and surgical interval. These findings were consistent in both univariate (new Fig. 5f-h) and multivariate analyses (new Fig. 5f-h). Importantly, the prognostic impact was retained in the recurrent setting (new Extended Data Fig. 8g,h)

We have further shown that morphological and functional disconnection of PDGC networks using BTP-2 was accompanied with a reduction of the connectivity signature score (Fig. 3i-m), whereas stimulation of GC networks by CHI3L1 OE led to increased connectivity signature scores. These data point towards the connectivity signature’s suitability as a predictive marker for TM-disrupting agents.

Matching the reviewer’s advice, we also seek to corroborate the robustness of the connectivity signature as a predictive biomarker in clinical setups. We intend to do this in the scope of the upcoming Persurge trial⁹ enrolling GB patients with relapsed GB in a randomized window of opportunity design. Briefly, patients receive standard-of-care treatment after initial diagnosis and neoadjuvant therapy with a GB-network disrupting agent after relapse. scRNA-Seq derived connectivity signature scores are determined from both primary and relapse tumor specimens. This procedure allows determination of the drug’s efficacy (I), if patients with higher connectivity scores in the primary setting are likely to benefit more from the GB-network disrupting agent (II) and if network disruption influences overall survival (III).

All in all, we have now clearly stated that the connectivity signature is a prognostic biomarker (page 34, line 2).

Moreover, the authors have not shown that their connectivity score predicts features of recurrence. There are several recent single-cell and bulk RNA-seq longitudinal studies which can be used for this analysis. Otherwise, the authors should clarify this point.

Reply: We acknowledge that further investigation of how tumor networks develop over time would add valuable insights into brain tumor biology. This was now added. Briefly, connectivity signature scores determined from GB specimens longitudinally collected from the same patients remained at a similar level (new Extended Data Fig. 8a). This interesting finding needs to be further discussed with respect to confounding factors, such as a potentially treatment-free interval prior to re-surgery. An ideal setting to investigate drug-related effects on TM-networks would sample during or directly after therapy. However, this was not given in the GLASS cohort, potentially biasing the discoveries (new Extended Data Fig. 8a) towards undetected transient drug effects.

We also investigated the functional stability of the connectivity cell state. Connectivity signature scores in both primary and relapse situations were prognostic for the overall survival and frequency of relapses (new Fig. 5f-h, new Extended Data Fig. 8g,h).

Taken together, these findings point towards a high transcriptional and functional stability of the connectivity “cell state”.

2. CHI3L1 has been described as a marker of GBM cells with a mesenchymal phenotype. Can the authors clarify if the mesenchymal phenotype is a consequence of TM-connectivity?

Reply: We thank the reviewer for raising this important point. We show that the direct association between CHI3L1 and TM is independent from the MES phenotype:

First, CHI3L1 associates with the mesenchymal TCGA subtype¹⁵ and MES1 cell state¹⁶, but within the MES1 PDGCs, we found that CHI3L1 has a much higher expression in SR101^{high} compared to SR101^{low} cells, which argues for a mechanism independent from the MES1 cell state (Fig. 6h). Furthermore, the association between CHI3L1 and SR101^{high} tumor cells is also valid in all other non-mesenchymal tumor cell states (Fig. 6h).

Second, Fig. 7e-f shows IF images that demonstrate the association of CHI3L1 with TMs in Ctrl and CHI3L1 OE PDGCs. Staining of CHI3L1 on selected paraffin embedded patient tissues identified not only a remarkably higher CHI3L1 histoscore in samples with high TM-connectivity and high connectivity signature score (Fig. 6j-k), but we also show that there are tumors that display low connectivity and low connectivity signature score (Fig. 6j-k lower panel middle tumor, compare with Fig. 4i), together with low CHI3L1 staining and display a mesenchymal expression subtype. This further points towards CHI3L1 as a new marker for TM connectivity, independent of the mesenchymal TCGA subtype¹⁵

Third, to investigate if CHI3L1 is not only a robust, mesenchymal-independent marker of GB connectivity but also able to directly influence TM networks, we performed functional experiments. Blocking of CHI3L1 activity with an antibody decreased the TM-length of PDGCs *in vitro* (Fig. 7a-b). Conversely, we have established overexpression (OE) of CHI3L1 in PDGCLs (Supplementary Fig. 9c-d) and found that OE of CHI3L1 increases the number of TMs in an *in vitro* model (Fig. 7c-d) as well as in an *in vivo* patient derived xenograft model (Fig. 7g-l). Substantiating the correlation of TM networks with the connectivity signature, OE of CHI3L1 led to an increase in the connectivity signature score based on RNA-Seq and proteomic data – after exclusion of CHI3L1 from the signature, as this is artificially overexpressed in the CHI3L1 overexpressing cell lines (new Fig. 7o). Furthermore, MES1 cell state genes were increased upon

CHI3L1 OE (new Extended Data Fig. 9h,i). This functional data strongly argues for a direct influence of CHI3L1 on TM-connectivity beyond being a pure mesenchymal marker.

3. There is limited overlap between the PDX-derived connectivity signature genes and the in vitro models.

Reply: We thank the reviewer for raising this important point. The observed heterogeneity between the gene sets is a result of different factors.

First, we used two different RNA-sequencing workflows (single cell and bulk). By way of example, transcriptomic fingerprinting of the very same SR101 populations (SR101^{high} and SR101^{low}) yielded different types and numbers of DEGs: 245 (RNA-Seq, Extended Data Fig. 2a) and 71 (scRNA-Seq, Fig. 1f), respectively, with 13 overlapping DEGs (18% of the 71 scRNA-Seq derived DEGs, Fig. 1g). Bulk RNA-Seq was able to pick up a higher number of differentially regulated genes than 10x Genomics-based scRNA-Seq. This is caused by a lower sequencing depth and, in line with the literature, resulted in a higher dropout rate in scRNA-Seq. Conversely, some genes were differentially regulated in scRNA-Seq but not in bulk RNA-Seq, potentially better reflecting heterogeneously expressed genes (Extended Data Fig. 2b).

Second, the cutoffs used for the scRNA-Seq were very strict and set to include high-fold changes in two PDGLCs or medium fold changes in all of the three PDGCLs (Fig. 1e). DEGs from bulk RNA-Seq did not require such a high fold-change and were only derived from S24 and T269 PDGCLs, as no cells could be obtained from the P3XX PDGCL due to a smaller tumor size. Thus, lowering the cutoffs for scRNA-Seq would have resulted in a higher number of overlapping genes between RNA-Seq and scRNA-Seq as demonstrated in new Extended Data Fig. 2c.

Third, the employed assays most likely have different sensitivities for capturing distinct TM-network features. SR101 intensities in the xenograft model reflect a PDGC's dye-uptake ability via

TMs (Fig. 1c,d), whereas the Caprola₆ model, best captures TM-network activity based on Ca²⁺ levels (Fig. 3b,c).

We clarified these aspects in the result and discussion sections (page 34, line 15-16).

Importantly, there was a high integrability of the connectivity signature with both SR101 xenograft bulk-RNA DEGs and the Calcium signature in the scRNA-Seq (R=0.87, Fig. 1i and R=0.68, Fig. 3e), TCGA (R=0.89; Fig. 1j) and snRNA-Seq (R=0.55; Fig 6h) datasets. This highlights the robustness of the connectivity signature.

4. The mesenchymal signature can be induced via AP1 signaling due to cellular stress...

Reply: Our data suggest that not only the MS TCGA expression subtype and MES1 cell state but also the AC cell state is preferentially enriched in highly connected cells (Fig 1f, Fig 4g), demonstrating a certain independence of the connectivity signature from the mesenchymal signature.

... Have the authors considered that dye uptake may induce a stress response...

Reply: As the reviewer points out, it was shown that hypoxia, environmental and genetic stress factors are related to increased mesenchymal-like cell states^{16,19,20}. According to previous research, primarily the MES2 signature correlates with stress response¹⁶. However, we did not find an association of the MES2 cell state and high connectivity signature scores in any of the dye-utilizing model-derived datasets.

SR101 xenograft model: SR101^{high} PDGCs had slightly less MES2-like cells compared to SR101^{low} PDGCs (Fig. 2e) and the velocity of SR101^{high} PDGCs was found to be preferentially towards an AC-like cell state but not towards the MES2 cell state (Fig. 2i-j, Extended Data Fig. 3e-f).

Caprola models: Relative to both medium and low calcium signature score groups, MES2 was not increased in the high calcium signature score group of the Caprola experiments (Fig. 3g).

...and drive mesenchymal transition?...

Reply: Arguing against a dye-triggered mesenchymal transition (indicated by molecular and phenotypical features) we found increased AC and MES1 cell state scores (Extended Data Fig. 9h,i), elevated connectivity signature scores (new Fig. 7o) and fostered TM-networks (Fig. 7g-l) in the dye-free CHI3L1 OE model. Likewise, altered molecular and phenotypical TM-network connectivity was dependent on pharmacological intervention (Fig. 3i-m), GB density (Fig. 3n-p) and cultivation method (Extended Data Fig. 5o-q), but not dye addition.

... Along these lines is there preferential dye uptake in certain cell types?

Reply: We showed that SR101 and CPY/CA dye levels positively correlated with the degree of morphological (SR101 xenograft model; Fig. 1c,d) and functional (Caprola₆ model; Fig. 3d) network connectivity, respectively. PDGCs with high dye uptake levels best matched the AC and MES1 cell states (Fig. 2e, Fig. 3g). To investigate if PDGCs of the AC and MES1 cell states have higher dye uptake rates *per se*, we harnessed Caprola_{on}. Its CPY-CA dye integration was rendered independent of calcium levels – in contrast to Caprola₆. We next FACS sorted and RNA sequenced Caprola_{on}-overexpressing PDGCs with different labeling intensities. Identified DEGs between PDGC groups with high, medium and low labeling intensities (Extended Data Fig. 5c) were further denoted as Caprola_{on} signature. Not the Caprola_{on} (Reviewer Fig. 3a) signature, but the Caprola₆ signature co-localized with the MES1/AC cell states (Reviewer Fig. 3b).

Moreover, the pooled GCs in AC/MES1/MES2 cell states, denoted as AC/MES, even had slightly lower Caprola_{on} signatures than the pooled GCs in OPC/NPC1/NPC2 cell states, denoted as OPC/NPC (Reviewer Fig. 3c), which contrasts our finding that AC/MES GCs have higher Caprola₆ signature scores than OPC/NPC (Reviewer Fig. 3d) GCs.

Of note, GCs in MES1 and MES2 cell states had lower Caprola_{on} signature scores compared to all other non-cycling cell states and GCs in AC cell states had comparable Caprola_{on} signature scores

to GCs in OPC and NPC cell states (Reviewer Fig. 3e), whereas GCs in AC and MES cell states clearly reached higher scores of the Ca²⁺-activity based Caprola₆ signature (Reviewer Fig. 3f).

Taken together, these facts underline that GCs in AC and especially MES1 states are likely to have higher connectivity signature scores because of a higher TM-degree and not because of higher dye uptake and related higher cellular stress. We have mentioned this in the discussion (page 34, lines 3-7)

Reviewer Fig. 3: Correlation of cell states with Caprola₆ and Caprola_{on} signatures in our 21 patient snRNA-Seq dataset (n = 213,444 GCs). **a-b**, Two-dimensional representation of patient GCs according to cell states and signature scores. **a**, Caprola_{on} signature. **b**, Caprola₆ signature. **c-d**, Density plot of signature scores in AC/MES1/MES2 and NPC1/OPC1/OPC2 GC groups. Dotted lines depict medians. **c**, Caprola_{on} signature. **d**, Caprola₆ signature. **e-f**, Signature scores in each cell state. **e**, Caprola_{on} signature. **f**, Caprola₆ signature. **a-f**, Signature scores were Z-score scaled and centered across cells and winsorized to -3 and 3.

Reviewer #3 (Remarks to the Author)

The manuscript represents a substantial body of work aiming to understand the mechanisms underlying treatment resistant in glioblastoma tumors, with focus on resistance related to the development of tumor microtubes (TMs). The investigations are based on pioneer work by the authors who identified functional, multicellular network microtube structures in astrocytomas, allowing cells to communicate via Ca²⁺ transients, linked to treatment resistance. Therefore, targeting of microtubes has emerged as a possible new therapeutic approach.

The authors establish a gene expression signature of this connectivity by performing a series of logically laid out experiments using animal models, cell lines and patient samples, as well as by utilizing large publicly available datasets. The results and methods are described at a high level of detail, the methodologies used are sound and conclusions supported by the experimental data.

The results will lead to improvements in the molecular understanding of the Glioblastoma (GB) TM network and identify prognostic biomarkers, as well as possible therapeutic candidates. The identification of CHI3L1 as a single, GB specific marker for increased connectivity further underlines the significance of this work for GB research. The work is highly significant for the field and of interest to a wider audience.

Reply: We thank the reviewer for the positive evaluation of our work and careful review, have addressed the concerns and discussed point by point below and additionally indicated newly added data in green in all main, extended and supplementary files:

Minor Points:

1. As a first step the authors aimed to develop a connectivity signature for GB. In order to establish a suitable experimental system the authors used tGFP overexpressing patient derived glioblastoma cells (PDGCs) xenografts, allowed to develop TM networks and separated according to high or low SR101 intensities, indicative of high or low TM mediated interconnection. scRNA-seq and bulk RNA-seq was performed to identify DE genes between the two categories. A schematic representation of the experimental scheme is shown in Figure 1. The scheme seems muddled in point 6 whereby both scRNA-seq and RNA-seq experiments seem to go through 10x GEMs. In fact, separate methods were used, the RNA-seq step utilizing SMARTER low input reagents. A modification of the figure is required to make this part clear.

Reply: We thank the reviewer for this comment and have now adjusted this panel accordingly (Fig. 1a).

2. The way the two bulk and scRNA-seq approaches were used for the development of the signature needs to be shown a bit better, in particular the overlap between the DE genes in both experiments and their respective direction of change. The reviewer feels that showing only the 13 gene overlap between the genes in the 71 and 245 gene signatures derived from the scRNA-seq and bulk RNA-seq data is fine, but are there DE genes found in bulk for which one or two scRNA-seq datasets were in agreement? It is possible that the criteria used (as shown in Fig1e) were too stringent, therefore a more comprehensive Venn diagram (extended version of Fig1e) to include the bulk DE genes would be informative. This can be shown as a supplementary figure.

Reply: We thank the reviewer for this valuable suggestion. We now added a Venn diagram to illustrate the overlap of the RNA-Seq derived DEGs with scRNA-Seq derived DEGs of the

individual PDGCLs (new Extended Data Fig. 2c). As a result of these loosened criteria the total number of overlapping DEGs between single cell and bulk RNA-Seq increased from 13 to 43.

However, our primary intention to choose a more stringent criterion for identifying the common DEGs between PDGCLs lied in the heterogenous genetic backgrounds (new Extended Data Fig. 1a-b and new Supplementary Table 1) of the three PDGCLs. We aimed to develop a more generalized connectivity signature from them. Therefore, we only included the DEGs with high-fold changes in two PDGLCs or medium fold changes in all of the three PDGCLs to avoid the bias to one specific PDGCL (Figure 1e).

3. Since the scRNA-seq derived signature was selected as “connectivity signature”, based on its performance in distinguishing the SR101 high form the SR101 low group, it makes sense to move Fig1g to supplementary and thus allow the rest of the figures in Figure 1 to gain space and easier to follow by indicating in Fig1f that in the 71 gene signature, 40 genes are up and 31 are down. Since this is mentioned in the text that follows (page 9) and in Fig2 it will make reading the manuscript easier.

Reply: We acknowledge the reviewer comment and have now shifted the respective figure to new Extended Data Fig. 2a.

4. The web app for connectivity was not accessible, requiring a password.

Reply: The web application is available with the following information:

URL: <https://connectivity-glioma.dkfz.de/>

username: reviewer

password: CQMYGYSDSS

The tool will be made publicly available after acceptance of the manuscript.

5. As a next step the authors extended their work using Caprola6 to identify a calcium signature based on sorting cells calls according to labelling intensities followed by RNA-seq. A positive correlation was found between the calcium and connectivity signatures, but only six genes overlapped. The authors should comment on this, is the correlation driven only by these genes?

Reply: We thank the reviewer for raising this important point. The observed heterogeneity between the gene sets is a result of different factors.

First, we used two different RNA-sequencing workflows (single cell and bulk). By way of example, transcriptomic fingerprinting of the very same SR101 populations (SR101^{high} and SR101^{low}) yielded different types and numbers of DEGs: 245 (RNA-Seq, Extended Data Fig. 2a) and 71 (scRNA-Seq, Fig. 1f), respectively, with 13 overlapping DEGs (18% of the 71 scRNA-Seq derived DEGs, Fig. 1g). Bulk RNA-Seq was able to pick up a higher number of differentially regulated genes than 10x Genomics-based scRNA-Seq. This is caused by a lower sequencing depth and, in line with the literature, resulted in a higher dropout rate in scRNA-Seq. Conversely, some genes were differentially regulated in scRNA-Seq but not in bulk RNA-Seq, potentially better reflecting heterogeneously expressed genes (Extended Data Fig. 2b).

Second, the cutoffs used for the scRNA-Seq were very strict and set to include high-fold changes in two PDGLCs or medium fold changes in all of the three PDGCLs (Fig. 1e). DEGs from bulk RNA-Seq did not require such a high fold-change and were only derived from S24 and T269 PDGCLs, as no cells could be obtained from the P3XX PDGCL due to a smaller tumor size. Thus, lowering the cutoffs for scRNA-Seq would have resulted in a higher number of overlapping genes between RNA-Seq and scRNA-Seq as demonstrated in new Extended Data Fig. 2c.

Third, the employed assays most likely have different sensitivities for capturing distinct TM-network features. SR101 intensities in the xenograft model reflect a PDGC's dye-uptake ability via

TMs (Fig. 1c,d), whereas the Caprola₆ model, best captures TM-network activity based on Ca²⁺ levels (Fig. 3b,c).

We clarified these aspects in the result and discussion sections (page 34, line 15-16).

Importantly, there was a high integrability of the connectivity signature with both SR101 xenograft bulk-RNA DEGs and the Calcium signature in the scRNA-Seq (R=0.87, Fig. 1i and R=0.68, Fig. 3e), TCGA (R=0.89; Fig. 1j) and snRNA-Seq (R=0.55; Fig 6h) datasets. This highlights the robustness of the connectivity signature.

6. The connectivity signature is tested on 21 clinical samples which were used to perform snRNA-seq in order to identify cell types and cell states in each tumor, and assess if application of the signature will identify tumors with higher connectivity scores. Results were validated by IHC. Since the authors demonstrate that the 71 gene signature can indeed identify tumors with high connectivity, they should define the sensitivity of high connectivity detection from bulk RNA-seq data depending on tumor cell content and presence of cell states associated with high connectivity (AC and MES1). This does not seem clear from the results presented in this section, but would be useful in order to mine bulk RNA-seq datasets. This applies also to the next step of the work utilizing TCGA and CGGA data and the sampling area association (lower in infiltration, higher in core areas).

Reply: We appreciate this very insightful and pivotal remark. To understand the interrelation of GB purity and connectivity signature scores derived from RNA-Seq, we first compared bulk GB and healthy brain tissue (new Extended Data Fig. 8c). Both connectivity signature scores and variance between specimens turned out to be higher in GB tumors (new Extended Data Fig. 8c). Connectivity signature scores in healthy brain populations varied, with astrocytes having the highest scores amongst all GB and healthy cell types (new Extended Data Fig. 8d-e). We next

assessed the tumor purity of the RNA sequenced TCGA samples based on IHC (new Extended Data Fig. 8b,²¹), as we regard this technique as the current gold-standard for purity estimation and further estimated tumor purity in our study based on it. The vast majority of GB specimens had a tumor content >75%. Connectivity signature scores and tumor content were not significantly correlated (new Extended Data Fig. 8b,²¹). All in all, these analyses show that connectivity signature scores were not greatly influenced by the tumor purity in datasets with high tumor content. We recommend to use samples with a tumor content of >75% for connectivity signature score grading from RNA-Seq data to prevent a bias.

To further investigate the relation of connectivity signature score and cell type composition in bulk RNA-Seq GB samples, we applied cell type deconvolution with the cell type signature matrix generated from our patient scRNA-Seq data using CIBERSORTx²². Similar to the TCGA IHC results, the total percentage of predicted nonmalignant cell types was lower than 25% in the vast majority of samples and therefore did not determine the level of connectivity signature score (Reviewer Fig. 4a). Malignant AC-like cells were the most abundant cell state in the GB specimen but did not influence connectivity signature scores (Reviewer Fig. 4b). We found connectivity signature scores to be associated with the percentage of the MES1 and much less abundant NPC1 cell states (Reviewer Fig. 4b). The latter findings highlight the association of a more connected phenotype with the MES1 cell state, as shown *in vivo* (Fig. 1c,d,; Fig. 2e, Fig. 7g-l,o). Lowly connected, invading PGGCs, in contrast, have been phenotypically associated with elevated NPC and OPC cell states^{1,2}.

Reviewer Fig. 4: Connectivity signature score and cell type composition in bulk RNA-Seq of 230 TCGA GB samples. a, Cell type composition in sample ordered from left to right according to their connectivity signature scores from low to high. The cell type deconvolution was performed based on the cell type signature matrix generated from our patient scRNA-Seq dataset using CIBERSORTx²². The cell type composition includes six non-malignant cell types (i.e., oligodendrocyte, astrocyte, T-cell, macrophage, endothelial cell and pericyte) and eight malignant cell states (i.e., AC, MES1, MES2, OPC, NPC1, NPC2, G1_S, and G2_M). **b,** Correlation between connectivity signature score and cell state percentage. Left, AC-like malignant cell state. Middle, MES1-like malignant cell state. Right, NPC1-like malignant cell state.

7. The authors analyzed TCGA and CGGA data to assess the predictive value of the signature in terms of disease outcome, as well as identify gene mutations associated with the connectivity score. Indeed NF1, PTEN and TP53 were identified as associated. Would these mutations predict connectivity in any way? Or can they be used in combination with the expression signature?

Reply: Mutations in these genes are indeed associated with higher or lower connectivity signature scores (Fig. 5c), but given the high variance (Fig. 5c), it is not sufficient to predict connectivity based on the occurrence of one of these mutations *per se*. We propose the connectivity signature obtained from scRNA-Seq, RNA-Seq or proteomics as the primary readout for TM-connectivity. The finding about the association of connectivity signature and certain gene mutations is exciting, but further investigations are necessary to shed light on the relative impact of individual mutations. This was now added to the discussion (page 35, line 1-5).

8. The authors show that the CHI3L1 gene is a good marker for TM network connectivity, specifically expressed in GBs and perform a series of functional experiments involving both blocking using antibodies and overexpression, showing that it is driver of network formation. Proteomic and phosphoproteomic analysis of overexpressing cells identified differentially expressed and phosphorylated proteins, identifying GAP43, which is associated with filament stabilizations. However, an expression analysis would also indicate whether the overexpression of CHI3L1 drives the connectivity expression signature and to what extent. Such information would be very useful, considering the thoroughness of the work far and it is not clear why this was not carried out (or this reviewer missed it).

Reply: We are in line with the reviewer that additional data on transcriptome data of CHI3L1 Ctrl and OE cells would add value. We now conducted a bulk RNA-Seq analysis comparing CHI3L1

OE with Ctrl PDGCs, have identified several DEGs (new Fig. 7m) and correlated their fold-changes with corresponding DEPs (new Fig. 7n).

9. Overall the main figures tend to be overloaded and difficult to read in an A4 format. Some sub figures can be moved to supplementary. For example Figure 5h is a main figure on its own and hard to read in the current format.

Reply: Several panels from the main figures including former Fig. 5h have now been shifted to the Extended Data Figures, which increased the readability of the manuscript.

Reviewer #4 (Remarks to the Author)

In this present manuscript, Hai et al. established a gene expression signature of tumor network connectivity utilizing single-cell RNA-sequencing xenografted primary glioblastoma cells, that could serve to determine the interconnectivity in individual tumors. Indeed, it is quite interesting work, I have my specific comments as below:

1. The genetic background of the PDGCL should be clarified, otherwise we cannot determine whether the genomic alterations of the tumors impact the connection

Reply: We have now added the methylation profiles (new Extended Data Fig. 1a), CNV profiles (new Extended Data Fig. 1a, Supplementary Data Table 1) as well as information about the mutational fingerprints (new Extended Data Fig. 1b) and transcriptomic landscape (Supplementary Table 1) of all used patient-derived glioblastoma cell lines (PDGCLs).

2. The FACS data should be presented, how did the author obtain the SR101 high or low cells, the cell purity should be clarified.

Reply: We have now added a scheme illustrating how the FACS gating and respective sorting of patient-derived glioblastoma cells (PDGCs) was carried out (new Extended Data Fig. 1j-l). The purity of the sorted and sequenced PDGCs is assumed to be very high as we xenografted PDGCLs overexpressing tGFP instead of wild-type PDGCLs. This allowed us to re-identify and isolate xenografted PDGCs via FACS after tumor resection. We sorted only tGFP^{high}, SR101^{high} (highly connected) and tGFP^{high}, SR101^{low} (lowly connected) PDGCs. This might have led to the loss of a low number of PDGCs due to tGFP signal fading during the phase of tumor growth, but precludes a significant number of host cells from sequencing. Thus, dilution of the connectivity signature through transcripts from normal host cells in our opinion is very unlikely.

3. Did the author perform any biological or technical repeats?

Reply: We thank the reviewer for commenting on this. The SR101 xenograft experiments were conducted after collaborative planning with the biostatistics department and PDGCs obtained from three different PDGCLs and n = 3 mice per PDGCL were used for sequencing. In general, information about cohort sizes, biological and technical replicates, independent experiments and the total number of datapoints analyzed is provided in the figure and table legends.

4. Missing value handling needs to be carefully described and justified.

Reply: The intensity distribution and cumulative fraction of proteins in the proteomics and phosphoproteomics datasets (Reviewer Fig. 5) showed that proteins with missing values had lower intensities. This observation suggests that the proteins with missing values were below the detection limit. To handle the missing values, we employed the deterministic minimum (MinDet)

method for imputation. This method replaced each missing value with the smallest detectable intensity (0.01 quantile) observed within each sample. Further descriptions about missing value handling were added to the “Material and Methods” part (page 78, lines 2-7).

Reviewer Fig. 5: Missing value handling. The densities (top two panels) and cumulative fractions (bottom two panels) of protein intensity in proteomics (left two panels) and phosphoproteomics (right two panels) datasets are plotted for proteins with (blue line) and without (red line) missing values.

5. The gene connective signatures were inferred based on transcriptomic data. However, since the proteins are the final executors, could the author perform further analysis and illustrate how the connective signatures expressed at proteomic level?

Reply: We appreciate this idea and now correlated connectivity signature scores derived from RNA-Seq and proteomics²⁴. We found a high correlation ($R = 0.85$; new Extended Data Fig. 8f)

suggesting that both transcriptome sequencing and proteomics readouts are feasible for connectivity signature scoring in clinical datasets.

Corroborating this, CHI3L1 overexpression increased connectivity signature scores in both RNA-Seq and proteomic derived datasets obtained from three PDGCLs (new Fig. 7o). We further demonstrated that patients with relatively higher connectivity signature scores, as measured by RNA-Seq, were characterized by relatively higher CHI3L1 protein levels (Fig. 6j-l).

6. How many legends and receptors were defined as connective signatures?

Reply: Following the reviewer's excellent suggestion, we now investigated ligand-receptor interactions of glioblastoma cells (GCs) within several GB cell states and healthy cells in our SR101 xenograft and patient single cell datasets using CellChat²⁵.

10% of the connectivity signature genes were found in the Cellchat ligand-receptor database (NMB, AGT, TNFRSF12A, SPARC, PTN, EGFR, DLL3 and DLL1).

In TM-unconnected PDGCs the total number of ligand-receptor interactions was relatively lower (Reviewer Fig. 6a,b). We particularly observed a higher interaction strength of the NOTCH pathway (Reviewer Fig. 6c-e) and an upregulation of the signaling pathway ligands DLL1 and DLL3, which are both part of the connectivity signature. These findings were in line with *in situ* experiments by our group correlating high NOTCH signaling with a rather unconnected phenotype in the perivascular niche⁶. Translating it to potential molecular interactions between different cell types, endothelial cells and pericytes had the highest number of interactions with NPC1 cells with low connectivity signature score (low_NPC1, Reviewer Fig. 6f), the least connected GCs in our dataset. This strengthens a potential suppression of TM network formation in the perivascular niche by vascular cells.

Moreover, exclusively unconnected PDGCs were characterized by strong signaling strength of the migration, invasion and axon guidance-correlated SEMA6 and NRXN ligand/receptor pairs (Reviewer Fig. 6g, ^{26,27}). This underlines the concept that unconnected GCs first invade the brain¹ and get ultimately connected with neurons²⁸. Specifically NLGN3, the ligand of NRXN identified in our CellChat analysis, has been shown to induce the formation of neuron-glioma synaptogenesis²⁹.

Reviewer Fig. 6: Cell-cell communication analysis using CellChat. a-e, scRNA-Seq dataset of SR101 xenograft models. **a**, The number of inferred interactions among cell states in TM-connected (Left) or TM-unconnected (Right) samples. **b**, The total number of inferred interactions and interaction strength between TM-connected and TM-unconnected samples. **c**, Relative interaction strength of each signaling pathway between TM-connected and TM-unconnected samples. **d**, The number of inferred interactions of NOTCH signaling pathway among cell states in TM-connected (Left) or TM-unconnected (Right) samples. **e**, The expression level of genes in NOTCH signaling pathway. **f**, snRNA-Seq dataset of GB patients. The number of inferred interactions among cell states/types in patient samples. The cell states are further separated into High/Medium/Low groups by connectivity signature scores (e.g., ‘High_AC’ indicates a group of cells was assigned to AC cell state and also a had high connectivity signature score). **g**, Overall signaling patterns of connected and unconnected PDGCs.

7. P-values seem to be unadjusted. It is required to adjust them for multiple hypothesis testing.

Reply: The false discovery rate (FDR) approach was used to adjust for multiple hypothesis testing whenever the latter was necessary to apply. We now clearly stated this in the method section “Statistics” (page 78, line 22 – page 79, line 6) and in the respective method sections.

8. The author performed proteomic and phosphoproteomic analysis to investigate the pathways that involved in CHI3L1 driven connectivity. However, the exact mechanism is still unclear. Further functional experiments should be conducted.

Reply: We sought to shed light on the functional mechanism of CHI3L1-driven connectivity by molecular characterization of PDGCLs using RNA-Seq, proteomics and phosphoproteomics.

To generate a holistic overview about the mechanism of CHI3L1-driven TM network induction, we predicted kinases responsible for the phosphorylation sites identified by our analysis and identified MEK/ERK and AKT signaling as being upregulated in CHI3L1 OE PDGCs (Extended Data Fig. 10c). This was in line with previous reports: PDGC-stimulation with CHI3L1 resulted in elevated activation of these pathways^{12,30}, whereas inhibition of CHI3L1 with a small molecule drug conferred the opposite effect¹³. Moreover, a phosphoproteomic characterization of neuronal growth cones independently identified elevated ERK activation¹⁴. We next focused on differences in the total gene expression. STAT3 was predicted to be activated and to serve as a signaling hub in CHI3L1 OE PDGCLs (Extended Data Fig. 10d), which matched other studies^{30,31}.

Interestingly, STAT3 signaling was previously correlated with the observed (Fig. 7 p, Extended Fig. 10a,b) elevated GAP43(pS41) levels³², pinpointing towards a hypothetical link between these two findings. Importantly, GAP43 is physically tightly associated with the growth cone membrane skeleton, a structure that confers morphology changes by spatially regulating actin polymerization^{33–36}. Phosphorylation of GAP43 at Serine41 was shown to induce subcellular localization changes of GAP43 resulting in higher plasticity³⁷ and induced neurite branching and outgrowth^{33,34,31,38,39}. Specifically, phosphorylation of GAP43 on Serine41 was found at the growth cones of actively extending axons³³ and where “growth cones makes productive, stable contacts with other cells”³³. In contrast, unphosphorylated GAP43 was found in retracting cones³³. This effects were based on the fact that GAP(pS41) stabilizes long actin filaments³³.

We added these exciting findings to the result section (page 29, lines 1-8).

9. The web app link was not accessible to me.

Reply: The link and login data are as following:

URL: <https://connectivity-glioma.dkfz.de/>

username: reviewer

password: CQMYGYSDSS

The tool will be made publicly available after acceptance of the manuscript.

References

1. Venkataramani, V. *et al.* Glioblastoma hijacks neuronal mechanisms for brain invasion. *Cell*; 10.1016/j.cell.2022.06.054 (2022).
2. Ratliff, M. *et al.* Individual glioblastoma cells harbor both proliferative and invasive capabilities during tumor progression. *Neuro-oncology*; 10.1093/neuonc/noad109 (2023).
3. Hausmann, D. *et al.* Autonomous rhythmic activity in glioma networks drives brain tumour growth. *Nature* 613, 179–186; 10.1038/s41586-022-05520-4 (2023).
4. Gilbertson, R. J. & Rich, J. N. Making a tumour's bed: glioblastoma stem cells and the vascular niche. *Nature reviews. Cancer* 7, 733–736; 10.1038/nrc2246 (2007).
5. Charles, N. *et al.* Perivascular nitric oxide activates notch signaling and promotes stem-like character in PDGF-induced glioma cells. *Cell stem cell* 6, 141–152; 10.1016/j.stem.2010.01.001 (2010).
6. Jung, E. *et al.* Tumor cell plasticity, heterogeneity, and resistance in crucial microenvironmental niches in glioma. *Nature communications* 12, 1014; 10.1038/s41467-021-21117-3 (2021).
7. Mohar, B. *et al.* Brain-wide measurement of protein turnover with high spatial and temporal resolution (2022).
8. Varn, F. S. *et al.* Glioma progression is shaped by genetic evolution and microenvironment interactions. *Cell* 185, 2184–2199.e16; 10.1016/j.cell.2022.04.038 (2022).
9. Venkataramani, V. *et al.* Disconnecting multicellular networks in brain tumours. *Nature reviews. Cancer* 22, 481–491; 10.1038/s41568-022-00475-0 (2022).
10. Osswald, M. *et al.* Brain tumour cells interconnect to a functional and resistant network. *Nature* 528, 93–98; 10.1038/nature16071 (2015).
11. Jung, E. *et al.* Tweety-Homolog 1 Drives Brain Colonization of Gliomas. *The Journal of neuroscience : the official journal of the Society for Neuroscience* 37, 6837–6850; 10.1523/JNEUROSCI.3532-16.2017 (2017).
12. Wurm, J. *et al.* Astroglial Release of Pro-Oncogenic Chitinase 3-Like 1 Causes MAPK Signaling in Glioblastoma. *Cancers* 11; 10.3390/cancers11101437 (2019).
13. Lee, Y. S. *et al.* A small molecule targeting CHI3L1 inhibits lung metastasis by blocking IL-13R α 2-mediated JNK-AP-1 signals. *Molecular oncology* 16, 508–526; 10.1002/1878-0261.13138 (2022).

14. Kawasaki, A. *et al.* Growth Cone Phosphoproteomics Reveals that GAP-43 Phosphorylated by JNK Is a Marker of Axon Growth and Regeneration. *iScience* 4, 190–203; 10.1016/j.isci.2018.05.019 (2018).
15. Wang, Q. *et al.* Tumor Evolution of Glioma-Intrinsic Gene Expression Subtypes Associates with Immunological Changes in the Microenvironment. *Cancer cell* 32, 42–56.e6; 10.1016/j.ccell.2017.06.003 (2017).
16. Neftel, C. *et al.* An Integrative Model of Cellular States, Plasticity, and Genetics for Glioblastoma. *Cell* 178, 835–849.e21; 10.1016/j.cell.2019.06.024 (2019).
17. Leu, B., Koch, E. & Schmidt, J. T. GAP43 phosphorylation is critical for growth and branching of retinotectal arbors in zebrafish. *Developmental neurobiology* 70, 897–911; 10.1002/dneu.20829 (2010).
18. He, L. *et al.* Analysis of the brain mural cell transcriptome. *Scientific reports* 6, 35108; 10.1038/srep35108 (2016).
19. Ravi, V. M. *et al.* Spatially resolved multi-omics deciphers bidirectional tumor-host interdependence in glioblastoma. *Cancer cell* 40, 639–655.e13; 10.1016/j.ccell.2022.05.009 (2022).
20. Johnson, K. C. *et al.* Single-cell multimodal glioma analyses identify epigenetic regulators of cellular plasticity and environmental stress response. *Nature genetics* 53, 1456–1468; 10.1038/s41588-021-00926-8 (2021).
21. Aran, D., Sirota, M. & Butte, A. J. Systematic pan-cancer analysis of tumour purity. *Nature communications* 6, 8971; 10.1038/ncomms9971 (2015).
22. Newman, A. M. *et al.* Determining cell type abundance and expression from bulk tissues with digital cytometry. *Nature biotechnology* 37, 773–782; 10.1038/s41587-019-0114-2 (2019).
23. Verhaak, R. G. W. *et al.* Integrated genomic analysis identifies clinically relevant subtypes of glioblastoma characterized by abnormalities in PDGFRA, IDH1, EGFR, and NF1. *Cancer cell* 17, 98–110; 10.1016/j.ccr.2009.12.020 (2010).
24. Wang, L.-B. *et al.* Proteogenomic and metabolomic characterization of human glioblastoma. *Cancer cell* 39, 509–528.e20; 10.1016/j.ccell.2021.01.006 (2021).
25. Jin, S. *et al.* Inference and analysis of cell-cell communication using CellChat. *Nature communications* 12, 1088; 10.1038/s41467-021-21246-9 (2021).
26. Day, B. W., Stringer, B. W. & Boyd, A. W. Eph receptors as therapeutic targets in glioblastoma. *British journal of cancer* 111, 1255–1261; 10.1038/bjc.2014.73 (2014).
27. Angelucci, C., Lama, G. & Sica, G. Multifaceted Functional Role of Semaphorins in Glioblastoma. *International journal of molecular sciences* 20; 10.3390/ijms20092144 (2019).
28. Venkataramani, V. *et al.* Glutamatergic synaptic input to glioma cells drives brain tumour progression. *Nature* 573, 532–538; 10.1038/s41586-019-1564-x (2019).
29. Venkatesh, H. S. *et al.* Electrical and synaptic integration of glioma into neural circuits. *Nature* 573, 539–545; 10.1038/s41586-019-1563-y (2019).

30. Guetta-Terrier, C. *et al.* Chi311 Is a Modulator of Glioma Stem Cell States and a Therapeutic Target in Glioblastoma. *Cancer research* 83, 1984–1999; 10.1158/0008-5472.CAN-21-3629 (2023).
31. Tran, H. T. *et al.* Chitinase 3-like 1 synergistically activates IL6-mediated STAT3 phosphorylation in intestinal epithelial cells in murine models of infectious colitis. *Inflammatory bowel diseases* 20, 835–846; 10.1097/MIB.000000000000033 (2014).
32. Hung, C.-C. *et al.* Astrocytic GAP43 Induced by the TLR4/NF- κ B/STAT3 Axis Attenuates Astrogliosis-Mediated Microglial Activation and Neurotoxicity. *The Journal of neuroscience : the official journal of the Society for Neuroscience* 36, 2027–2043; 10.1523/JNEUROSCI.3457-15.2016 (2016).
33. He, Q., Dent, E. W. & Meiri, K. F. Modulation of actin filament behavior by GAP-43 (neuromodulin) is dependent on the phosphorylation status of serine 41, the protein kinase C site. *The Journal of neuroscience : the official journal of the Society for Neuroscience* 17, 3515–3524; 10.1523/JNEUROSCI.17-10-03515.1997 (1997).
34. Ditlevsen, D. K., Povlsen, G. K., Berezin, V. & Bock, E. NCAM-induced intracellular signaling revisited. *Journal of neuroscience research* 86, 727–743; 10.1002/jnr.21551 (2008).
35. Callender, J. A. & Newton, A. C. Conventional protein kinase C in the brain: 40 years later. *Neuronal signaling* 1, NS20160005; 10.1042/NS20160005 (2017).
36. Tejero-Díez, P., Rodríguez-Sánchez, P., Martín-Cófreces, N. B. & Díez-Guerra, F. J. bFGF stimulates GAP-43 phosphorylation at ser41 and modifies its intracellular localization in cultured hippocampal neurons. *Molecular and cellular neurosciences* 16, 766–780; 10.1006/mcne.2000.0915 (2000).
37. Holahan, M. GAP-43 in synaptic plasticity: molecular perspectives. *RRBC*, 137; 10.2147/RRBC.S73846 (2015).
38. Holahan, M. R. A Shift from a Pivotal to Supporting Role for the Growth-Associated Protein (GAP-43) in the Coordination of Axonal Structural and Functional Plasticity. *Frontiers in cellular neuroscience* 11, 266; 10.3389/fncel.2017.00266 (2017).
39. Benowitz, L. I. & Routtenberg, A. GAP-43: an intrinsic determinant of neuronal development and plasticity. *Trends in neurosciences* 20, 84–91; 10.1016/s0166-2236(96)10072-2 (1997).

REVIEWERS' COMMENTS

Reviewer #1 (Remarks to the Author):

I appreciate the efforts the authors have taken to address my comments and I feel that the quality of the manuscript has significantly improved. The authors have done a number of important new experiments and analyses as well as refined the main text.

Reviewer #2 (Remarks to the Author):

The authors have addressed all of my concerns.

Reviewer #4 (Remarks to the Author):

The authors have adequately addressed all my comments.